# Intriguing Properties of Input-dependent Randomized Smoothing

## Abstract

Randomized smoothing is currently considered the state-of-the-art method to obtain certifiably robust classifiers. Despite its remarkable performance, the method is associated with various serious problems such as "certified accuracy waterfalls", certification vs. accuracy trade-off, or even fairness issues. Input-dependent smoothing approaches have been proposed with intention of overcoming these flaws. However, we demonstrate that these methods lack formal guarantees and so the resulting certificates are not justified. We show that the input-dependent smoothing, in general, suffers from the curse of dimensionality, forcing the variance function to have low semi-elasticity. On the other hand, we provide a theoretical and practical framework that enables the usage of input-dependent smoothing even in the presence of the curse of dimensionality, under strict restrictions. We present one concrete design of the smoothing variance and test it on CIFAR10 and MNIST. Our design solves some of the problems of classical smoothing and is formally underlined, yet further improvement of the design is still necessary.

## 1 Introduction

Deep neural networks are one of the dominating recently used machine learning methods. They achieve state-of-the-art performance in a variety of applications like computer vision, natural language processing, and many others. The key property that makes neural networks so powerful is their expressivity (Gühring et al., 2020). However, as a prize, they possess a weakness - a vulnerability against *adversarial attacks* (Szegedy et al., 2013; Biggio et al., 2013). The adversarial attack on a sample $x$ is a point $x'$ such that the distance $d(x', x)$ is small, yet the model $f$'s predictions on $x$ and $x'$ differ. Such examples are often easy to construct, for example by optimizing for a change in prediction $f(x)$ (Biggio et al., 2013). Even worse, these attacks are present even if the model's prediction on $x$ is unequivocal.

This property is highly undesirable because in several sensitive applications, misclassifying a sample just because it does not follow the natural distribution might lead to serious and harmful consequences. A well-known example would be a sticker placed on a traffic sign, which could possibly confuse the self-driving car and cause an accident (Eykholt et al., 2018). To prevent this behaviour, the robustness of classifiers against adversarial examples has begun to be a strongly discussed topic. Though many methods claim to provide robust classifiers, just some of them are *certifiably* robust, i.e. the robustness is mathematically guaranteed. The certifiability turns out to be essential since more sophisticated attacks can break empirical defenses (Carlini & Wagner, 2017).

Currently, the dominating method to achieve the certifiable robustness is *randomized smoothing* (RS). This clever idea to get rid of adversarial examples using randomization of input was introduced by Lecuyer et al. (2019) and Li et al. (2019) and fully formalized and improved in Cohen et al. (2019). Let $f$ be a classifier classifying inputs $x \in \mathbb{R}^N$ as one of the classes $C \in \mathcal{C}$. Assume now a random deviation $\epsilon \sim \mathcal{N}(0, \sigma^2 I)$. The *smoothed classifier* $g$, made of $f$, is defined as: $g(x) = \arg\max_C \mathbb{P}(f(x + \epsilon) = C)$, for $C \in \mathcal{C}$. In other words, the smoothed classifier classifies a class that has the highest probability under the sampling of $f(x + \epsilon)$. Consequently, an adversarial attack $x'$ on $f$ is less dangerous for $g$, because $g$ does not look directly at $x'$, but rather at its whole neighborhood, in a weighted manner. This way, we can get rid of local artifacts that $f$ possesses – thus the name "smoothing". It turns out, that $g$ enjoys strong robustness properties against attacks

bounded by a specifically computed $l_2$-norm threshold, especially if $f$ is trained under a Gaussian noise augmentation (Cohen et al., 2019).

Unfortunately, since the introduction of the RS, several serious problems were reported to be connected to the technique. Cohen et al. (2019) mention two of them. First is the usage of lower confidence bounds to estimate the leading class's probability. With a high probability, this leads to smaller reported certified radiuses in comparison with the true ones. Moreover, it yields a theoretical threshold, which upper-bounds the maximal possible certified radius and causes the "certified accuracy waterfalls", significantly decreasing the certified accuracy. This problem is particularly pronounced for small levels of the used smoothing variance $\sigma^2$, which motivates to use larger variance. Second, RS possesses a robustness vs. accuracy trade-off problem. The bigger $\sigma$ we use as the smoothing variance, the smaller *clean accuracy* will the smoothed classifier have. This motivates to use rather smaller levels of $\sigma$. Third, as pointed out by Mohapatra et al. (2020a), RS smoothens the decision boundary of $f$ in such a way, that bounded or convex regions begin to *shrink* as $\sigma$ increases, while the unbounded and anti-convex regions expand. This, as the authors empirically demonstrate, creates a disbalance in class-wise accuracies of $g$ and causes serious fairness issues. Therefore, again, the smaller values of $\sigma$ are more preferable. See Appendix A for a detailed discussion.

Clearly, the usage of a global, constant $\sigma$ is suboptimal. For the samples close to the decision boundary, we want to use small $\sigma$, so that $f$ and $g$ have similar decision boundaries and the expressivity of $f$ is not lost (where not necessary). On the other hand, far from the decision boundary of $f$, where the probability of the dominant class is close to 1, we need bigger $\sigma$ to avoid the severe under-certification (see Appendix A). Together, using a non-constant $\sigma(x)$ rather than constant $\sigma$, a suitable smoothing variance could be used to achieve optimal robustness. Yet there are some works introducing this concept (see Appendix C), most of them lack mathematical reasoning about the correctness of their method, which, as we show, turns out to be critical.

To support this argumentation, we present a toy example. We train a network on a 2D dataset of a circular shape with the classes being two complementary sectors, where one is of a very small angle. In Figure 1 we show the difference between constant and input-dependent $\sigma$. Using non-constant $\sigma(x)$ defined in Equation 1, we obtain an improvement both in terms of the certified radiuses as well as clean accuracy. For more details see Appendix A.

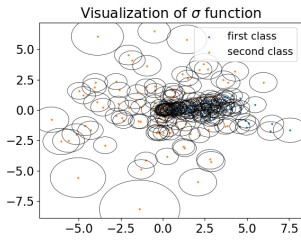 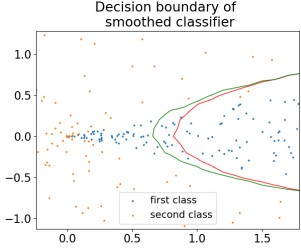 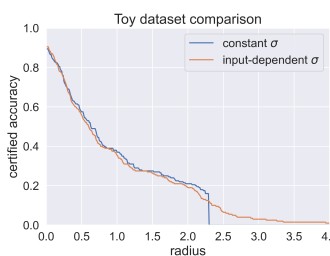

Figure 1: Motivating toy experiment. The constant $\sigma = 0.6$ and the input-dependent $\sigma(x)$ equal in average to the constant $\sigma$ are used. **Left:** Dataset and the variance function depicted as circles with the radius equal to $\sigma(x)$ and centers at the data points. **Middle:** Zoomed in part of the dataset and decision boundaries of the smoothed classifiers with constant $\sigma$ (red) and input-dependent $\sigma(x)$ (green). Note that we recover a part of the misclassified data points by using a more appropriate smoothing strength close to the decision boundary. **Right:** Certified accuracy plot. The waterfall effect vanishes since the points far from the decision boundary are certified with a correspondingly large $\sigma(x)$.

The main contributions of this work are fourfold. First, we generalize the methodology of Cohen et al. (2019)'s work for the case of input-dependent RS (IDRS), obtaining useful and important insights about how to use the Neyman-Pearson lemma in this, general, case. Second and most importantly, we show that the IDRS suffers from the curse of dimensionality in the sense that the semi-elasticity coefficient $r$ of $\sigma(x)$ (that is $|\log(\sigma(x_0)) - \log(\sigma(x_1))| \leq r\|x_0 - x_1\| \; \forall x_0, x_1 \in \mathbb{R}^N$) in a high-dimensional setting is restricted to be very small. This means, that even if we wanted to vary the $\sigma(x)$ significantly with varying $x$, we can't. The maximal reasonable speed of change of $\sigma(x)$ turns out to be almost too small to handle, especially in high dimensions. Third, in contrast,

we also study the conditions on $\sigma(x)$ under which it is applicable in high-dimensional regime and prepare a theoretical framework necessary to build an efficient certification algorithm. We are the first to do so for $\sigma(x)$ functions, which are not locally constant (as in Wang et al. (2021)). Finally, we provide a concrete design of the $\sigma(x)$ function and test it extensively and compare it to the classical RS on the CIFAR10 and MNIST datasets. We discuss to what extent the method treats the issues mentioned above.

## 2 INPUT-DEPENDENT RS AND THE CURSE OF DIMENSIONALITY

Let $\mathcal{C}$ be the set of classes, $f : \mathbb{R}^N \to \mathcal{C}$ a classifier (reffered to as the *base* classifier), $\sigma : \mathbb{R}^N \to \mathbb{R}$ a non-negative function and $\mathcal{P}(\mathcal{C})$ a set of distributions over $\mathcal{C}$. Then we call $G_f : \mathbb{R}^N \to \mathcal{P}(\mathcal{C})$ the *smoothed class probability predictor*, if $G_f(x)_C = \mathbb{P}(f(x + \epsilon) = C)$, where $\epsilon \sim \mathcal{N}(0, \sigma(x)^2 I)$ and $g_f : \mathbb{R}^N \to \mathcal{C}$ is called *smoothed classifier* if $g_f(x) = \arg\max_C G_f(x)_C$, for $C \in \mathcal{C}$. We will omit the subscript $f$ in $g_f$ often, since it is usually clear from the context, to which base classifier the $g$ corresponds. Furthermore, let $A := g(x)$ refer to the most likely class under the random variable $f(\mathcal{N}(x, \sigma^2 I))$, while $B$ denote the second most likely class. Define $p_A = G_f(x)_A$ and $p_B = G_f(x)_B$ as the respective probabilities. It is important to note that in practice, it is impossible to estimate $p_A$ and $p_B$ precisely. Instead, $p_A$ is estimated as a lower confidence bound (LCB) of the relative occurence of class $A$ in $f$'s predictions given certain number of Monte-Carlo samples $n$ and a confidence level $\alpha$ and the estimate is denoted as $\underline{p_A}$. We use the exact Clopper-Pearson interval for estimation of the LCB. Similarly for $p_B$. We work with $l_2$-norms denoted as $\|x\|$.

First of all, we summarize the main steps in the derivation of certified radius using any method that relies on a use of Neyman-Pearson lemma.

1. For a potential adversary $x'$ specify the *worst-case* classifier $f^*$, such that $\mathbb{P}(f^*(\mathcal{N}(x, \sigma^2 I)) = A) = p_A$, while $\mathbb{P}(f^*(\mathcal{N}(x', \sigma^2 I)) = B)$ is maximized.

2. Express the probability $G_{f^*}(x')_B$ as a function depending on $x'$.

3. Determine the conditions on $x'$ (possibly related to $\|x - x'\|$) for which this probability is $\leq 0.5$. From these condtions, derive the certified radius.

Cohen et al. (2019) proceeded in this way to obtain a tight certified radius $R = \frac{\sigma}{2}(\Phi^{-1}(\underline{p_A}) - \Phi^{-1}(\overline{p_B}))$. Unfortunately, their result is not directly applicable to the input-dependent case. The constant $\sigma$ simplifies the derivation of $f^*$ that turns out to be a linear classifier. This is not the case for non-constant $\sigma(x)$ anymore. Therefore, we need to generalize the methodology of Cohen et al. (2019). We put $p_B = 1 - p_A$ for simplicity (yet it is not necessary to assume this, see Appendix B.5). Let $x_0$ be the point to certify, $x_1$ the potential adversary point, $\delta = x_1 - x_0$ the noise and $\sigma_0 = \sigma(x_0)$, $\sigma_1 = \sigma(x_1)$ the standard deviations used in $x_0$ and $x_1$ respectively. Furthermore, let $f_i$ be a density and $\mathbb{P}_i$ a probability measure corresponding to $\mathcal{N}(x_i, \sigma_i^2 I)$, $i \in \{0, 1\}$.

**Lemma 1.** Out of all possible classifiers $f$ such that $G_f(x)_B \leq p_B = 1 - p_A$, the one, for which $G_f(x + \delta)_B$ is maximized is the one, which predicts class $B$ in a region determined by the likelihood ratio:

$$B = \left\{ x \in \mathbb{R}^N : \frac{f_1(x)}{f_0(x)} \geq \frac{1}{r} \right\},$$

where $r$ is fixed, such that $\mathbb{P}_0(B) = p_B$. Note, that we use $B$ to denote both the class and the region of that class.

We use this lemma to compute the decision boundary of the worst-case classifier $f^*$.

**Theorem 2.** If $\sigma_0 > \sigma_1$, then $B$ is a $N$-dimensional ball with the center at $S_>$ and radius $R_>$:

$$S_> = x + \frac{\sigma_0^2}{\sigma_0^2 - \sigma_1^2}\delta, \; R_> = \sqrt{\frac{\sigma_0^2\sigma_1^2}{(\sigma_0^2 - \sigma_1^2)^2}\|\delta\|^2 + 2N\frac{\sigma_0^2\sigma_1^2}{\sigma_0^2 - \sigma_1^2}\log\left(\frac{\sigma_0}{\sigma_1}\right) + \frac{2\sigma_0^2\sigma_1^2}{\sigma_0^2 - \sigma_1^2}\log(r)}.$$

If $\sigma_0 < \sigma_1$, then $B$ is the complement of a $N$-dimensional ball with the center at $S_<$ and radius $R_<$:

$$S_< = x - \frac{\sigma_0^2}{\sigma_1^2 - \sigma_0^2}\delta, \; R_< = \sqrt{\frac{2\sigma_0^4 - \sigma_0^2\sigma_1^2}{(\sigma_1^2 - \sigma_0^2)^2}\|\delta\|^2 + 2N\frac{\sigma_0^2\sigma_1^2}{\sigma_1^2 - \sigma_0^2}\log\left(\frac{\sigma_1}{\sigma_0}\right) - \frac{2\sigma_0^2\sigma_1^2}{\sigma_1^2 - \sigma_0^2}\log(r)}.$$

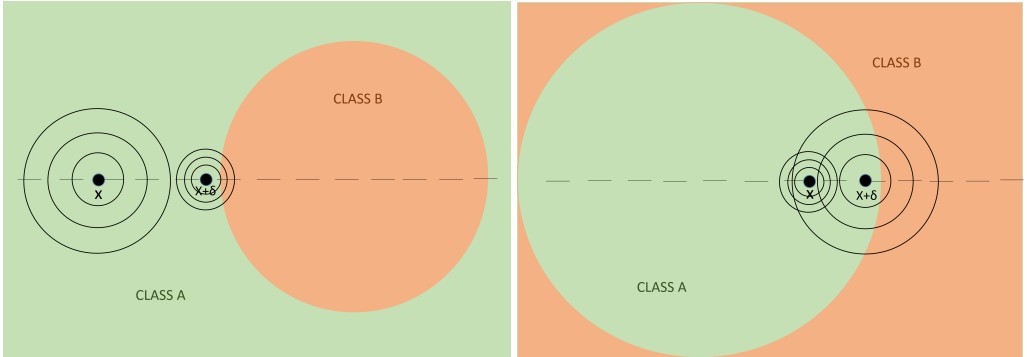

Figure 2: Decision regions of the worst-case classifier $f^*$. **Left:** $\sigma_0 > \sigma_1$ **Right:** $\sigma_0 < \sigma_1$

As we depict on Figure 2, both resulting balls are centered on the line connecting $x_0, x_1$. Moreover, the centers of the balls are always further from $x_0$, than $x_1$ is from $x_0$ (even in the case $\sigma_0 < \sigma_1$). In both cases, it depends on $p_A$ (since $r$ is fixed such that $\mathbb{P}_0(B) = p_B$) and the ball can, but might not cover $x_0$ and/or $x_1$. Note that if $\sigma_0 = \sigma_1$, what can happen even in input-dependent regime, the worst-case classifier is the half-space described by Cohen et al. (2019).

To compute the probability of a ball under a probability measure with an isotropic Gaussian density is far more challenging than to compute the probability of a half-space. In fact, there is no closed-form formula for such probability. However, this probability is connected to the non-central chi-square distribution (NCCHSQ). More precisely, the probability of an $N$-dimensional ball centered at $z$ with radius $r$ under $\mathcal{N}(0, I)$ can be expressed as a cumulative distribution fucntion (cdf) of NCCHSQ with $N$ degrees of freedom, non-centrality parameter $\|z\|^2$ and argument $r^2$. With this knowledge, we can express $\mathbb{P}_0(B)$ and $\mathbb{P}_1(B)$ in terms of the cdf of NCCHSQ as follows.

**Theorem 3.**

$$\mathbb{P}_0(B) = \chi_N^2\left(\frac{\sigma_0^2}{(\sigma_0^2 - \sigma_1^2)^2}\|\delta\|^2, \frac{R_{<,>}^2}{\sigma_0^2}\right), \mathbb{P}_1(B) = \chi_N^2\left(\frac{\sigma_1^2}{(\sigma_0^2 - \sigma_1^2)^2}\|\delta\|^2, \frac{R_{<,>}^2}{\sigma_1^2}\right),$$

where the sign $<$ or $>$ is chosen according to the inequality between $\sigma_0$ and $\sigma_1$.

Note, that both Theorem 2 and Theorem 3 work well also for $\delta = 0$. In this case, we encounter a ball centered at $x_0 = x_1$ and all the cdf functions become cdf functions of *central* chi-squared.

We expressed probabilities of the worst-case class $B$'s decision region using the cdf of NCCHCSQ. Now, how do we do the certification? We start with the certification just for two points, $x_0$ and $x_1$. We question, under which circumstances can $x_1$ be certified from the point of view of $x_0$. Having $x_0$, $p_A$ and $\sigma_0 > \sigma_1$, we can obtain such $R$, that $\mathbb{P}_0(B) = \chi_N^2\left(\|\delta\|^2\sigma_0^2/(\sigma_0^2 - \sigma_1^2)^2, R^2\right) = 1 - p_A = p_B$ simply by putting it into the quantile function: $R^2 = \chi_{N,qf}^2\left(\|\delta\|^2\sigma_0^2/(\sigma_0^2 - \sigma_1^2)^2, 1 - p_A\right)$. Then, we can substitute into $\mathbb{P}_1(B) = \chi_N^2\left(\|\delta\|^2\sigma_1^2/(\sigma_0^2 - \sigma_1^2)^2, R^2\sigma_0^2/\sigma_1^2\right)$. This way, we obtain $\mathbb{P}_1(B)$ and can judge, whether $\mathbb{P}_1(B) < 1/2$ or not. Similar computation can be done if $\sigma_0 < \sigma_1$. Denote $a := \|\delta\|$. We can express the $\mathbb{P}_1(B)$ more simply for $\sigma_0 > \sigma_1$ as

$$\xi_>(a) := \mathbb{P}_1(B) = \chi_N^2\left(\frac{\sigma_1^2}{(\sigma_0^2 - \sigma_1^2)^2}a^2, \frac{\sigma_0^2}{\sigma_1^2}\chi_{N,qf}^2\left(\frac{\sigma_0^2}{(\sigma_0^2 - \sigma_1^2)^2}a^2, 1 - p_A\right)\right)$$

and for $\sigma_0 < \sigma_1$ as

$$\xi_<(a) := \mathbb{P}_1(B) = 1 - \chi_N^2\left(\frac{\sigma_1^2}{(\sigma_1^2 - \sigma_0^2)^2}a^2, \frac{\sigma_0^2}{\sigma_1^2}\chi_{N,qf}^2\left(\frac{\sigma_0^2}{(\sigma_1^2 - \sigma_0^2)^2}a^2, p_A\right)\right).$$

With this in mind, if we have $x_0, x_1, p_A, \sigma_0, \sigma_1$, then we can certify $x_1$ w.r.t $x_0$ simply by choosing the correct sign $(<, >)$ and computing $\xi_<(\|x_0 - x_1\|)$ or $\xi_>(\|x_0 - x_1\|)$, comparing them with $0.5$. The sample plots of these $\xi$ functions can be found in Appendix B.

Now, we are ready to discuss the curse of dimensionality. The problem that arises is that having a high dimension $N$ and $\sigma_0, \sigma_1$ differing a lot from each other, $\xi$ functions are already big at 0, even

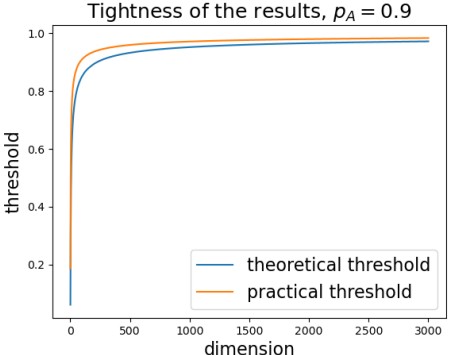
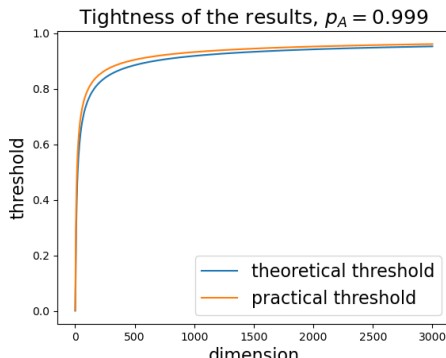

Figure 3: Plots depicting tightness of results of Theorem 4. On both figures, the biggest possible threshold of $\sigma_1/\sigma_0$ for which the condition in Theorem 4 is satisfied (theoretical threshold) and the numerically computed threshold for which $\xi_>(0)$ passes the threshold 0.5 (practical threshold) are depicted. **Left:** Plot for $p_A = 0.9$, **Right:** Plot for $p_A = 0.999$.

for considerably small $p_B$. For fixed ratio $\sigma_0/\sigma_1$ and probability $p_B$, with increasing dimension, the $\xi(0)$ increases and soon becomes bigger than 0.5. This, together with monotonicity of $\xi$ function yields that any $x_1$ cannot be certified w.r.t. $x_0$, if $\sigma_0, \sigma_1$ are used. The more dissimilar the $\sigma_0$ and $\sigma_1$ are, the smaller the dimension $N$ needs to be for this situation to occur. If we want to certify $x_1$ in a reasonable distance from $x_0$, we need to use similar $\sigma_0, \sigma_1$. This restricts the variability of $\sigma(x)$ function. We will formalize the curse of dimensionality in the following theorems. More on why the curse of dimensionality is present is in Appendix B.2.

**Theorem 4** (curse of dimensionality). Let $x_0, x_1, p_A, \sigma_0, \sigma_1, N$ be as usual. Then, the following two implications hold:

1. If $\sigma_0 > \sigma_1$ and
$$\log\left(\frac{\sigma_1^2}{\sigma_0^2}\right) + 1 - \frac{\sigma_1^2}{\sigma_0^2} < \frac{2\log(1-p_A)}{N},$$
then $x_1$ is not certified w.r.t. $x_0$.

2. If $\sigma_0 < \sigma_1$ and
$$\log\left(\frac{\sigma_1^2}{\sigma_0^2}\frac{N-1}{N}\right) + 1 - \frac{\sigma_1^2}{\sigma_0^2}\frac{N-1}{N} < \frac{2\log(1-p_A)}{N},$$
then $x_1$ is not certified w.r.t. $x_0$.

**Corollary 5** (one-sided simpler bound). Let $x_0, x_1, p_A, \sigma_0, \sigma_1, N$ be as usual and assume now $\sigma_0 > \sigma_1$. Then, if
$$\frac{\sigma_1}{\sigma_0} < \sqrt{1 - 2\sqrt{\frac{-\log(1-p_A)}{N}}},$$
then $x_1$ is not certified w.r.t $x_0$.

Note, that both Theorem 4 and Corollary 5 can be adjusted to the case where we have a separate estimate $\overline{p_B}$ and do not put $\overline{p_B} = 1 - p_A$ (see Appendix B.5). We must emphasize, that the bounds obtained in Theorem 4 are very tight. In other words, if the ratio $\frac{\sigma_1}{\sigma_0}$ is just slightly bigger than the minimal possible threshold determined in Theorem 4, $\xi_>(0)$ becomes smaller than 0.5 and similarly for $\xi_>(0)$. The reason for this is, that the only two estimates used in the proof of Theorem 4 are the estimates on the median, which are very tight and constant with respect to $N$ and the Chernoff bound, which is generally considered to be tight too and improves for larger $N$. The tightness is depicted on Figure 3, where we plot the minimal possible threshold $\sigma_1/\sigma_0$ given by Theorem 4 and minimal threshold for which $\xi_>(0) < 0.5$ as a function of $N$.

To get a better feeling about the concrete numbers, we provide a simple table, which shows the theoretical threshold values provided by Theorem 4. If $\sigma_1/\sigma_0$ is smaller than the threshold, we are not able to certify any pair of $x_0, x_1$ using $\sigma_0, \sigma_1$.

| $p_A$ | 0.9 | 0.99 | 0.999 | 0.99993 |
|---|---|---|---|---|
| MNIST | 0.946 | 0.924 | 0.908 | 0.892 |
| CIFAR10 | 0.973 | 0.961 | 0.953 | 0.945 |
| ImageNet | 0.997 | 0.995 | 0.994 | 0.993 |

Table 1: Theoretical lower-thresholds for $\sigma_1/\sigma_0$ for different data dimensions and class $A$ probabilities. The ImageNet spatial size is assumed to be 3x256x256.

Results from Table 1 are very restrictive. Assume we have a CIFAR10 sample with $p_A = 0.999$. For such a probability, *constant* $\sigma = 0.5$ is more than sufficient to guarantee the certified radius of more than 1. However, in the non-constant regime, to certify $R \geq 1$, we first need to guarantee that *no* sample within the distance of 1 from $x_0$ uses $\sigma_1 < 0.953\sigma_0$. To even strengthen this statement, note that one needs to guarantee $\sigma_1$ to be even much closer to $\sigma_0$ in practice. Why? The results of Theorem 4 lower-bound the $\xi$ functions at 0. However, since $\xi$ functions are strictly increasing (as shown in Appendix F), one usually needs $\sigma_0$ and $\sigma_1$ to be much closer to each other to guarantee $\xi$ being smaller than 0.5 at $a \gg 0$. This not only forces the $\sigma(x)$ function to have really small semi-elasticity but also makes it problematic to define a stochastic $\sigma(x)$. For more, see Appendix B.2.

To fully understand, how the curse of dimensionality affects the usage of IDRS, we mention two more significant effects. First, with increasing dimension, the average distance between samples tends to grow as $\sqrt{N}$. This enables bigger distance to change $\sigma(x)$. On the other hand, the average level of $\sigma(x)$ used (like $\sim 0.12, 0.25, \dots$) needs to be adjusted also as $\sqrt{N}$ with increasing dimension. The bigger average level of $\sigma(x)$ we use, the more is the semi-elasticity of $\sigma(x)$ restricted by Theorem 4 and Theorem 7. All together, these two effects combine in a final trend that for $\sigma_0$ and $\sigma_1$ being variances used in two random test samples, $|\sigma_0/\sigma_1 - 1|$ is restricted to go to 0 as $1/\sqrt{N}$. For detailed explanation, see Appendix B.4.

## 3 How to use input-dependent smoothing properly

As we discuss above, usage of IDRS is challenging. How can we use $\sigma(x)$ and obtain valid, mathematically justified certified radiuses? Fix some design $\sigma(x)$. If $\sigma(x)$ is not trivial, to obtain a certified radius at $x_0$, we need to go over all the possible adversaries $x_1$ in the neighborhood of $x_0$, compute $\sigma_1$ and $\xi_{<,>}(a)$. Then, the certified radius is the infimum over $\|x_0 - x_1\|$ for all *uncertified* $x_1$ points. Of course, this is a priori infeasible. Fortunately, the $\xi$ functions possess a property that helps to simplify this procedure. For convenience, we extend the notation of $\xi$ functions such that $\xi(a, \sigma_1)$ additionally denotes the dependance on the $\sigma_1$ value.

**Theorem 6.** Let $x_0, x_1, p_A, \sigma_0$ be as usual and let $\|x_0 - x_1\| = R$. Then, the following two statements hold:

1. Let $\sigma_1 \leq \sigma_0$. Then, for all $\sigma_2 : \sigma_1 \leq \sigma_2 \leq \sigma_0$, if $\xi_>(R, \sigma_2) > 0.5$, then $\xi_>(R, \sigma_1) > 0.5$.

2. Let $\sigma_1 \geq \sigma_0$. Then, for all $\sigma_2 : \sigma_1 \geq \sigma_2 \geq \sigma_0$, if $\xi_<(R, \sigma_2) > 0.5$, then $\xi_<(R, \sigma_1) > 0.5$.

Theorem 6 serves as a kind of monotonicity property. The main gain is that now, for each distance $R$ from $x_0$, we need to pick just two adversaries – the one with biggest $\sigma_1$ (if bigger than $\sigma_0$) and the one with the smallest $\sigma_1$ (if smaller than $\sigma_0$). If we cannot certify some point $x_1$ at the distance $R$ from $x_0$, then we will for sure not be able to certify at least one of the two adversaries with the most extreme $\sigma_1$ values.

This does, however, not suffice for most of the reasonable $\sigma(x)$ designs, since it might be still too hard to determine the two most extreme $\sigma_1$'s at some distance from $x_0$. Therefore, we need to assume that our $\sigma(x)$ has a bounded semi-elasticity coefficient $r$. Then we have a guarantee that $\sigma(x_0) \exp(-ra) \leq \sigma(x_1) \leq \sigma(x_0) \exp(ra)$. Thus, we can assume the worst-case extreme $\sigma_1$'s for every distance $a$ from $x_0$. Using this, we guarantee the following certified radius.

**Theorem 7.** Let $\sigma(x)$ be $r$-semi-elastic function and $x_0, p_A, N, \sigma_0$ as usual. Then, the certified radius at $x_0$ guaranteed by our method is

$$\mathrm{CR}(x_0) = \max\left\{0, \sup\left\{R \geq 0; \ \xi_>(R, \sigma_0 \exp(-rR)) < 0.5 \ \text{ and } \ \xi_<(R, \sigma_0 \exp(rR)) < 0.5\right\}\right\}.$$

Note that Theorem 7 can be adjusted to the case where we have a separate estimate $\overline{p_B}$ and do not put $\overline{p_B} = 1 - p_A$ (see Appendix B.5). Since the bigger the semi-elasticity constant of $\sigma(x)$ is, the worse certifications we obtain, it is important to estimate the constant tightly. Even with a good estimate of $r$, we still get smaller certified radiuses in comparison with using the $\sigma(x)$ exactly, but that is a prize that is inevitable for the feasibility of the method.

The practical algorithm is then very easy - we just need to pick sufficiently dense space of possible radiuses and determine the smallest, for which either $\xi_>(R, \sigma_0 \exp(-rR))$ or $\xi_<(R, \sigma_0 \exp(rR))$ becomes bigger than a half. The only non-trivial part is, how to evaluate the $\xi$ functions. For small values of $R$, the $\exp(-rR)$ is very close to 1 and from the definition of $\xi$ functions it is obvious that this results in extremely big inputs to the cdf and quantile function of NCCHSQ. To avoid numerical problems, we employ a simple hack where we assume thresholds for $\sigma_1$ such that for $R$ small enough, these thresholds are used instead of $\sigma_0 \exp(\pm rR)$. Unfortunately, the numerical stability still disables the usage of this method on really high-dimensional datasets like ImageNet. For more details on implementation, see Appendix D.

## 4 THE DESIGN OF $\sigma(x)$ AND EXPERIMENTS

The only missing ingredient to finally being able to use IDRS is the $\sigma(x)$ function. As we have seen, this function has to be $r$-semi-elastic for rather small $r$ and ideally deterministic. Yet it should at least roughly fulfill the requirements imposed by the motivation – it should possess big values for points far from the decision boundary of $f$ and rather small for points close to it. Adhering to these restrictions, we use the following function:

$$\sigma(x) = \sigma_b \exp\left( r \left( \frac{1}{k} \left( \sum_{x_i \in \mathcal{N}_k(x)} \|x - x_i\| \right) - m \right) \right), \tag{1}$$

for $\sigma_b$ being a *base standard deviation*, $r$ being the required semi-elasticity, $\{x_i\}_{i=1}^d$ the training set, $\mathcal{N}_k(x)$ the $k$ nearest neighbors of $x$ and $m$ the normalization constant. Intuitively, if a sample is far from all other samples, it will be far from the decision boundary, unless the network overfits to this sample. On the other hand, the dense clusters of samples are more likely to be positioned near the decision boundary, since such clusters have a high leverage on the network's weights, forcing the decision boundary to adapt well to the geometry of the cluster. To use such a function, however, we first prove that it is indeed $r$-semi-elastic.

**Theorem 8.** The $\sigma(x)$ defined in equation 1 is $r$-semi-elastic.

We test our IDRS and our $\sigma(x)$ functions extensively. For both CIFAR10 (Krizhevsky, 2009) and MNIST (LeCun et al., 1999) datasets, we analyze series of different experimental setups, including experiments with an input-dependent train-time Gaussian data augmentation. We present a direct comparison of our method with the constant $\sigma$ method using evaluation strategy from Cohen et al. (2019) (all other experiments, including ablation studies, and the discussion on the hyperparameter selection are presented in Appendix E). Here, we compare Cohen et al. (2019)'s evaluations for $\sigma = 0.12, 0.25, 0.50$ with our evaluations, setting $\sigma_b = \sigma$, $r = 0.01$, $k = 20$, $m = 5, 1.5$ (for CIFAR10 and MNIST, respectively), applied on models trained with Gaussian data augmentation, using constant standard deviation roughly equal to the average test-time $\sigma(x)$ or test-time $\sigma$. For CIFAR10, these levels of train-time standard deviation are $\sigma_{tr} = 0.126, 0.263, 0.53$ and for MNIST $\sigma_{tr} = 0.124, 0.258, 0.517$. In this way, the levels of $\sigma(x)$ we use in the direct comparison are spread from the values roughly equal to Cohen et al. (2019)'s constant $\sigma$ to higher values. The results are depicted in Figure 4.

From Figure 4 we see that we outperform the constant $\sigma$ for small levels of $\sigma$, such as $0.12$ or $0.25$. On higher levels of $\sigma$, we are, in general, worse (see explanation in Appendix B.3). The most visible improvement is in mitigation of the truncation of certified accuracy (certified accuracy waterfall). To comment on the other two issues, we provide Tables 2 and 3 with the clean accuracies and class-wise accuracy standard deviations. These results are averages of 8 independent runs and in Table 2, the displayed error values are equal to empirical standard deviations.

From Tables 2 and 3, we draw two conclusions - first, it is not easy to judge about the robustness vs. accuracy trade-off, because the differences in clean accuracies are not statistically significant in

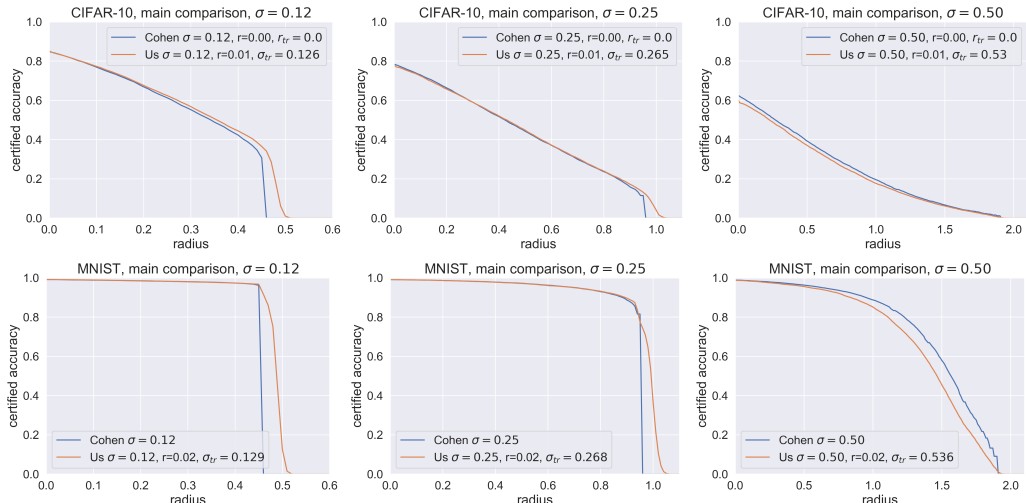

Figure 4: Comparison of certified accuracy plots for Cohen et al. (2019) and our work.

|  | dataset | $\sigma = 0.12$ | $\sigma = 0.25$ | $\sigma = 0.50$ |
|---|---|---|---|---|
| $r = 0.01, \sigma_{tr}$ increased | C | $0.852 \pm 0.002$ | $0.780 \pm 0.013$ | $0.673 \pm 0.008$ |
| $r = 0.00$ | C | $0.851 \pm 0.006$ | $0.792 \pm 0.004$ | $0.674 \pm 0.018$ |
| $r = 0.01, \sigma_{tr}$ increased | M | $0.9912 \pm 0.0003$ | $0.9910 \pm 0.0006$ | $0.9881 \pm 0.0003$ |
| $r = 0.00$ | M | $0.9914 \pm 0.0004$ | $0.9907 \pm 0.0004$ | $0.9886 \pm 0.0005$ |

Table 2: Clean accuracies for both input-dependent and constant $\sigma$ evaluation strategies on CIFAR10 (C) and MNIST (M).

|  | dataset | $\sigma = 0.12$ | $\sigma = 0.25$ | $\sigma = 0.50$ |
|---|---|---|---|---|
| $r = 0.01, \sigma_{tr}$ increased | C | 0.076 | 0.099 | 0.120 |
| $r = 0.00$ | C | 0.076 | 0.097 | 0.122 |
| $r = 0.01, \sigma_{tr}$ increased | M | 0.00775 | 0.00777 | 0.00930 |
| $r = 0.00$ | M | 0.00751 | 0.00778 | 0.00934 |

Table 3: Class-wise accuracy standard deviations for both input-dependent and constant $\sigma$ evaluation strategies on CIFAR10 (C) and MNIST (M).

any of the experiments (not even for CIFAR10 and $\sigma = 0.25$, where the difference is at least pronounced). However, the general trend in Table 2 indicates that the clean accuracies tend to slightly decrease with increasing rate. Nevertheless, the differences are not large enough to compensate negatively the fact that we outperform constant $\sigma$ in terms of the certified accuracies. Second, the standard deviations of the class-wise accuracies, which serve as a good measure of the impact of the shrinking phenomenon and subsequent fairness, don't significantly change after applying the non-constant RS.

## 5  RELATED WORK

Since the vulnerability of deep neural networks against adversarial attacks has been noticed by Szegedy et al. (2013); Biggio et al. (2013) a lot of effort has been put into making neural nets more robust. There are two types of solutions – empirical and certified defenses. While empirical defenses suggest heuristics to make models robust, certified approaches additionally provide a way to compute a mathematically valid robust radius.

One of the most effective empirical defenses, *adversarial training* (Goodfellow et al., 2014; Kurakin et al., 2016; Madry et al., 2017), is based on a very intuitive idea to use adversarial examples for training. Unfortunately, together with adversarial training, other promising empirical defenses were subsequently broken by more sophisticated adversarial methods (for instance Carlini & Wagner (2017); Athalye & Carlini (2018); Athalye et al. (2018), among many others).

Among many certified defenses (Tsuzuku et al., 2018; Anil et al., 2019; Hein & Andriushchenko, 2017; Wong & Kolter, 2018; Raghunathan et al., 2018; Mirman et al., 2018; Weng et al., 2018), one of the most successful yet is RS. While Lecuyer et al. (2019) introduced the method within the context of differential privacy, Li et al. (2019) proceeded via the knowledge of Rényi divergences. Possibly the most prominent work on RS is that of Cohen et al. (2019), where authors fully established RS and proved tight certification guarantees.

Later, a lot of authors further worked with RS. The work of Yang et al. (2020) generalizes the certification provided by Cohen et al. (2019), to certifications with respect to the general $l_p$ norms and provide the optimal smoothing distributions for each of the norms. Other works point out different problems or weaknesses of RS like the curse of dimensionality (Kumar et al., 2020; Hayes, 2020; Wu et al., 2021), robustness vs. accuracy trade-off (Gao et al., 2020) or a shrinking phenomenon(Mohapatra et al., 2020a), which yields serious fairness issues (Mohapatra et al., 2020a).

The work of Mohapatra et al. (2020b) improves RS further by introducing the first-order information about $g$. In this work, authors not only estimate $g(x)$, but also $\nabla g(x)$, making more restrictions on the possible base models $f$ that might have created $g$. Zhai et al. (2020) and Salman et al. (2019) improve the training procedure of $f$ to yield better robustness guarantees of $g$. Salman et al. (2019) directly use adversarial training of the base classifier $f$. Finally, Zhai et al. (2020) introduce so-called *soft smoothing*, which enables to compute gradients directly for $g$ and construct a training method, which optimizes directly for the robustness of $g$ via the gradient descent.

To address several issues connected to randomized smoothing, there have been already four works that introduce the usage of IDRS. Wang et al. (2021) divide $\mathbb{R}^N$ into several regions $R_i, i \in \{1, \ldots, K\}$ and optimize for $\sigma_i, i \in \{1, \ldots, K\}$ locally, such that $\sigma_i$ is a most suitable choice for the region $R_i$. Yet this work partially solves some problems of randomized smoothing, it also possesses some practical and philosophical issues (see Appendix C). Alfarra et al. (2020); Eiras et al. (2021); Chen et al. (2021), suggest to optimize for locally optimal $\sigma_i$, for each sample $x_i$ from the test set. A similar strategy is proposed by these works in the training phase, with the intention of obtaining the base model $f$ that is most suitable for the construction of the smoothed classifier $g$. They demonstrate, that by using this input-dependent approach, one can overcome some of the main problems of randomized smoothing. However, as we demonstrate in Appendix C, their methodology is not valid and therefore their results are not trustworthy.

# 6 Conclusions

We show in this work that input-dependent randomized smoothing suffers from the curse of dimensionality. In the high-dimensional regime, the usage of input-dependent $\sigma(x)$ is being put under strict constraints. The $\sigma(x)$ function is forced to have very small semi-elasticity. This is in conflict with some recent works, which have used the input-dependent randomized smoothing without mathematical justification and therefore claim invalid results. It seems that input-dependent randomized smoothing has limited potential of improvement over the classical, constant-$\sigma$ RS. Moreover, due to numerical instability, the computation of certified radiuses on high-dimensional datasets like ImageNet remains to be an open challenge.

On the other hand, we prepare a ready-to-use mathematically underlined framework for the usage of the input-dependent RS and show, that it works well for small to medium-sized problems. We also show via extensive experiments, that our concrete design of the $\sigma(x)$ function reasonably treats the truncation issue connected to constant-$\sigma$ RS and is partially capable of mitigating the robustness vs. accuracy one. The most intriguing and promising direction for the future work lies in the development of new $\sigma(x)$ functions, which are capable of even better treatment of the mentioned issues.

## 7 REPRODUCIBILITY STATEMENT

Both our theoretical and practical results and experiments are reproducible. In the theoretical part, we provide all the relevant proofs, insights and all the important reasoning. We use public datasets for our experiments and our code contains just public, well-known libraries. The code will be publicly available after the review process. We do not use seeds for stochastic algorithms, but the level of variance is not large enough to obtain qualitatively different results. Upon request, we are willing to provide models trained by us to obtain exactly the same results.

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

## A  THE ISSUES OF CONSTANT $\sigma$ SMOOTHING

### A.1  TOY EXAMPLE

To better demonstrate our ideas, we prepared a two-dimensional simple toy dataset. This dataset can be seen in Figure 5. The dataset is generated in polar coordinates, having uniform angle and the distance distributed as a square root of suitable chi-square distribution. The classes are positioned in a circle sectors, one in a sector with a very sharp angle. The number of training samples is 500 for each class, number of test samples is 100 for each class (except demonstrative figures, where we increased it to 300). The model that was trained on this dataset was a simple fully connected three-layer neural network with ReLU activations and a maximal width of 20.

### A.2  UNDERCERTIFICATION CAUSED BY THE USE OF LOWER CONFIDENCE BOUNDS

As we mention in Section 1, one can not usually obtain exact values of $p_A$ and $p_B$. However, it is obvious, that for vast majority of evaluated samples, $\underline{p_A} < p_A$ and $\overline{p_B} > p_B$. Given the nature of our certified radius, it follows that $\underline{R} < R$, where $\underline{R}$ denotes the certified radius coming from the certification procedure with $\underline{p_A}$ and $\overline{p_B}$, while $R$ here stands for the certified radius corresponding to true values $p_A, p_B$.

It is, therefore, clear, that we face a certain level of under-certification. But how serious undercertification it is? Assume the case with a linear base classifier. Imagine, that we move the point $x$ further and further away from the decision boundary. Therefore, $p_A \to 1$. At some point, the

probability will be so large, that with high probability, all $n$ samplings in our evaluation of $\underline{p_A}$ will be classified as $A$, obtaining $\hat{p}_A = 1$ - the empirical probability. The lower confidence bound $\underline{p_A}$ is therefore bounded by having $\hat{p}_A = 1$. Thus, from some point, the certification will yield the same $\underline{p_A}$ regardless of the true value of $p_A$. So in practice, we have an upper bound on the certified radius $\overline{R}$ in the case of the linear boundary. In Figure 7 (left), we see the truncation effect. Using $\sigma = 1$, from a distance of roughly 4, we can no longer achieve a better certified radius, despite its theoretical value equals the distance. Similarly, if we fix a distance of $x$ from decision boundary and vary $\sigma$, for very small values of $\sigma$, the value of $\Phi^{-1}(p_A)$ will no longer increase, but the values of $\sigma$ will pull $R$ towards zero. This behaviour is depicted in Figure 7 (right).

We can also look at it differently - what is the ratio between $\Phi^{-1}(p_A)$ and $\Phi^{-1}(\underline{p_A})$ for different values of $p_A$? Since $\underline{R} = \sigma\Phi^{-1}(\underline{p_A})$ and $R = \sigma\Phi^{-1}(p_A)$, the ratio represents the "undercertification rate". In Figure 6 we plot $\frac{\Phi^{-1}(p_A)}{\Phi^{-1}(\underline{p_A})}$ as a function of $p_A$ for two different ranges of values. The situation is worst for very small and very big values of $p_A$. In the case of very big values, this can be explained due to extreme nature of $\Phi^{-1}$. For small values of $p_A$, it can be explained as a consequence of a fact, that even small difference between $p_A$ and $\underline{p_A}$ will yield big ratio between $\Phi^{-1}(p_A)$ and $\Phi^{-1}(\underline{p_A})$ due to the fact, that these values are close to 0.

If we look at the left plot on Figure 8 we see, that the certified accuracy plots also possess the truncations. Above some radius, no sample is certified anymore. The problem is obviously more serious for small values of $\sigma$. On the right plot of Figure 8, we see, that samples far from the decision boundary are obviously under-certified. We can also see, that certified radiuses remain constant, even though in reality they would increase with increasing distance from the decision boundary.

All the observations so far motivate us to use rather large values of $\sigma$ in order to avoid the truncation problem. However, as we will see in the next sections, using a large $\sigma$ carries a different, yet equally serious burden.

### A.3 ROBUSTNESS VS. ACCURACY TRADE-OFF

As we demonstrate in the previous subsection, it is be useful to use large values of $\sigma$ to prevent the under-certification. But does it come without a prize? If we have a closer look at Figure 8 (right), we might notice, that the accuracy on the threshold 0, i.e. "clean accuracy", decreases as $\sigma$ increases. This effect has been noticed in the literature (Cohen et al., 2019; Gao et al., 2020; Mohapatra et al., 2020a) and is called *robustness vs. accuracy tradeoff*.

There are several reasons, why this problem occurs. Generally, changing $\sigma$ changes the decision boundary of $g$ and we might assume, that due to the high complexity of the boundary of $f$, the decision boundary of $g$ becomes smoother. If $\sigma$ is too large, however, the decision boundary will be so smooth, that it might lose some amount of the base classifier's expressivity. Another reason for the accuracy drop is also the increase in the number of samples, for which the evaluation is abstained. This is because using big values of $\sigma$ makes more classes "within the reach of our distribution", making the $p_A$ and $\underline{p_A}$ small. If $\underline{p_A} < 0.5$ and we do not estimate $p_B$ but set $\overline{p}_B = 1 - \underline{p_A}$, then

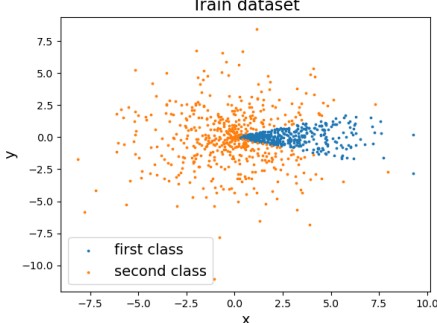
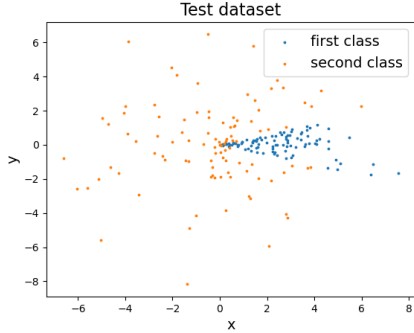

Figure 5: The toy dataset.

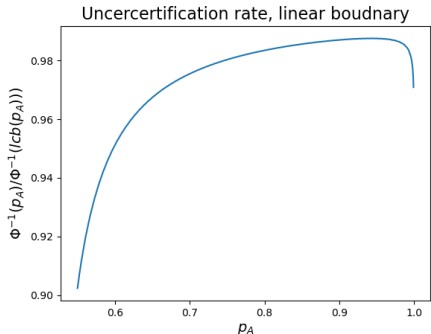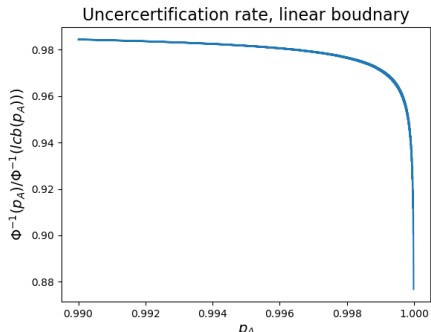

Figure 6: The ratio between certified radius if using lower confidence bounds and if using exact values for the case of linear boundary.

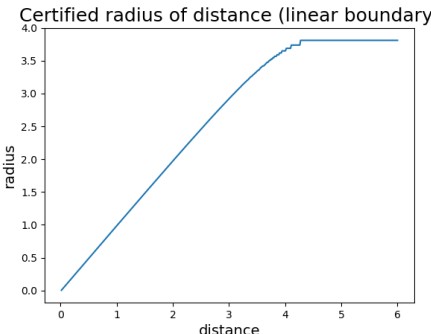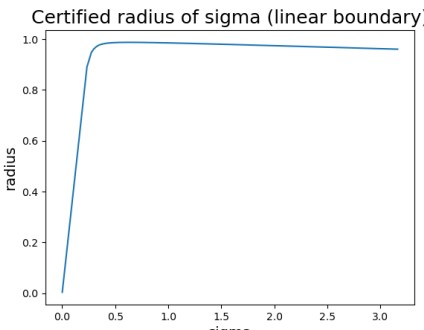

Figure 7: **Left:** Certified radius as a function of distance in linear boundary case. The truncation is due to the use of lower confidence bounds. The parameters are $n = 100000, \alpha = 0.001, \sigma = 1$. **Right:** Certified radius for a point $x$ at fixed distance 1 from linear boundary as a function of used $\sigma$. The undercertification follows from usage of lower confidence bounds.

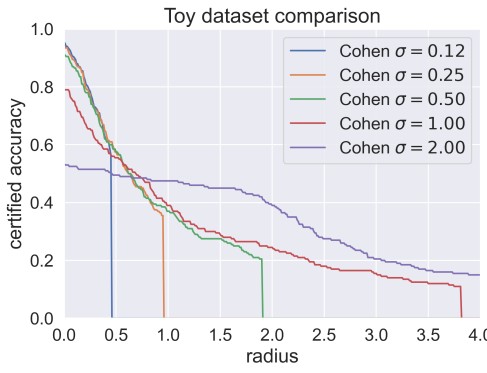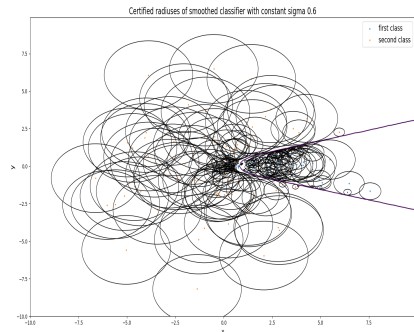

Figure 8: Results of certification on toy dataset. **Left:** Certified accuracy for different levels of $\sigma$. **Right:** Certified radiuses and decision boundary of $g$ visualized directly on test set.

we are not able to classify the sample as class $A$, yet we cannot classify it as a different class either, which forces us to abstain. To demonstrate these results, we computed not only the clean accuracies of Cohen et al. (2019) evaluations but also the abstention rates. Results are depicted in the Table 4.

From the table, it is obvious, that the abstention rate is possibly even bigger cause of accuracy drop than the "clean misclassification". This problem can be partially solved if one estimated $\overline{p_B}$ together

| | Accuracy | Abstention rate | Misclassification rate |
|---|---|---|---|
| $\sigma = 0.12$ | 0.814 | 0.038 | 0.148 |
| $\sigma = 0.25$ | 0.748 | 0.086 | 0.166 |
| $\sigma = 0.50$ | 0.652 | 0.166 | 0.182 |
| $\sigma = 1.00$ | 0.472 | 0.29 | 0.238 |

Table 4: Accuracies, rates of abstentions and misclassification rates of Cohen et al. (2019) evaluation for different levels of $\sigma$.

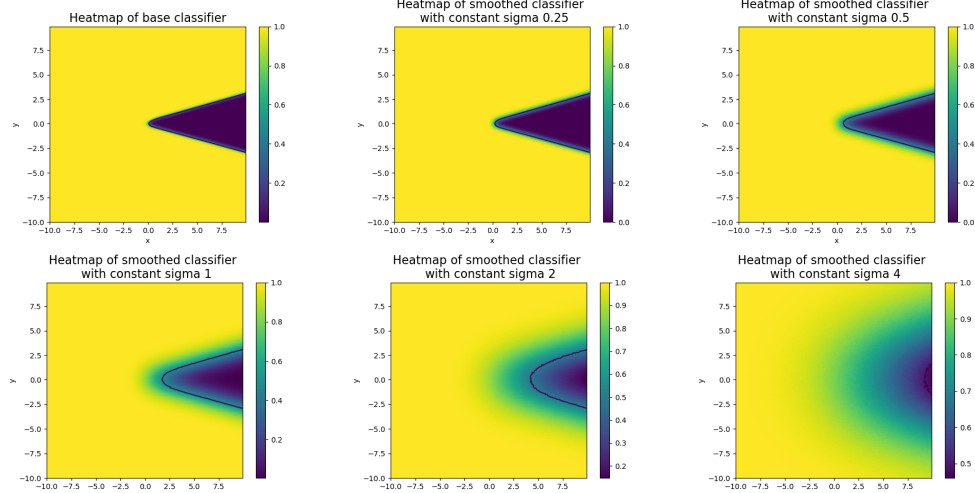

Figure 9: Heatmaps and decision boudnary of base classifier (top left) and the smoothed classifier for increasing levels of $\sigma$. As $\sigma$ increases, the classifier is more smooth and the decision boundary recedes.

with $p_A$ too. In this way, using big $\sigma$ yields generally small estimated class probabilities, but since $p_A \geq p_B$, the problematic $\overline{p_B} \geq p_A$ occur just very rarely. Another option is to increase the number of Monte-Carlo samplings for the classification decision, what is almost for free.

Yet another reason for the decrease in the accuracy is the so-called *shrinking phenomenon*, which we will discuss in the next subsection.

In contrast with the truncation effect, the robustness vs. accuracy trade-off motivates the usage of smaller values of $\sigma$ in order to prevent the accuracy loss, which is definitely a very serious issue.

## A.4 SHRINKING PHENOMENON

How exactly does the decision boundary of $g$ change, as we change the $\sigma$? For instance, if $f$ is a linear classifier, then the boundary does not change at all. To answer this question, we employ the following experiment: For our toy base classifier $f$ on our toy dataset, we increase $\sigma$ and plot the heatmap of $f, g$, together with its decision boundary. This experiment is depicted on Figure 9. As we see from the plots, increasing $\sigma$ causes several effects. First of all, the heatmap becomes more and more blurred, what proves, that stronger smoothing implies stronger smoothness.

Second, crucial, effect is that the bigger the $\sigma$, the smaller the decision boundary of a submissive class is. The shrinkage becomes pronounced from $\sigma = 1$. Already for $\sigma = 4$, there is hardly any decision boundary anymore. Generally, as $\sigma \to \infty$, $g$ will predict the class with the biggest volume in the input space (in the case of bounded input space, like in image domain, this is very well defined). For extreme values of sigma, the $p_A$ will practically just be the ratio between the volume of $A$ and the actual volume of the input space (for bounded input spaces).

Following from these results, but also from basic intuition, it seems, that an undesired effect becomes present as $\sigma$ increases - the bounded/convex regions become to shrink, like in Figure 9, while the unbounded/big/anti-convex regions expand. This is called *shrinking phenomenon*. Mohapatra et al. (2020a) investigate this effect rather closely. They define the shrinkage and vanishing of regions formally and prove rigorously, that if $\sigma \to \infty$, bounded regions, or semi-bounded regions (see Mohapatra et al. (2020a)) will eventually vanish. We formulate the main result in this direction.

**Theorem 9.** Let us have $K$ the number of classes and the dimension $N$. Assume, that we have some bounded decision region $\mathcal{D}$ for a specific class roughly centered around 0. Further assume, that this is the only region where the class is classified. Let $R$ be a smallest radius such that $\mathcal{D} \subset B_R(0)$. Then, this decision region will vanish at most for $\sigma \geq \frac{R\sqrt{K}}{\sqrt{N}}$.

*Proof.* The idea of the proof is not very hard. First, the authors prove, that the smoothed region will be a subset of the smoothed $B_R(0)$. Then, they upper-bound the vanishing threshold of such a ball in two steps. First, they show, that if 0 is not classified as the class, then no other point will be (this is quite an intuitive geometrical statement. The $B_R(0)$ has the biggest probability under $\mathcal{N}(x, \sigma^2 I)$ if $x \equiv 0$). Second, they upper-bound the threshold for $\sigma$, under which $B_R(0)$ will have probability below $\frac{1}{K}$ (since they use slightly different setting as Cohen et al. (2019)) for the $\mathcal{N}(x, \sigma^2 I)$. Using some insights about incomplete gamma function, which is known to be also the cdf of central chi-square distribution, and some other integration tricks, they obtain the resulting bound. □

Besides Theorem 9, authors also claim many other statements abound shrinking, including shrinking of semi-bounded regions. Moreover, they also conduct experiments on CIFAR10 and ImageNet to support their theoretical findings. They also point out serious fairness issue that comes out as a consequence of the shrinkage phenomenon. For increasing levels of $\sigma$, they measure the class-wise clean accuracy of the smoothed classifier. If $f$ is trained with gaussian data augmentation (what is known to be a good practice in randomized smoothing), using $\sigma = 0.12$, the worst class *cat* has test accuracy of 67%, while the best class *automobile* attains the accuracy of 92%. The figures, however, change drastically, if we use $\sigma = 1$ instead. In this case, the worst predicted class *cat* has accuracy of poor 22%, while *ship* has reasonable accuracy 68%. As authors claim, this is a consequence of the fact that samples of *cat* are situated more in bounded, convex regions, that suffer from shrinking, while samples of *ship* are mostly placed in expanded regions of anti-convex shape that will expand as the $\sigma$ grows. In addition, the authors also show, that the gaussian data augmentation or adversarial training will reduce the shrinking phenomenon just partially and for moderate and high values of $\sigma$, this effect will be present anyway.

We must emphasize, that this is a serious fairness issue, that has to be treated before randomized smoothing can be fully used in practice. For instance, if we trained a neural network to classify humans into several categories, fairness of classification is inevitable and the neural network cannot be used until this issue is solved.

Similarly as the robustness vs. accuracy tradeoff, this issue also motivates to use rather smaller values of $\sigma$. We see, that it is not possible to address all three problems consistently because they disagree on whether to use smaller, or bigger values of $\sigma$ .

### A.5 EXPERIMENTS ON HIGH-DIMENSIONAL TOY DATASET

In this subsection, we present the results of our motivational experiment on a synthetic dataset. Before reading this section, please read our main text, because we will use the necessary notation of the paper.

The dataset we evaluated our method on is a generalization of the dataset visualized on Figure 1. The data points from one class lie in a cone of small angle and the points are generated such that the density is higher near the vertex of the cone (which is put in origin). Points from other class are generated from a spherically symmetrical distribution (where points sampled into the cone are excluded) with density again highest in the center (note, that the density peak is more pronounced than in the case of normal distribution, where the density around the center resembles uniform distribution). This dataset is chosen so that the $\sigma(x)$ function designed in Equation 1 well corresponds to the geometry of the decision boundary. Moreover it is chosen so that the conic decision region

| Dimension | $\sigma$ ($\sigma_b$) | $r$ | Accuracy |
|-----------|-----------|-------|----------|
| 2 | 0.5 | - | 0.943 |
| 2 | 0.4 | 0.2 | 0.96 |
| 2 | 0.5 | 0.2 | 0.943 |
| 6 | 0.5 | - | 0.946 |
| 6 | 0.4 | 0.1 | 0.963 |
| 18 | 1.0 | - | 0.86 |
| 18 | 0.8 | 0.05 | 0.886 |
| 60 | 1.0 | - | 0.83 |
| 60 | 0.8 | 0.03 | 0.85 |
| 60 | 1.0 | 0.03 | 0.83 |
| 180 | 2.0 | - | 0.713 |
| 180 | 1.9 | 0.01 | 0.726 |
| 400 | 2.0 | - | 0.623 |
| 400 | 1.95 | 0.005 | 0.623 |

Table 5: Clean accuracies of different evaluations of our toy experiment.

will shrink rather fast with increasing $\sigma$. The motivation of this example is to show that if the $\sigma(x)$ function is well-designed, our IDRS can outperform the constant RS considerably.

The setup of our experiment is as follows: We evaluate dimensions $N = 2, 6, 18, 60, 180, 400$. The $\sigma$ used for constant smoothing is $\sigma = 0.5, 0.5, 1.0, 1.0, 2.0, 2.0$ respectively. The $\sigma_b$ used is $0.4, 0.5$ for $N = 2$, $0.4$ for $N = 6$, $0.8$ for $N = 18$, $0.8, 1.0$ for $N = 60$, $1.9$ for $N = 180$ and $1.95$ for $N = 400$. The rates are $r = 0.2, 0.1, 0.05, 0.03, 0.01, 0.005$ respectively. The training was executed without data augmentation (because samples from different classes are very close to each other). Moreover, we have set maximal $\sigma(x)$ threshold for numerical purposes, because some samples were outliers and were way too far from other samples (and if the $\sigma(x)$ is way too big, the method encounteres numerical problems). In this case we set $\sigma(x) \leq 5\sigma_b$, but we are aware that also much bigger thresholds would have been possible. We present our comparisons in Figure 10 and Table 5.

From both the Figure 10 an Table 5 it is clear that the IDRS can outperform the constant $\sigma$ RS considerably, if we use really suitable $\sigma(x)$ function. We manage to improve significantly the certified radiuses without losing a single correct classification. On the other hand, in cases where $\sigma_b < \sigma$, we outperform constant $\sigma$ both in clean accuracy and in certified radiuses. This example is synthetic and designed in our favour. The main message is not how perfect our design of $\sigma(x)$ is, but the fact, that if $\sigma(x)$ is designed well, the IDRS can bring real advantages, even in moderate dimensions.

## B MORE ON THEORY

### B.1 GENERALIZATION OF RESULTS FROM LI ET AL. (2019)

In our main text, we mostly focus on the generalization of the methods from Cohen et al. (2019). This is because these methods yield tight radiuses and because the application of Neyman-Pearson lemma is beautiful. However, the methodology from Li et al. (2019) can also be generalized for the input-dependent RS. To be able to do it, we need some auxiliary statements about the Rényi divergence.

**Lemma 10.** The Rényi divergence between two one-dimensional normal distributions is as follows:

$$D_\alpha(\mathcal{N}(\mu_1, \sigma_1^2)||\mathcal{N}(\mu_0, \sigma_0^2)) = \frac{\alpha(\mu_1 - \mu_2)^2}{2\sigma_\alpha^2} + \frac{1}{1-\alpha} \log\left(\frac{\sigma_\alpha}{\sigma_1^{1-\alpha}\sigma_0^\alpha}\right),$$

provided, that $\sigma_\alpha^2 := (1 - \alpha)\sigma_1^2 + \alpha\sigma_0^2 \geq 0$.

*Proof.* See Van Erven & Harremos (2014). □

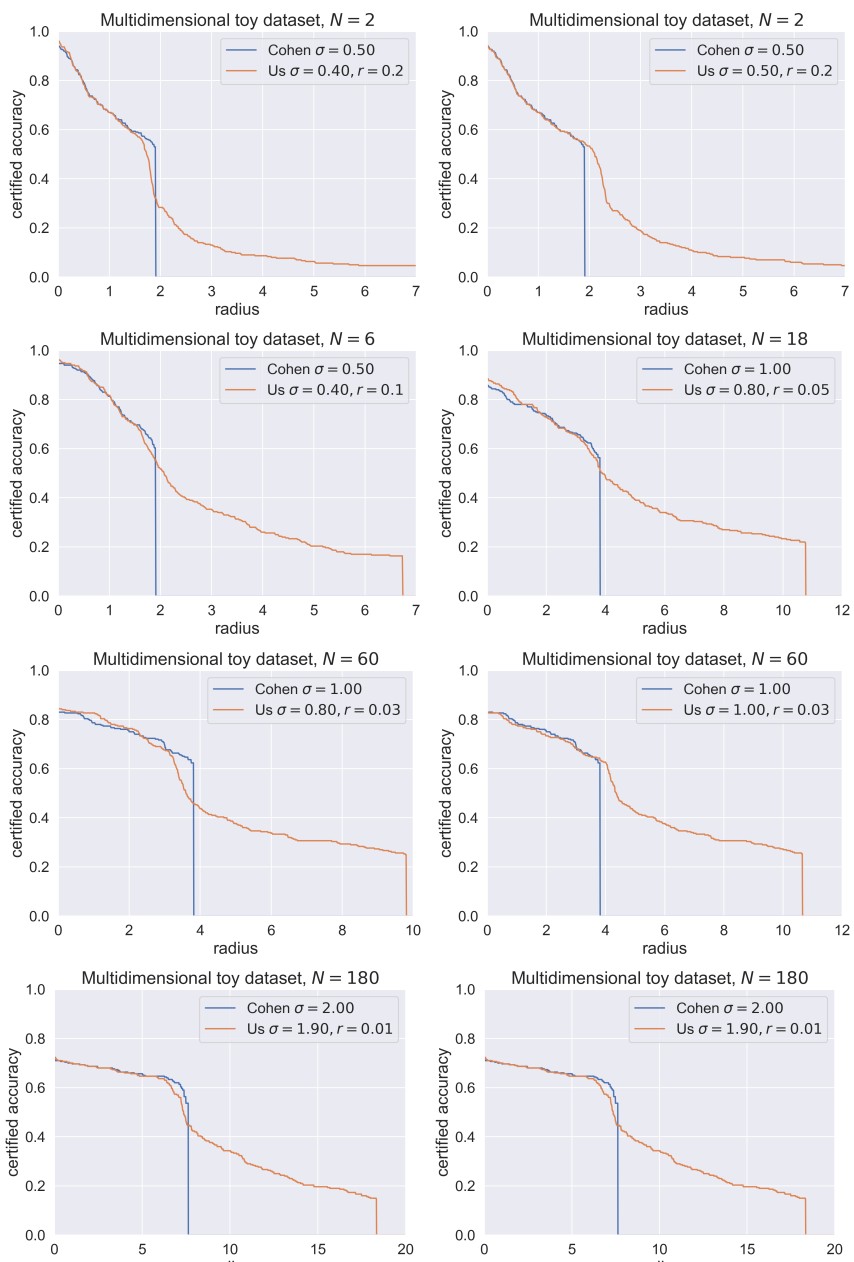

Figure 10: Certified accuracy plots of our multidimensional toy experiments.

Note, that this proposition induces some assumptions on how $\sigma_0, \sigma_1, \alpha$ should be related. If $\sigma_0 > \sigma_1$, then the required inequality holds for any $1 \neq \alpha > 0$. If $\sigma_0 < \sigma_1$, then $\alpha$ is restricted and we need to keep that in mind.

**Lemma 11.** Assume, we have some one-dimensional distributions $\mathcal{P}_1, \mathcal{P}_1, \ldots, \mathcal{P}_N$ and $\mathcal{Q}_1, \mathcal{Q}_2, \ldots, \mathcal{Q}_N$ defined on common space for pairs with the same index. Then, assuming product space with product $\sigma$-algebra, we have the following identity:

$$D_\alpha(\mathcal{P}_1 \times \mathcal{P}_2 \times \cdots \times \mathcal{P}_N || \mathcal{Q}_1 \times \mathcal{Q}_2 \times \cdots \times \mathcal{Q}_N) = \sum_{i=1}^{N} D_\alpha(\mathcal{P}_i || \mathcal{Q}_i).$$

*Proof.* See Van Erven & Harremos (2014). □

Using these two propositions, we are now able to derive a formula for Rényi divergence between two multivariate isotropic normal distributions:

**Lemma 12.**

$$D_\alpha(\mathcal{N}(x_1, \sigma_1^2 I)||\mathcal{N}(x_0, \sigma_0^2 I)) = \frac{\alpha\|x_0 - x_1\|^2}{2\sigma_1^2 + 2\alpha(\sigma_0^2 - \sigma_1^1)} + N\frac{\log\left(\frac{\sigma_\alpha}{\sigma_1}\right)}{1-\alpha} - N\frac{\alpha}{1-\alpha}\log\left(\frac{\sigma_0}{\sigma_1}\right).$$

*Proof.* Imporant property that is needed here is, that isotropic gaussian distributions factorize to one-dimensinal independent marignals. In other words:

$$\mathcal{N}(x_1, \sigma_1^2 I) = \mathcal{N}(x_{11}, \sigma_1^2) \times \mathcal{N}(x_{12}, \sigma_1^2) \times \cdots \times \mathcal{N}(x_{1N}, \sigma_1^2),$$

and analogically for $x_0$. Therefore, using Lemma 11 we see:

$$D_\alpha(\mathcal{N}(x_1, \sigma_1^2 I)||\mathcal{N}(x_0, \sigma_0^2 I)) = \sum_{i=1}^N D_\alpha(\mathcal{N}(x_{1i}, \sigma_1^2)||\mathcal{N}(x_{0i}, \sigma_0^2)).$$

Now, it suffices to plug in the formula from Proposition 10 to obtain the required result:

$$D_\alpha(\mathcal{N}(x_{1i}, \sigma_1^2 I)||\mathcal{N}(x_{0i}, \sigma_0^2 I)) = \frac{\alpha(x_{1i} - x_{2i})^2}{2\sigma_\alpha^2} + \frac{1}{1-\alpha}\log\left(\frac{\sigma_\alpha}{\sigma_1^{1-\alpha}\sigma_0^\alpha}\right)$$

$$= \frac{\alpha(x_{1i} - x_{2i})^2}{2\sigma_1^2 + 2\alpha(\sigma_0^2 - \sigma_1^1)} + \frac{\log\left(\frac{\sigma_\alpha}{\sigma_1}\right)}{1-\alpha} - \frac{\alpha}{1-\alpha}\log\left(\frac{\sigma_0}{\sigma_1}\right)$$

Now it suffices to sum up over $i$ and the result follows. □

To obtain the certified radius, we also need a result from Li et al. (2019), which gives a guarantee that two measures on the set of classes will share the modus if the Rnyi divergence between them is small enough.

**Lemma 13.** Let $\mathbb{P} = (p_1, p_2, \ldots, p_K)$ and $\mathbb{Q} = (q_1, q_2, \ldots, q_K)$ two discrete measures on $\mathcal{C}$. Let $p_A, p_B$ correspond to two biggest probabilities in distribution $\mathbb{P}$. Let $M_1(a,b) = \frac{a+b}{2}$ and $M_{1-\alpha}(a,b) = (\frac{a^{1-\alpha}+b^{1-\alpha}}{2})^{\frac{1}{1-\alpha}}$ If

$$D_\alpha(\mathbb{Q}||\mathbb{P}) \leq -\log(1 - 2M_1(p_A, p_B) + 2M_{1-\alpha}(p_A, p_B)),$$

then the distributions $\mathbb{P}$ and $\mathbb{Q}$ agree on the class with maximal assigned probability.

*Proof.* This lemma can be proved by directly computing the minimal required $D_\alpha$ to be able to disagree on the maximal class probabilities via a constrained optimization problem (with variables $p_i, q_i, i \in \{1, \ldots, K\}$), solving KKT conditions. For details, consult Li et al. (2019). □

Having explicit formula for the Rnyi divergence, we can mimic the methodology of Li et al. (2019) to obtain the certified radius:

**Theorem 14.** Given $x_0, p_A, p_B, \sigma_0, N$, the certified radius squared for all $x_1$ such that fixed $\sigma_1$ is used is:

$$R^2 = \sup_{\alpha \in S_{\sigma_0, \sigma_1}} \frac{2\sigma_1^2 + 2\alpha(\sigma_0^2 - \sigma_1^1)}{\alpha}\left(N\frac{\alpha}{1-\alpha}\log\left(\frac{\sigma_0}{\sigma_1}\right) - N\frac{\log\left(\frac{\sigma_\alpha}{\sigma_1}\right)}{1-\alpha}\right.$$

$$\left. - \log(1 - 2M_1(p_A, p_B) + 2M_{1-\alpha}(p_A, p_B))\right),$$

where $S_{\sigma_0, \sigma_1} = \mathbb{R}_+$, if $\sigma_0 > \sigma_1$ and $S_{\sigma_0, \sigma_1} = \left(0, \frac{\sigma_1^2}{\sigma_1^2 - \sigma_0^2}\right]$ if $\sigma_0 < \sigma_1$.

*Proof.* Let us fix $x_1$ and assume, that $\alpha \in S_{\sigma_0,\sigma_1}$. Then, due to post-processing inequality for Renyi divergence, it follows that

$$D_\alpha(f(x_1 + \mathcal{N}(0, \sigma_1^2 I))\|f(x_0 + \mathcal{N}(0, \sigma_0^2 I))) \leq D_\alpha(x_1 + \mathcal{N}(0, \sigma_1^2 I)\|x_0 + \mathcal{N}(0, \sigma_0^2 I))$$

$$= \frac{\alpha\|x_0 - x_1\|^2}{2\sigma_1^2 + 2\alpha(\sigma_0^2 - \sigma_1^1)} + N\frac{\log\left(\frac{\sigma_\alpha}{\sigma_1}\right)}{1 - \alpha} - N\frac{\alpha}{1 - \alpha}\log\left(\frac{\sigma_0}{\sigma_1}\right).$$

Due to Lemma 13, it suffices that the following inequality holds for *some* $\alpha \in S_{\sigma_0,\sigma_1}$:

$$\frac{\alpha\|x_0 - x_1\|^2}{2\sigma_1^2 + 2\alpha(\sigma_0^2 - \sigma_1^1)} + N\frac{\log\left(\frac{\sigma_\alpha}{\sigma_1}\right)}{1 - \alpha} - N\frac{\alpha}{1 - \alpha}\log\left(\frac{\sigma_0}{\sigma_1}\right) \leq$$
$$- \log(1 - 2M_1(p_A, p_B) + 2M_{1-\alpha}(p_A, p_B)).$$

This can be rewritten w.r.t. $\|x_0 - x_1\|^2$:

$$\|x_0 - x_1\|^2 \geq \frac{2\sigma_1^2 + 2\alpha(\sigma_0^2 - \sigma_1^1)}{\alpha}\left(N\frac{\alpha}{1 - \alpha}\log\left(\frac{\sigma_0}{\sigma_1}\right) - N\frac{\log\left(\frac{\sigma_\alpha}{\sigma_1}\right)}{1 - \alpha}\right.$$
$$\left. - \log(1 - 2M_1(p_A, p_B) + 2M_{1-\alpha}(p_A, p_B))\right).$$

The resulting certified radius squared is now simply obtained by taking maximum over $\|x_0 - x_1\|^2$ s.t. $\exists \alpha \in S_{\sigma_0,\sigma_1}$ such that the preceding inequality holds. $\qquad\square$

Note, that this theorem is formulated assuming, that except in $x_0$, we use $\sigma_1$ everywhere. It would require some further work to generalize this for general $\sigma(x)$ functions, but to demonstrate the next point, it is not even necessary. Looking at the expression, we can observe that

$$N\frac{\alpha}{1 - \alpha}\log\left(\frac{\sigma_0}{\sigma_1}\right) - N\frac{\log\left(\frac{\sigma_\alpha}{\sigma_1}\right)}{1 - \alpha}$$

depends highly on $N$ and even for a ratio of $\frac{\sigma_0}{\sigma_1}$ close to 1, we already obtain very strong negative values for high dimensions. The expression $\log(1 - 2M_1(p_A, p_B) + 2M_{1-\alpha}(p_A, p_B))$ is far less sensitive w.r.t $p_A$ and for large dimensions of $N$ it is easily "beaten" by the first expression. Therefore, the higher the dimension $N$ is, the bigger $p_A$ or the closer to 1 the $\frac{\sigma_0}{\sigma_1}$ has to be in order to obtain even valid certified radius (not to speak about big). This points out that also the method of Li et al. (2019) suffers from the curse of dimensionality, as we know it must have done. This method is not useful for big $N$, because the conditions on $p_A, \sigma_0, \sigma_1$ are so extreme, that barely any inputs would yield a positive certified radius. This fact is depicted in the Figure 11.

The key reason why this happens if done via Rényi divergences is that while the divergence $D_\alpha(\mathcal{N}(x_1, \sigma_1^2 I)\|\mathcal{N}(x_0, \sigma_0^2 I))$ grows independently of dimension as $\|x_0 - x_1\|$ grows, it drastically increases for big $N$ even if $x_1 = x_0$! This reflects the effect, that if $\sigma_0 \neq \sigma_1$, then the more dimensions we have, the more dissimilar are $\mathcal{N}(x_1, \sigma_1^2 I)$ and $\mathcal{N}(x_0, \sigma_0^2 I)$. We can think of it as a consequence of standard fact from statistics that the more data we have, the more confident statistics against the null hypothesis $\sigma_0 = \sigma_1$ will we get if the null hypothesis is false. Since isotropic normal distributions can be actually treated as a sample of one-dimensional normal distributions, this is in accordance with our multivariate distributions setting.

## B.2 THE EXPLANATION OF THE CURSE OF DIMENSIONALITY

In the Section 2 we show that input-dependent RS suffers from the curse of dimensionality. Now we will elaborate a bit more on this phenomenon and try to explain why it occurs. First, it is obvious from the Subsection B.1, that also the generalized method of Li et al. (2019) suffers from the curse of dimensionality, because the Rényi divergence between two isotropic Gaussians with different variances grows considerably with respect to dimension. This suggests that the input-dependent RS might suffer from the curse of dimensionality in general. To motivate this idea even further, we present this easy observation:

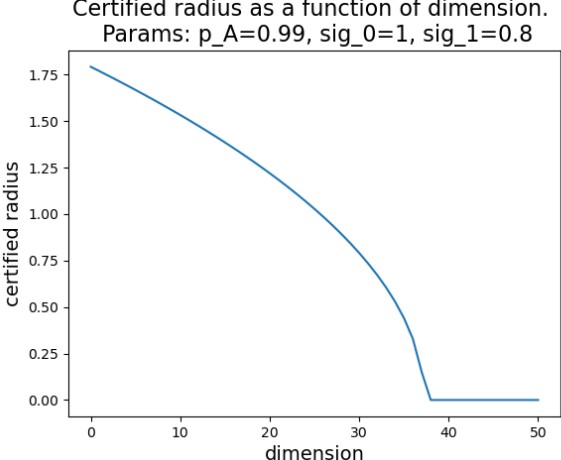

Figure 11: The certified radius as a function of dimension. Paremeters are $p_A = 0.99, \sigma_0 = 1, \sigma_1 = 0.8$

**Theorem 15.** Denote $R_C$ to be a certified radius given for $p_A$ and $\sigma_0$ at $x_0$ assuming the constant $\sigma_0$ and following the certification of Cohen et al. (2019) [1]. Assume, that we do the certification for each $x_1$ by assuming the worst case-classifier as in Theorem 2. Then, for any $x_0$, any function $\sigma(x)$ and any $p_A$, the following inequality holds:

$$R \leq R_C$$

*Proof.* Fix $x_1$ and $\sigma_1$. From Theorem 2 we know that the worst-case classifier $f^*$ defines a ball $B$ such that $\mathbb{P}_0(B) = 1 - p_A$. From this it obviously follows, that the linear classifier $f_l$ and the linear space $B_l$ that assume constant $\sigma_0$ also for $x_1$ and is the worst-case for $\sigma_0$ such that $\mathbb{P}_0(B_l) = 1 - p_A$ is *not* worst-case for the case of using $\sigma_1$ instead. Therefore, $\mathbb{P}_1(B_l) \leq \mathbb{P}_1(B)$.

Moreover, let $\mathbb{P}_1^C$ be a probability measure corresponding to $\mathcal{N}(x_1, \sigma_0 I)$, i.e. the probability measure assuming constant $\sigma_0$. It is easy to see that $\mathbb{P}_1^C(B_l) > 0.5 \iff \mathbb{P}_1(B_l) > 0.5$ because the probability of a linear half-space under isotropic normal distribution is bigger than half if and only if the mean is contained in the half-space.

Assume, for contradiction that $R > R_C$. From that, it exists a particular $x_1$ such that $\mathbb{P}_1^C(B_l) > 0.5 > \mathbb{P}_1(B)$, because otherwise there would be no such point, which would cause $R > R_C$. However, $\mathbb{P}_1^C(B_l) > 0.5 \implies \mathbb{P}_1(B_l) > 0.5$, thus $\mathbb{P}_1(B_l) > \mathbb{P}_1(B)$ and that is contradiction. $\square$

This theorem shows, that we can never achieve a better certified radius at $x_0$ using $\sigma_0$ and having probability $p_A$ than that, which we would get by Cohen et al. (2019)'s certification. Of course, this does not mean, that using non-constant $\sigma$ is useless, since $\sigma_0$ can vary. The question is, how much do we lose using non-constant $\sigma$. To get a better intuition, we plot the functions $\xi_<$ and $\xi_>$ under different setups in Figure 12, together with $\mathbb{P}_1(B_l)$ from the proof of Theorem 15. From the top row we can deduce that dimension $N$ has a very significant impact on the probabilities and therefore also on the certified radius. We particularly point out the fact, that even $\xi_>(0), \xi_<(0)$ can have significant margin w.r.t. to the probability coming out of linear classifier.[2] Already for $N = 90$, we are not able to certify $p_A = 0.99$ for rather conservative value of $\frac{\sigma_0}{\sigma_1}$. From middle row we see, that decreasing $\frac{\sigma_0}{\sigma_1}$ can mitigate this effect strongly. For instance, for $\sigma_0 = 1, \sigma_1 = 0.95$ the difference between $\mathbb{P}_1(B)$ and $\mathbb{P}_1(B_l)$ is almost negotiated. Bottom row compares $\xi_>(a), \xi_<(a)$ and the respective linear classifier probabilities. We can see, that the case $\sigma_0 < \sigma_1$ might cause stronger restrictions on our certification (yet we deduce it just form the picture).

---

[1] The "C" in the subscript of certified radius might come both from "constant" and "Cohen et. al."

[2] Notice the similarity with Rényi divergence, which also has positive value even for $x_0 = x_1$ if $\sigma_0 \neq \sigma_1$ and then grows rather reasonably with distance.

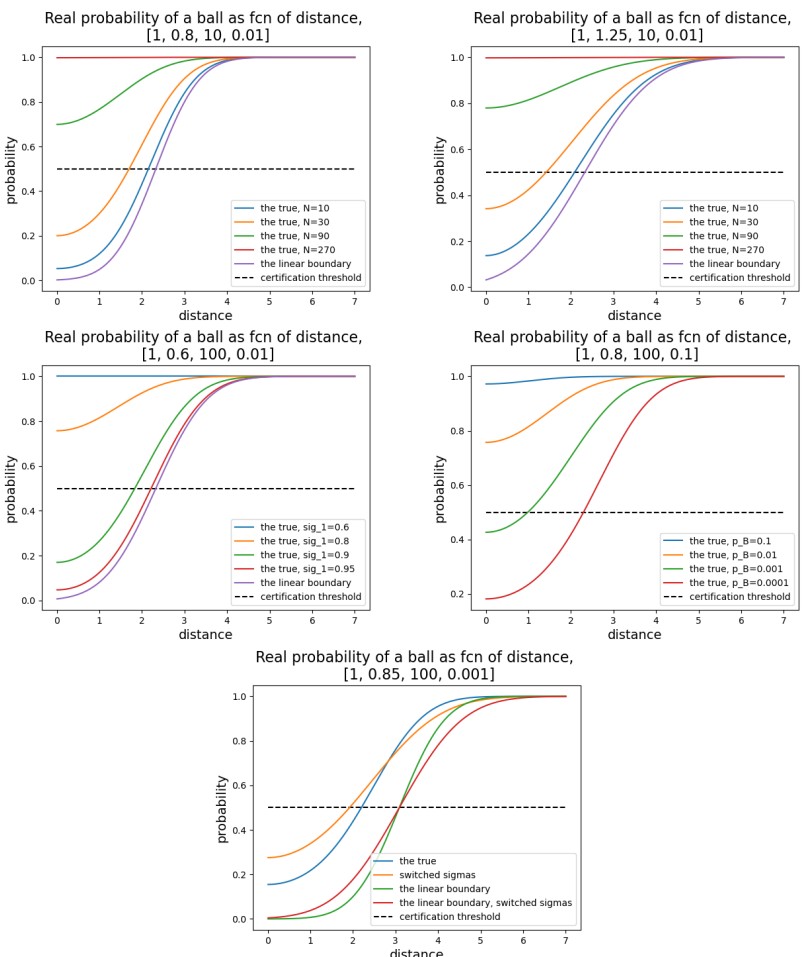

Figure 12: Plots of $\xi_>(a), \xi_<(a)$ for different setups. Coding for parameters is: $[\sigma_0, \sigma_1, N, p_B]$ **Top:** $\xi_>(a)$ left, $\xi_<(a)$ right, varying values of $N$. **Center:** On the left, $\xi_>(a)$ for varying $\sigma_1$, on the right $\xi_>(a)$ for varying $p_B$. **Bottom:** $\xi_>(a)$ and $\xi_<(a)$ compared.

What is the reason for $\xi_>(a), \xi_<(a)$ being so big even at 0? The problem is following: Assume $\sigma_0 > \sigma_1$. If $x_0 = x_1$, the worst-case classifier coming from Lemma 2 will be a ball $B$ centered right at $x_0$, such that $\mathbb{P}_0(B) = 1 - p_A$. If we look at $\mathbb{P}_1(B)$, we see, that we have the same ball centered directly at the mean, but the variance of the distribution is smaller. Using spherical symmetry of the isotropic gaussian distribution, this is equivalent to evaluating the probability of a bigger ball. If we fix $\frac{\sigma_0}{\sigma_1}$ and look at the ratio of probabilities $\frac{\mathbb{P}_1(B)}{\mathbb{P}_0(B)}$ with increasing $N$, the curse of dimensionality comes into the game. For $N = 2$, the ratio is not too big. However, if $N = 3072$, like in CIFAR10, this ratio is far bigger. This can be intuitively seen from a property of chi-square distribution (which is present in the case $x_0 = x_1$), that while expectation is $N$, the standard deviation is "just" $\sqrt{2N}$, i.e. $\frac{\sqrt{Var(\chi_N^2)}}{\mathbb{E}(\chi_N^2)} \to 0$ as $N \to \infty$.

### B.3 WHY DOES THE INPUT-DEPENDENT SMOOTHING WORK BETTER FOR SMALL $\sigma$ VALUES?

As can be observed in Section 4 and Appendix E, the bigger the $\sigma_b = \sigma$ we use, the harder it is to keep up to standards of constant smoothing. An interesting question is, why is the usage of small $\sigma_b = \sigma$ helpful for the input-dependent smoothing?

Assume fixed $\sigma$, say $\sigma_b = \sigma = 0.12$. The theoretical bound on the certified radius given 100000 Monte-Carlo samplings and 0.001 confidence level using constant smoothing is about 0.48. Having

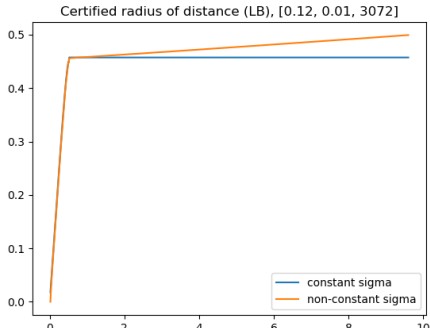 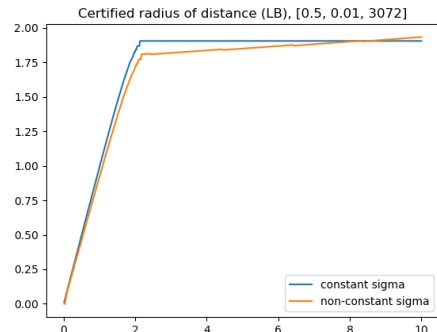

Figure 13: Comparison of certified radius as a function of distance for constant and input-dependent smoothing. Left: $\sigma_b = \sigma = 0.12$, right: $\sigma_b = \sigma = 0.50$.

$\sigma(x) \sim 0.12$, we cannot expect much bigger certified radius. Therefore, if we follow Theorem 7, the values of $\exp(-rR)$ and $\exp(rR)$ in the critical distance $\sim 0.5$ will be much closer to 1, than the values of $\exp(-rR)$ and $\exp(rR)$ if we used $\sigma_b = \sigma = 0.50$ instead, where the critical values of $R$ could be much bigger than 0.5. Therefore, the "gain" in $\mathbb{P}_1(B)$ imposed by the curse of dimensionality, compared to $\mathbb{P}_1(B)$ assuming constant $\sigma$ will not be that severe yet. This means, that the loss in certified radius caused by the curse of dimensionality will be much less pronounced on the "active" range of certified radiuses (those for which the constant smoothing still works), compared to using big $\sigma_b = \sigma$. To support this idea, we demonstrate it on Figure 13, where we depict the certified radius as a function of distance from decision boundary, assuming $f$ to be a linear classifier, using $\sigma_b = \sigma = 0.12$ and $\sigma_b = \sigma = 0.50$ for comparison.

### B.4 How does the curse of dimensionality affect the total possible variability of $\sigma(x)$?

Fix certain type of task, say RGB image classification with images of similar object, but consider many possible resolutions (dimensions $N$). Given two random images from the test set $(x_0, x_1)$, what is the biggest reasonable value of $|\sigma(x_0)/\sigma(x_1) - 1|$? Theoretically, the expression is bounded by $|\exp(\pm r\|x_0 - x_1\|) - 1|$, given that $r$ is the semi-elasticity constant of $\sigma(x)$. However, the average distance between two samples from a test set of constant size, but increasing dimension scales as $\sqrt{N}$. Therefore, with constant $r$, this upper-bound increases.

The increasing distance between samples is, therefore, a countereffect to the curse of dimensionality. In simple words, we have "more distance to change $\sigma(x_0)$ to $\sigma(x_1)$". Even if the $r$ decreased just as $1/\sqrt{N}$, the increasing distances would cancel the effect of the curse of dimensionaity and as a result, the maximal reasonable value of $|\sigma(x_0)/\sigma(x_1) - 1|$ would remain roughly constant w.r.t. $N$. However, we need to take into account another effect. As the dimension increases, also the average distance of samples from the decision boundary increases. This is because the distances in general grow with dimension and if we assume that the number of intersections of a line segment between $x_0$ and $x_1$ with the decision boundary of the network remains roughly constant then the average distance from the decision boundary grows as $\sqrt{N}$ too. In order to compensate for this, we need to adjust the basic level of $\sigma(x)$ (which we later call $\sigma_b$ and can be understood as the general offset of our $\sigma(x)$) as $\sqrt{N}$ too. This is because the maximal attainable certified radius given fixed confidence level $\alpha$ and the number of Monte-Carlo samples is a constant multiple of $\sigma(x)$.

However, with increased $\sigma$, we need to decrease the semi-elasticity rate $r$ in order to obtain full certifications (see also Appendix B.3 for intuition behind this).

As a sketch of proof, we provide a simple computation, which tells us the approximate asymptotic behavior of $|\sigma(x_0)/\sigma(x_1) - 1|$. By Theorem 5 it holds:

$$\exp(-rc\sqrt{N}) \geq \sqrt{1 - 2\sqrt{\frac{-\log(p_B)}{N}}},$$

if we want to be able to predict a certified radius of $c\sqrt{N}$ (though this is just a necessary condition. For sufficiency, the LHS must be much closer to 1). After simple manipulation, we obtain:

$$r \leq -\frac{1}{2c\sqrt{N}}\log\left(1 - 2\sqrt{\frac{-\log(p_B)}{N}}\right) \sim -\frac{1}{2c\sqrt{N}}(-1)2\sqrt{\frac{-\log(p_B)}{N}} = \frac{\sqrt{-\log(p_B)}}{cN}.$$

So the rate scales as $1/N$. Now we have:

$$|\sigma(x_0)/\sigma(x_1) - 1| \leq |\exp(\pm r\|x_0 - x_1\|) - 1| \leq \exp(r\|x_0 - x_1\|) - 1 \leq$$

$$\exp(\frac{\sqrt{-\log(p_B)}}{cN}C\sqrt{N}) - 1 \sim \frac{\sqrt{-C\log(p_B)}}{c\sqrt{N}}.$$

### B.5 Does the curse of dimensionality apply in multi-class regime?

In the main text, we presented a setup, where $\overline{p_B}$ is set to be $1 - p_A$. This is equivalent to pretending that we have just 2 classes. By not estimating the proper value of $p_B$ we lose some amount of power and the resulting certified radius is smaller than it could have been, did we have the $\overline{p_B}$ as well. This is most pronounced for datasets with many classes. The natural question, therefore, is, whether we could avoid the curse of dimensionality by properly estimating the $p_B$ together with $p_A$. The answer is no. The problem is that the the theory in Section 2 already implicitly works with the estimate of $p_B$ in a form of $1 - p_A$. The theory would work also with any other estimate of $p_B$. Assuming constant $p_B$, instead of constant $p_A$, as we did in Section 2, will, therefore, yield the same conclusions. Moreover, there is neither theoretical, nor practical reason, why should $p_B$ decrease with increasing dimension.

This insight even applies to the question of the usage of input-dependent RS in practice. The assumption $p_B = 1 - p_A$ is no more important in Section 3 than in Section 2. Therefore, we can apply our method also for the $\overline{p_B}$ obtained directly by Monte-Carlo sampling for the class $B$ (or by any other estimation method).

## C Concurrent work

As we mention in Section 1, the idea to use input-dependent RS is not new. It has popped out in years 2020 and 2021 in at least four works from three completely distinct groups of authors, even though none of these works has been successfully published yet. We find it necessary to comment on all of these works because of two orthogonal reasons. First, it is a good practice to compare our work with the concurrent work to see what are pros and cons of these similar approaches and to what extend the approaches differ. Second, we are convinced, that three of these four works claim results, which are not mathematically valid. We find this to be a particularly critical problem in a domain such as certifiable robustness, which is by definition based on rigorous, mathematical certifications.

### C.1 The work of Wang et al. (2021)

This work, submitted for the ICLR conference 2021 is the only work that seems to be mathematically functional. In this work, authors have two main contributions – first, they propose a two-phase training, where in the second phase, for each sample $x_i$, roughly the optimal $\sigma_i$ is being found and then this sample $x_i$ is being augmented with this $\sigma_i$ as an augmentation standard deviation. Authors call this method *pretrain to finetune*. Second, they provide a specific version of input-dependent RS. Essentially, they try to overcome the mathematical problems connected to the usage of non-constant $\sigma(x)$ by splitting the input space in so called *robust regions* $R_i$, where the constant $\sigma_i$ is guaranteed to be used. All the certified balls are guaranteed to lie within just one of these robust regions, making sure that within one certified region, constant level of $\sigma$ is used. Authors test this method on CIFAR10 and MNIST and show, that the method can outperform existing state-of-the-art approaches, mainly on the more complex CIFAR10 dataset.

However, we make several points, which make the results of this work, as well as the proposed method less impressive:

- The computational complexity of both their train-time and test-time algorithms seems to be quite high.

- The final smoothed classifier depends on the order of the incoming samples. As a consequence, it is not clear, whether the method works well for any permutation of the would-be tested samples. This creates another adversarial attack possibility - to attack the final smoothed classifier by manipulating the test set so that the order of samples is inappropriate for the good functionality of the final smoothed classifier.

- Even more, the fact, that the smoothed classifier depends on the order of the would-be tested samples makes it necessary, that the same smoothed classifier is used all the time for some test session in a real-world applications. For instance, a camera recognizing faces to approve an entry to a high-security building would need to keep the same model for its whole functional life, because restarting the model would enable attackers to create attacks on the predictions from the previous session. This might lead to significant restrictions on the practical usability of this method.

## C.2 THE WORKS OF ALFARRA ET AL. (2020) AND EIRAS ET AL. (2021)

In these works, similarly as in the work of Chen et al. (2021), authors suggest to optimize *in each* test point $x$ for such a $\sigma(x)$, that maximizes the certified radius given by Zhai et al. (2020), which is an extension of Cohen et al. (2019)'s certified radius for soft smoothing. The optimization for $\sigma(x)$ differs but is similar in some respect (as will be discussed).

Besides, all three works further propose input-dependent training procedure, for which $\sigma(x)$ - the standard deviation of gaussian data augmentation is also optimized. Altogether, both authors claim strong improvements over all the previous impactful works like Cohen et al. (2019); Zhai et al. (2020); Salman et al. (2019). The only significant difference between the works of Alfarra et al. (2020) and Eiras et al. (2021) (which have strong author intersections) is that in Eiras et al. (2021), authors build upon Alfarra et al. (2020)'s work and move from the isotropic smoothing to the smoothing with some specific anisotropic distributions.

As mentioned, authors first deviate from the setup of Cohen et al. (2019) and turn to the setup introduced by Zhai et al. (2020), i.e. they use soft smoothed classifier $G$ defined as

$$G_F(x)_C = \mathbb{E}_{\delta \sim \mathcal{N}(0,\sigma^2 I)} F(x+\delta)_C.$$

The key property of soft smoothed classifiers is that the Cohen et al. (2019)'s result on certified radius holds for them too.

**Theorem 16** (certified radius for soft smoothed classifiers). Let $G$ be the soft smoothed probability predictor. Let $x$ be s.t.

$$G(x)_A \geq \underline{E_A} \geq \overline{E_B} \geq G(x)_B.$$

Then, the smoothed classifier $g$ is robust at $x$ with radius

$$R = \frac{\sigma}{2}(\Phi^{-1}(\underline{E_A}) - \Phi^{-1}(\overline{E_B})) = \sigma\frac{\Phi^{-1}(\underline{E_A}) + \Phi^{-1}(1 - \overline{E_B}))}{2},$$

where $\Phi^{-1}$ denotes the quantile function of standard normal distribution.

*Proof.* Is provided in Zhai et al. (2020). $\square$

Note, that it is, similarly as in the hard randomized smoothing version of this theorem, essential to provide lower and upper confidence bounds for $G(x)_A$ and $G(x)_B$, otherwise we cannot use this theorem with the required probability that the certified radius is valid. Denote $G(x, \sigma)$ to be the soft smoothed classifier using $\sigma$ in $x$. Authors propose to use the following theoretical $\sigma(x)$ function:

$$\sigma(x) = \arg\max_{\sigma>0}\frac{\sigma}{2}(\Phi^{-1}(G(x,\sigma(x))_A) - \Phi^{-1}(G(x,\sigma(x))_B)). \tag{2}$$

It is of course not possible to optimize for this particular function since it is not known. It is also not feasible to run the Monte-Carlo sampling for each $\sigma$, because that is too costly and moreover due

---

**Algorithm 1** Data dependent certification (Alfarra et al., 2020)

**function** OPTIMIZESIGMA($F, x, \beta, \sigma_0, M, K$):
    **for** $k = 0, \ldots, K$ **do**
        sample $\delta_1, \ldots, \delta_M \sim \mathcal{N}(0, I)$
        $\phi(\sigma_k) = \frac{1}{M} \sum\limits_{i=1}^{M} F(x + \sigma\delta_i)$
        $\hat{E}_A(\sigma_k) = \max_C \phi(\sigma_k)_C$
        $\hat{E}_B(\sigma_k) = \max_{C \neq A} \phi(\sigma_k)_C$
        $R(\sigma_k) = \frac{\sigma_k}{2}(\Phi^{-1}(\hat{E}_A(\sigma_k)) - \Phi^{-1}(\hat{E}_B(\sigma_k)))$
        $\sigma_{k+1} \leftarrow \sigma_k + \beta\nabla_{\sigma_k} R(\sigma_k)$
    $\sigma^* = \sigma_K$
    **return** $\sigma^*$

---

to stochasticity, it would lead to discontinuous function. Treatment of this problem is probably the most pronounced difference between the works of Alfarra et al. (2020) and Chen et al. (2021).

Alfarra et al. (2020) use the following easy observation: $\mathcal{N}(0, \sigma^2 I) \equiv \sigma\mathcal{N}(0, I)$. Assume we have $\delta_i, i \in \{1, \ldots, M\}$ be i.i.d. sample from $\mathcal{N}(0, I)$. Obviously, $G(x, \sigma(x))_A \sim \frac{1}{M} \sum\limits_{i=1}^{M} F(x + \sigma\delta_i)_A$, since this is just the empirical mean of the theoretical expectation. Then, Expression 2 can be approximated as:

$$\sigma(x) = \arg\max_{\sigma > 0} \frac{\sigma}{2}\left(\Phi^{-1}\left(\frac{1}{M}\sum_{i=1}^{M} F(x + \sigma\delta_i)_A\right) - \Phi^{-1}\left(\frac{1}{M}\sum_{i=1}^{M} F(x + \sigma\delta_i)_B\right)\right). \quad (3)$$

Here, $M$ is the number of Monte-Carlo samplings used to approximate this function. Note, that this function is a random realization of stochastic process in $\sigma$ which is driven by the stochasticity in the sample $\delta_i, i \in \{1, \ldots, M\}$. To find the maximum of this function, authors furhter propose to use simple gradient ascent, which is possible due to the simple differentiable form of Expression 3. This differentiability is one of the main motivations to switch from hard to soft randomized smoothing. Now, we are able to state the exact optimization algorithm of Alfarra et al. (2020):

Note, that being done in this way, this algorithm can be viewed as a stochastic gradient ascent. After obtaining $\sigma^* \equiv \sigma(x)$, authors further run the Monte-Carlo sampling to estimate the certified radius exactly as in Cohen et al. (2019), but with $\sigma(x)$ instead of some global $\sigma$. Using this algorithm, authors achieve significant improvement over the Cohen et al. (2019)'s results, particularly getting rid of the first problem mentioned in Appendix A, the truncation issue. For the results, we refer to Alfarra et al. (2020). We will now give several comments on this algorithm and this method.

To begin with, in this optimization, authors do not adjust the estimated expectations and therefore don't use lower confidence bounds, but rather raw estimates. This is not incorrect, since these estimates are not used directly for the estimation of certified radius, but it is inconsistent with the resulting estimation. In other words, authors optimize for a slightly different function than they then use. The difference is, however, not very big apart from extreme values of $E_A$, where the difference might be really significant.

To overcome slightly this inconsistence, authors further (without comment) use clamping of the $\hat{E}_A(\sigma_k)$ and $\hat{E}_B(\sigma_k)$ on the interval $[0.02, 0.98]$. I.e. if $\hat{E}_A(\sigma_k) > 0.98$, it will be set to 0.98 and this is also taken into account in the computation of gradients. This way, authors get rid of the inconvenient issue, that if $G(x)_A \sim 1$, then $\hat{E}_A(\sigma_k) \sim 1$ for $\sigma_k \sim 0$, what might cause very big value of $\Phi^{-1}(\hat{E}_A(\sigma_k))$, yielding strong inconsistency with what would be obtained, if lower confidence bound was used instead.

However, the clamping causes even stronger inconsistence in the end. Note, that if $G(x)_A \sim 1$, then the true value of $E_A(\sigma_k)$ would be really close to 1, yielding high values of $\Phi^{-1}(E_A(\sigma_k))$. This value would be far better approximated by the lower confidence bound than with the clamping, since

the lower confidence bound of 1 for $M = 100000$ and $\alpha = 0.001$ is more than 0.9999, while the clamped value is just 0.98. This makes small values of $\sigma$ highly disadvantageous, since $\frac{\sigma}{2} \to 0$ as $\sigma \to 0$, yet $\Phi^{-1}(\hat{E}_A(\sigma_k))$ is being stuck on $\Phi^{-1}(0.98)$. In other words, this way authors artificially force the resulting $\sigma(x)$ to be big enough, s.t. $E(\sigma(x))_A \leq 0.98$. This assumption is not commented in the article and might result in intransparent behaviour.

Second of all, authors use $M = 1$ for their experiments. This can be interpreted as using batch size 1 in classical SGD. We suppose that this small batch size is suboptimal since it yields an insanely high variance of the gradient.

Third of all, during the search for $\sigma(x)$, it is not taken into account, whether the prediction is correct or not. This is, of course, a scientifically correct approach, since we cannot look at the label of the test sample before the very final evaluation. However, it is also problematic, since the function in Expression 2 might attain its optimum in such a $\sigma(x)$, which leads to misclassification. This could have been avoided if constant $\sigma$ was used instead.

To further illustrate this issue, assume $F(x) = \mathbf{1}(B_1(0))$, i.e. $F(x)$ predicts class 1 if and only if $\|x\| \leq 1$, otherwise predicts class 0. Assume we are certifying $x \equiv 0$ and assume that $\sigma_0$ in Algorithm 1 is initialized such that class 0 is already dominating. Then, we will have positive gradient $\nabla_{\sigma_k} R(\sigma_k)$ in all steps, because $F(\sigma\delta_i)$ is obviously non-increasing, so the number of points classified as class 1 for fixed sample $\delta_i, i \in 1, \ldots, M$ is decreasing, yielding $\hat{E}_A(\sigma_k)$ non-decreasing in $\sigma_k$, while $\frac{\sigma_k}{2}$ strictly increasing in $\sigma_k$. This way, the $\sigma_k$ will diverge to $\infty$ for $k \to \infty$. However, point $x \equiv 0$ is classified as class 1, yielding misclassification which is, moreover, assigned very high certified radius.

This issue is actually even more general - the function in Expression 2 does in most cases (assuming infinite region $\mathbb{R}^N$) *not* possess global maximum, because usually

$$\lim_{\sigma \to \infty} \frac{\sigma}{2}(\Phi^{-1}(G(x, \sigma(x))_A) - \Phi^{-1}(G(x, \sigma(x))_B)) = \infty.$$

This can be seen, for instance, easily for the $F(x) = \mathbf{1}(B_1(0))$, but it is the case for any hard classifier, for which one region becomes to have dominating area as the radius around some $x_0$ goes to infinity. This is because, if some region becomes to be dominating (for instance if all other regions are bounded), then $\frac{\sigma}{2}$ grows, while $\Phi^{-1}(G(x, \sigma(x))_A) - \Phi^{-1}(G(x, \sigma(x))_B)$ either grows too, or stagnates, making the whole function strictly increasing with sufficiently high slope.

This issue also throws the hyperparameter $K$ under closer inspection. What is the effect of this hyperparameter on the performance of the algorithm? From the previous paragraph, it seems, that this parameter serves not only as the " scaled number of epochs", but also as some stability parameter, which, however, does not have theoretical, but rather practical justification.

Another issue is, that the function in Expression 2 might be non-convex and might possess many different local minima, from which not all (or rather just a few) are actually reasonable. Therefore, the Algorithm 1 is very sensitive to initialization $\sigma_0$.

However, probably the biggest issue of all is connected to the impossibility result showed in Section 2, which shows, that the Algorithm 1 actually yields invalid certified radiuses. Why it is so?

First of all, we must justify, that our impossibility result is applicable also for the *soft* randomized smoothing. This is because classifiers of type $F(x)_C = \mathbf{1}(x \in R_C)$ for $R_C$ being decision region for class $C$ are among applicable classifiers s.t. $G(x, \sigma)_A = E_A$. With such classifiers, however, there is no difference between soft and hard smoothing and moreover $E_A \equiv p_A$ from our setup. This way we can construct the worst-case classifiers $F^*$ exactly as in our setup and therefore the same worst-case classifiers and subsequent adversarial examples are applicable here as well. In other words, for fixed value of soft smoothed $G(x, \sigma)_A = E_A$ we can denote $p_A = E_A$ and find the worst-case hard classifier $F$ defined as indicator of the worst-case ball, which will yield $E_B \equiv \mathbb{P}_1(B)$ from Theorem 3 in some queried point $x_1$.

As we have seen in previous paragraphs, the resulting $\sigma(x)$ yielded in Algorithm 1 is very instable and stochastic - it depends heavily on $F, \sigma_0, K, \beta, M$ and of course $\delta_i, i \in \{1, \ldots, M\}$ for each iteration of the for cycle. Now, for instance for CIFAR10 and $p_A = 0.99$, we have the minimal possible ratio $\frac{\sigma_0}{\sigma_1}$ equal to more than 0.96. It is hard to believe, that such instable, highly stochastic and non-regularized (except for $K, \beta$) method will yield $\sigma(x)$ sufficiently slowly varying such that

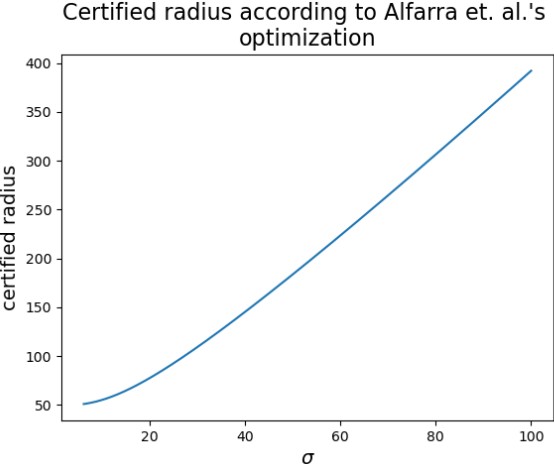

Figure 14: The theoretical certified radius as in Expression 2. The function is monotonically increasing on interval $[0, 100]$ and will further be increasing too.

within the certified radius around $x_0$, there will be no $x_1$ for which $\sigma_1$ deviates more than by this strict threshold from $\sigma_0$. This is even more pronounced on ImageNet, where the minimal possible ratio $\frac{\sigma_1}{\sigma_0}$ is above 0.99 for any $p_A$ or $E_A$.

Even without the help of curse of dimensionality, we can construct a counterexample for which the algorithm will not yield valid certified radius. Assume again $F(x) = \mathbf{1}(B_1(0))$ and assume modest dimension $N = 2$. Assume we try to certify point $x_0 \equiv [50, 0]$. Then, the theoretical $\sigma$-dependent function from Equation 2 is depicted on Figure 14.

We can see, that the resulting $\sigma(x_0)$ will be as big as our regularizers $K$ and $\beta$ in Algorithm 1 will allow. Therefore, if we run the algorithm for $K$ high-enough, surely the resulting certified radius will be far bigger than 50. However, if we certify the point $x_0 \equiv [0, 0]$ and we start with $\sigma_0 = 0.25$, for instance, then the $\sigma$-dependent certified radius in Expression 2 will be decreasing in this $\sigma_0$, yielding $\sigma(x_0) < 0.25$, which will result in classification of class 1. This point $[0, 0]$ lies within the "certified" range of $[50, 0]$, yet it is not classified the same, because, obviously, $[50, 0]$ is classified as class 0. This is therefore a counterexample to the validity of Alfarra et al. (2020)'s certification method and their results.

Note that even though our counterexample is a bit "extreme" and one could argue that in practice such a situation would not occur, we must emphasize, that this counterexample is constructed even without the help of the curse of dimensionality. In practice, it fully suffices, that for $x_0$ and some certified radius $R$ in $x_0$, there exists $x_1$ within the range of this certified radius, s.t. $\sigma_1$ is quite dissimilar to $\sigma_0$. If such situation occurs, then $R$ surely is *not* a valid certified radius.

### C.3 THE WORK OF CHEN ET AL. (2021)

The methodology of Chen et al. (2021) is rather similar to that of Alfarra et al. (2020). The biggest difference consists in the optimization of Expression 2.

Instead of stochastic gradient descent, they use more sophisticated version of grid search - so called *multiple-start fast gradient sign search*. Simply speaking, this method first generates a set of pairs $(\sigma_0, s)_i, i \in \{1, \ldots, K\}$ and then for each of the $i$ runs, it runs a $j$-while cycle, where in each step $j$, it increases $\sigma_j^2 = \sigma_0^2 + js$ to $\sigma_{j+1}^2 = \sigma_0^2 + (j+1)s$ and checks, whether the $\sigma$-dependent empirical certified radius in Expression 3 increases or not. If yes, they continue until $j$ is above some threshold $T$, if not, they break and report $\sigma_i$ as the $\sigma_j$ from the inner step where while cycle was broken. After obtaining $\sigma_i$ for $i \in \{1, \ldots, K\}$, they choose $\sigma(x)$ to be the one, that maximizes Expression 3. More concretely, their multiple-start fast gradient sign search algorithm looks as follows:

It is not entirely clear from the text of Chen et al. (2021), how exactly are $l_k, s_k$ sampled, but it is written there, that the interval for $l$ is $[1, 16]$ and for $s$ it is $(-1, 1)$. Moreover, the authors don't

---

**Algorithm 2** Instance-wise multiple-start FGSS (Chen et al., 2021)

> **function** OPTIMIZESIGMA($F, x, \sigma_0, M, K, T$):
>> generate $(l_k, s_k), k \in \{1, \ldots, K\}$
>> **for** $k = 1, \ldots, K$ **do**
>>> sample $\delta_1, \ldots, \delta_M \sim \mathcal{N}(0, l_k \sigma_0^2 I)$
>>> $\phi(\sqrt{l_k}\sigma_0) = \frac{1}{M} \sum\limits_{i=1}^{M} F(x + \delta_i)$
>>> $\hat{E}_A(\sqrt{l_k}\sigma_0) = \max_C \phi(\sqrt{l_k}\sigma_0)_C$
>>> $R(\sqrt{l_k}\sigma_0) = \sqrt{l_k}\sigma_0 \Phi^{-1}(\hat{E}_A(\sqrt{l_k}\sigma_0))$
>>> $m_k = R(\sqrt{l_k}\sigma_0)$
>>> **while** $l_k \in [1, T]$ **do**
>>>> sample $\delta_1, \ldots, \delta_M \sim \mathcal{N}(0, (l_k + s_k)\sigma_0^2 I)$
>>>> $\phi(\sqrt{l_k + s_k}\sigma_0) = \frac{1}{M} \sum\limits_{i=1}^{M} F(x + \delta_i)$
>>>> $\hat{E}_A(\sqrt{l_k + s_k}\sigma_0) = \max_C \phi(\sqrt{l_k + s_k}\sigma_0)_C$
>>>> $R(\sqrt{l_k + s_k}\sigma_0) = \sqrt{l_k + s_k}\sigma_0 \Phi^{-1}(\hat{E}_A(\sqrt{l_k + s_k}\sigma_0))$
>>>> **if** $R(\sqrt{l_k + s_k}\sigma_0) \geq R(\sqrt{l_k}\sigma_0)$ **then**
>>>>> $l_k \leftarrow l_k + s_k$
>>>>> $m_k = R(\sqrt{l_k + s_k}\sigma_0)$
>>>> **else**
>>>>> break
>>> $\sigma(x) = \max\limits_{k \in \{1, \ldots, K\}} m_k$
>> **return** $\sigma(x)$

---

provide the code and from the text, it seems, that they don't use lower confidence bounds during the evaluation of certified radiuses, what we consider to be a serious mistake (if really the case). However, we add some comments to this method regardless of the lower confidence bounds.

Generally, this method possesses most of the disadvantages mentioned in Section C.2. They use the same function for optimization, the Expression 2 and its empirical version 3. This means, that the method suffers from having several local optima, having no global optimum in general (and in most cases with limit infinity). Similarly like before, here is also no control over the correctness of the prediction, i.e. many or all local optima might lead to misclassification.

On the other hand, in this paper authors use $M = 500$ (the effective batch size), which is definitely more reasonable than $M = 1$ as in Alfarra et al. (2020). Furthermore, they use multiple initializations, making the optimization more robust and improving the chances to obtain global, or at least very good local minimum.

However, the main problem, the curse of dimensionality yielding invalid results is even more pronounced here. Unlike the "continuous approach" in Alfarra et al. (2020), here authors for each $x_0$ sample just some discrete grid (more complex, since there are more initializations) of possible values of $\sigma(x)$. For instance, if $s = 1$, then the smallest possible ratio between two consecutive $l$'s in the Algorithm 2 is $\sqrt{15}/4 \sim 0.97$, making it impossible to certify some $x_1$ w.r.t. $x_0$ if for both $s = 1$ and $l_0 \neq l_1$ on ImageNet and also for a lot of samples on CIFAR10. Of course, the fact that $s$ is randomly sampled from $(-1, 1)$ makes this counter-argumentation more difficult, but it is, again, highly unlikely that this highly stochastic method without control over $\sigma(x)$ would yield function with sufficiently small semi-elasticity. Therefore also the impressive results of Chen et al. (2021) are, unfortunately, scientifically invalid.

## D   IMPLEMENTATION DETAILS

Even though our algorithm is rather easy, there are some perks that should be discussed before one can safely use it in practice. First, we show the actual Algorithm 3

---

**Algorithm 3** Pseudocode for certification and prediction of my method based on Cohen et al. (2019)

---

*# evaluate $g$ at $x_0$*
**function** PREDICT$(f, \sigma_0, x_0, n, \alpha)$:
    `counts` $\leftarrow$ SampleUnderNoise$(f, x_0, n, \sigma_0)$
    $\hat{c}_A, \hat{c}_B \leftarrow$ two top indices in `counts`
    $n_A, n_B \leftarrow$ `counts`$(c_A)$, `counts`$(\hat{c}_B)$
    **if** BinomPValue$(n_A, n_A + n_B, 0.5) \leq \alpha$ **then return** $\hat{c}_A$
    **else return** ABSTAIN

*# certify the robustness of $g$ around $x_0$*
**function** CERTIFY$(f, \sigma_0, x_0, n_0, n, \alpha)$:
    `counts0` $\leftarrow$ SampleUnderNoise$(f, x_0, n_0, \sigma_0)$
    $\hat{c}_A \leftarrow$ top index in `counts0`
    `counts` $\leftarrow$ SampleUnderNoise$(f, x_0, n, \sigma_0)$
    $\underline{p_A} \leftarrow$ LowerConfBound(`counts`$[\hat{c}_A], n, 1 - \alpha)$
    **if** $\underline{p_A} > 1/2$ **then return** prediction $\hat{c}_A$ and radius ComputeCertifiedRadius$(\sigma_0, r, N, \underline{p_A}, \text{num\_steps})$
    **else return** ABSTAIN

**function** COMPUTECERTIFIEDRADIUS$(\sigma_0, r, N, \underline{p_A}, \text{num\_steps})$
    `radiuses` $\leftarrow$ linspace(num_space)
    **for** $R$ in `radiuses` **do**
        $\sigma_{11} \leftarrow \sigma_0 \exp(-rR)$
        $\sigma_{12} \leftarrow \sigma_0 \exp(rR)$
        `xi_bigger` $\leftarrow \xi_>(R, \sigma_{11})$
        `xi_lower` $\leftarrow \xi_<(R, \sigma_{12})$
        **if** $\max\{$`xi_bigger`, `xi_lower`$\} > 0.5$ **then** BREAK
    **return** $R$

---

Note that the function `ComputeCertifiedRadius` is a bit more complicated than depicted in Algorithm 3. We don't use a simple for-loop, but rather quite an efficient search method.

Theoretically speaking, this algorithm works perfectly. However, in practice, it is a bit problematic. The issue is, that since we use $\sigma_{11}$ and $\sigma_{12}$, which are extremely close to $\sigma_0$ for small tested radiuses $R$, the NCCHSQ functions will get extremely high inputs, making the results numerically instable. To prevent this, we use a simple trick. Since the more extreme $\sigma_1$ will we assume in evaluation at particular distance $R$, the worse for us, we can prevent numerical issues simply by putting $\sigma_t < \sigma_0$ and $\sigma_T > \sigma_0$ to be maximal and minimal used $\sigma$'s in our evaluation, i.e. the true $\sigma$ used will be $\min\{\sigma_t, \sigma_0 \exp(-rR)\}$ and $\max\{\sigma_T, \sigma_0 \exp(rR)\}$. This way, we avoid numerical issues, because we can put $\sigma_t, \sigma_T$ to be s.t. $\frac{1}{\sigma_0^2 - \sigma_t^2}$ is not too big and in the same time maintaining the correct certification thanks to the Lemma 6. The problems of this workarounds are first that it decreases the certification power, since it assumes $\sigma_1$'s that are even worse than the theoretically guaranteed worst-case possibilities and second, more importantly, that it requires some engineering to design the $\sigma_t, \sigma_T$ designs. It is submoptimal to put one constant value for these thresholds, because the numerical problems occur at different ratio thresholds of $\sigma_1/\sigma_0$ for different class probabilities $\underline{p_A}$ *and* the dimension $N$. This requires to design a specific $\sigma_t(\underline{p_A})$ and $\sigma_T(\underline{p_A})$ functions for each dimension $N$ which we want to apply. For instance, we use $\sigma_t(\underline{p_A}) = 0.9993 + 0.001 \log_{10}(\overline{p_B})$, and $\sigma_T(\underline{p_A}) = 1/\sigma_t(\underline{p_A})$ for CIFAR10, while for MNIST we use $\sigma_t(\underline{p_A}) = 0.9988 + 0.001 \log_{10}(\overline{p_B})$, and $\sigma_T(\underline{p_A}) = 1/\sigma_t(\underline{p_A})$. To design such functions, one needs to plot `plot_real_probability_of_a_ball_with_fixed_variances_as_fcn_of_dist` function, which computes the $\xi$ functions, for particular $N$ and several values of $\underline{p_A}$ and look, whether it computes correctly. As an example, we provide such a plots for well and ill working setups on Figure 15.

Another performance trick is to not evaluate $\xi$ for each $R_i$, where $R_i$ is $i$-th grid point of evaluation, but rather evaluate sequentially $R_{i^2}$, i.e. just every $i^2$-th point, until we reach value $> 0.5$ and then to search just the interval $[(i-1)^2, i^2]$, where $i$ is the first iteration for which $\xi_>(R_{i^2}, \sigma_1) \geq 0.5$.

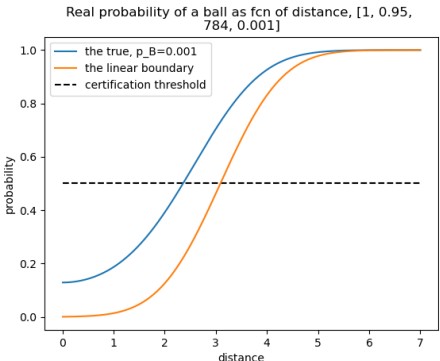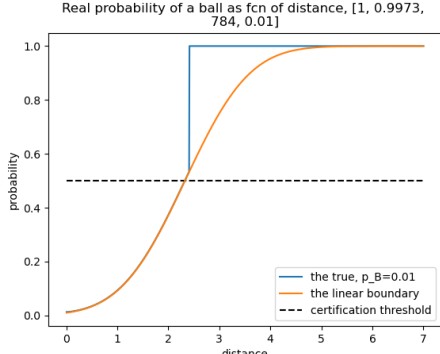

Figure 15: Well and ill working function
`plot_real_probability_of_a_ball_with_fixed_variances_as_fcn_of_dist`,
which computes the $\xi$ functions. The coding is $[\sigma_0, \sigma_1, N, p_A]$.

# E   MORE TO EXPERIEMNTS AND ABLATIONS

Before we present our further results, we must emphasize that our certification procedure is barely any slower than that of Cohen et al. (2019). More specifically, given 100000 iterations of monte-carlo sampling, certification of one sample using Cohen et al. (2019)'s algorithm on CIFAR10 takes $\sim 15$ seconds on our machine, while certification of a sample using our Algorithm 3 takes $15 - 20$ seconds depending on the $\sigma_b, r$ setup. If at least one of $\sigma_b$ and $r$ is not small, then our method runs practically instantly. If both parameters are small, then one evaluation can take up to 5 seconds depending on the exact value of parameters and on the $\underline{p_A}$. Note, that this part of the certification is dimension-independent and therefore can run in the same time also on much higher-dimensional problems.

Besides the actual certification, we have to compute $\sigma(x)$ for each of the test examples. This part of the algorithm is being executed before the actual certification and usually takes around 1 minute on our machine and on CIFAR10.

All in all, even in the really worst-case scenario, our method runs at most $1/3$-times longer than the old method on CIFAR10. On MNIST, the ratio between our run and the original run is higher, since MNIST is smaller-dimensional problem. However, since our part of evaluation is practically independent of the setup (except the values of $\sigma_b$ and $r$, which, however, can yield just some upper-bounded amount of slow-down), our algorithm does not bring any added asymptotic time complexity.

## E.1   HOW TO CHOOSE THE HYPERPARAMETERS?

Our design of $\sigma(x)$ function defined in Equation 1 uses several hyperparameters. These are: $r$ for the rate, $m$ for the scaling, $\sigma_b$ for the base sigma and $k$ for the $k$-nearest neighbors. How do we choose these hyperparameters?

The $m$ parameter depends on our goals. We can set it so that $\sigma(x)$ achieves lowest values at $\sigma_b$ by setting it so that it is roughly equal to the minimal distance from $k$ nearest neighbors across, for instance, training samples. Other possibility is to set it so that it is roughly equal to the average distance from $k$ nearest neighbors, to ensure that the average $\sigma(x)$ will roughly correspond to the $\sigma_b$.

The $k$ parameter needs to be set with two objectives in mind. Firstly, it would be unwise to set it too small, because then the distance from $k$ nearest neighbors would be too noisy. On the other hand, we don't want it too high, because then it will not be changing fast enough with changing the position of $x$. The suitable value can be obtained by looking at histograms of average distances from $k$ nearest neighbors and choosing the $k$ for which the histogram is enough scattered, but it is not too small.

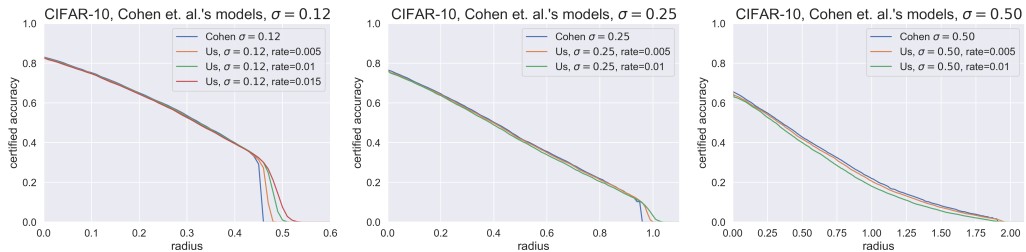

Figure 16: Comparison of certified accuracy plots for Cohen et al. (2019) and our work. For each plot, the same base model $f$ is used for evaluation.

|  | $\sigma = 0.12$ | $\sigma = 0.25$ | $\sigma = 0.50$ |
|---|---|---|---|
| $r = 0.00$ | 0.831 | 0.766 | 0.658 |
| $r = 0.005$ | 0.830 | 0.766 | 0.654 |
| $r = 0.01$ | 0.828 | 0.762 | 0.649 |
| $r = 0.015$ | 0.826 | - | - |

Table 6: Clean accuracies for Cohen's models and our non-constant $\sigma(x)$ models.

|  | $\sigma = 0.12$ | $\sigma = 0.25$ | $\sigma = 0.50$ |
|---|---|---|---|
| $r = 0.00$ | 0.084 | 0.108 | 0.131 |
| $r = 0.005$ | 0.086 | 0.112 | 0.135 |
| $r = 0.01$ | 0.088 | 0.119 | 0.142 |

Table 7: Standard deviations of class-wise accuracies for different levels of $\sigma$ and $r$.

The $r$ parameter needs to be chosen so that we can have some significant advantage over constant smoothing, but it cannot be too big, because otherwise the curse of dimensionality would apply. The value can be decided either by trial and error, or by plotting the certified radius given linear classifier, or from Theorem 4, setting the rate low-enough so that within the expected certified radius range, the ratio $\frac{\sigma_1}{\sigma_0}$ can't move anywhere near the theoretical thresholds implied by Theorem 4.

The $\sigma_b$ is the base $\sigma$ and should be used according to the level of smoothing variance we want to use. More discussion on this can be found in Cohen et al. (2019).

### E.2 COMPARISON WITH COHEN ET AL. (2019) METHODOLOGY ON CIFAR10 DATASETS

Here, we compare Cohen et al. (2019)'s evaluations for $\sigma = 0.12, 0.25, 0.50$ with our evaluations directly on models trained by Cohen et al. (2019), setting $\sigma_b = \sigma$, $r = 0.005, 0.01$ and $0.015$ for $\sigma_b = \sigma = 0.12$, $k = 20$ and $m = 5$. In this way, the levels of $\sigma(x)$ used in direct comparison will rise from the values roughly equal to Cohen et al. (2019)'s constant $\sigma$ to higher values. The results are depicted in Figure 16.

Note, that this evaluation is being done on the models trained directly by Cohen et al. (2019) and therefore the variance of Gaussian data augmentation is not entirely consistent with the optimal variance that should be used for non-constant $\sigma$, which should be either the same, $\sigma(x)$ or constant, but in average equal to $\sigma(x)$. The results are similar as in the Section 4. Note, that for $\sigma_b = \sigma = 0.50$, the curse of dimensionality becomes most pronounced, as explained in Appendix B. Further, we provide the Tables 6, 7, where the clean accuracies and class-wise standard deviations are displayed.

The results are, again, similar as in the section 4.

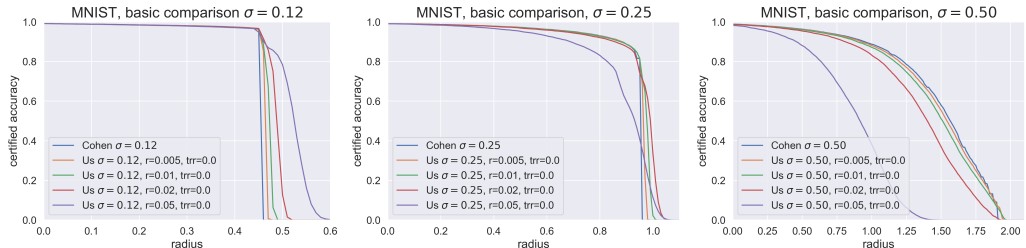

Figure 17: Comparison of certified accuracy plots for Cohen et al. (2019) and our work, MNIST. For each plot, the same base model $f$ is used for evaluation. The term $trr$ stands for *train-time rate*, will be discussed later and can be ignored now.

|            | $\sigma = 0.12$ | $\sigma = 0.25$ | $\sigma = 0.50$ |
|------------|-----------------|-----------------|-----------------|
| $r = 0.00$ | 0.9913          | 0.9910          | 0.9888          |
| $r = 0.005$| 0.9914          | 0.9912          | 0.9885          |
| $r = 0.01$ | 0.9914          | 0.9910          | 0.9887          |
| $r = 0.02$ | 0.9914          | 0.9912          | 0.9876          |
| $r = 0.05$ | 0.9914          | 0.9906          | 0.9836          |

Table 8: Clean accuracies for Cohen's models and our non-constant $\sigma(x)$ models on MNIST.

|            | $\sigma = 0.12$ | $\sigma = 0.25$ | $\sigma = 0.50$ |
|------------|-----------------|-----------------|-----------------|
| $r = 0.00$ | 0.677           | 0.729           | 0.909           |
| $r = 0.005$| 0.659           | 0.735           | 0.905           |
| $r = 0.01$ | 0.659           | 0.722           | 0.9318          |
| $r = 0.02$ | 0.659           | 0.713           | 0.960           |
| $r = 0.05$ | 0.715           | 0.796           | 1.159           |

Table 9: Standard deviations of class-wise accuracies for different levels of $\sigma$ and $r$. The printed values are multiples of 100 of the real standard deviations.

### E.3 COMPARISON WITH COHEN ET AL. (2019) METHODOLOGY ON MNIST DATASETS

Here, we present similar comparison as in Subsection E.2, but on MNIST and with models trained by us. Again, the setup is similar as in the Section 4. We compare $\sigma = \sigma_b = 0.12, 0.25, 0.50$ with test-time rates $r = 0.005, 0.01, 0.02, 0.05$ *and* train-time level of $\sigma$ again equal to $\sigma = \sigma_b$. It is important to note, that we use different normalization constant $m$ in the MNIST case. In CIFAR10, we set $m = 5$, in MNIST, the suitable $m$ is 1.5. This way we assure, that the smallest $\sigma(x)$ values in the test set will roughly equal the $\sigma_b = \sigma$. The certified accuracy plots are depicted on Figure 17. We also add the clean accuracy table and class-wise clean accuracies standard deviation table (8, 9).

All the results are, again, very similar to those presented in Section 4, even though the gain in certified accuracies is marginally worse, since our evaluations run on models trained with in average smaller train-time data-augmentation standard deviation $\sigma_b = \sigma$.

### E.4 INVESTIGATION OF THE EFFECT OF TRAINING WITH INPUT-DEPENDENT GAUSSIAN AUGMENATATION

It has been shown by many works, that apart from a good test-time certification method, also the appropriate training plays a very important role in the final robustness of our smoothed classifier $g$. Already Cohen et al. (2019) realize this and propose to train with gaussian data augmentation with constant $\sigma$. They experiment with different levels of $\sigma$ during training and conclude that training with the same level of $\sigma$ that will be later used in the test time is usually the most suitable option.

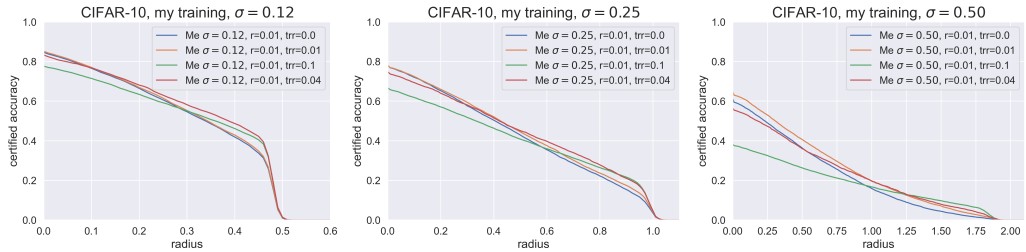

Figure 18: The certified accuracies of our procedure on CIFAR10 for $\sigma_b = 0.12, 0.25, 0.50$, rate $r = 0.01$ and training rate $trr = 0.0, 0.01, 0.04, 0.1$.

The question of best-possible training to boost the certified robustness didn't stay without the interest of different researchers. Both Zhai et al. (2020) and Salman et al. (2019) try to improve the way of training and propose two different, yet interesting and effective training methods. While Zhai et al. (2020) manage to incorporate the adversarial robustness into the training loss function, therefore training directly for the robustness, Salman et al. (2019) propose to use adversarial training to achieve more robust classifiers.

Both Alfarra et al. (2020) and Chen et al. (2021) already propose to use training with input-dependent $\sigma$ as the variance of gaussian data augmentation. Both of them proceed similarly as during test time - to obtain training $\sigma(x)$, they optimize for such, that would maximize the certified accuracy of training samples.

In this section, we propose and test our own training method. We propose to use again gaussian data augmentation with input-dependent $\sigma(x)$, but we suggest to use the simple $\sigma(x)$ defined in Equation 1. In other words, we suggest using the same $\sigma(x)$ during training as during testing (up to parametrization, which might differ).

Note, that, unlike the certification, the training procedure does not require any mathematical analysis nor certification. It is totally up to us how we train the base classifier $f$ and the way of training does not influence the validity of subsequent certification guarantees during test time. However, it is good to have a reasonable training procedure, because otherwise, we would achieve a satisfactory model neither in terms of clean accuracy nor in terms of adversarial robustness.

In the subsequent analysis, we evaluate and compare our certification procedures on models trained with different training parametrizations. For this particular section, we run the comparison only on the CIFAR10 dataset. For each test-time $\sigma_b, r$, we evaluate our method with these parameters on base models $f$ trained with the same $\sigma_b$, but different level of *training rate $trr$*. The training rate $trr$ plays exactly the same role as the evaluation rate $r$ but is used exclusively during training. Note, that this makes our $\sigma(x)$ different during training and testing since it is parametrized with different rates.

On the Figure 18 we plot evaluations on CIFAR10 of our method for rate $0.01$, all levels of $\sigma_b = 0.12, 0.25, 0.50$ and each of these test-time setups is evaluated on 4 different levels of train-time rate $trr = 0.0, 0.01, 0.04, 0.1$.

From the results, we judge, that our training procedure works satisfactorily well. It can generally outperform the constant $\sigma$ training, yet the standard accuracy vs. robustness trade-off is present in some cases. If we train with small train-time rate, the improvement of the certified accuracies is not pronounced (the case for $\sigma_b = \sigma = 0.50$ is slightly misleading, since such a configuration is just a result of the variance of clean accuracy w.r.t different traning runs) enough, but we also don't lose almost any clean accuracy. Increasing the rate to $trr = 0.04$ results in much more pronounced improvements in high certified accuracies, yet also comes at a prize of clean accuracy drop, especially for large $\sigma$ levels. Even bigger training rate, such as $trr = 0.1$ seems to be too big and does not bring almost any improvement over the rate $trr = 0.04$, yet loses a large amount of clean accuracy.

These results suggest, that the input-dependent training with a carefully chosen training rate for $\sigma(x)$ can lead to significant improvements in certifiable robustness. However, it is important to note, that

|  | $\sigma = 0.12$ | $\sigma = 0.25$ | $\sigma = 0.50$ |
|---|---|---|---|
| $trr = 0.00$ | 0.084 | 0.107 | 0.153 |
| $trr = 0.01$ | 0.078 | 0.099 | 0.126 |
| $trr = 0.04$ | 0.068 | 0.081 | 0.117 |
| $trr = 0.1$ | 0.088 | 0.099 | 0.230 |

Table 10: Standard deviations of class-wise accuracies for different levels of $\sigma$ and $trr$, under constant rate $r = 0.01$.

the optimal $trr$ seems to be dependent on the $\sigma_b$, therefore for each value of $\sigma_b$, some effort has to be invested to find the optimal hyperparameters.

Besides, we were also interested, whether using an input-dependent $\sigma(x)$ during training influences the class-wise accuracy balance. In Table 10 we report the standard deviations of class-wise accuracies.

We can observe, that unlike the pure input-dependent evaluation, the input-dependent training is partially capable of mitigating the effects of the shrinking. For instance, the $trr = 0.04$ for $\sigma_b = 0.12$ provides obvious improvement in establishing class-wise balance. Similarly successful are trainings with $trr = 0.01$ for $\sigma_b = 0.12$ and both $trr = 0.01, 0.04$ for $\sigma_b = 0.25$. Also for $\sigma = 0.50$ the mitigation is present for small-enough training rates. However, we must emphasize, that if we use too big training rate, the disbalance between class accuracies will be re-established and in some cases even magnified. Therefore, we must be careful to choose the appropriate training rate for the $\sigma_b, r$.

### E.5 WHY DO WE NOT COMPARE WITH THE CURRENT STATE-OF-THE-ART?

Briefly speaking – we could, but we don't consider it necessary. Since we claim just one type of improvement over the Cohen et al. (2019)'s model (experiment-wise) and don't claim new state-of-the-art training method, we didn't find it necessary to measure our strengths with methods of Salman et al. (2019) and Zhai et al. (2020). It is obvious that we would outperform these methods in the question of *certified accuracy waterfalls* anyway, since these methods focus on the training phase. Since we do not outperform Cohen et al. (2019) neither in terms of the clean accuracies nor in terms of class-wise accuracies, it is not our belief that we would outperform the two modern methods in these metrics. Moreover, we find the comparison with Cohen et al. (2019) most structured, since we extend the theory built by them.

### E.6 ABLATIONS

Even though our results so far might look impressive, we can't claim that it is fully due to our particular method until we exclude the possibility, that some different effects play an essential role in the improvement over Cohen et al. (2019)'s work.

To investigate, whether our particular method dominates the contribution to the performance boost, we conduct several ablation studies - first, we study the variance of our evaluations and trainings, second, we study the effect of input-dependent test-time randomized smoothing, and third, we study the effect of input-dependent train-time data augmentation.

### E.6.1 VARIANCE OF THE EVALUATION

To find out, whether there is a significant variance in the evaluation of certified radiuses, we conduct a simple experiment - we train a single model on CIFAR10 and evaluate our method on this model for the very same setup of parameters multiple times. This way, the only present stochasticity is in the Monte-Carlo sampling, which influences the evaluation of certified radiuses. We pick the parameters as follows: $\sigma_b = 0.50, r = 0.01, trr = 0.0$, since the $\sigma_b = 0.50$ turns out to have biggest variance in the training. The results are depicted in Figure 19.

From the results, it is obvious that the variance in the evaluation phase is absolutely negiable. Therefore, there is no need to run the same evaluation setup more times.

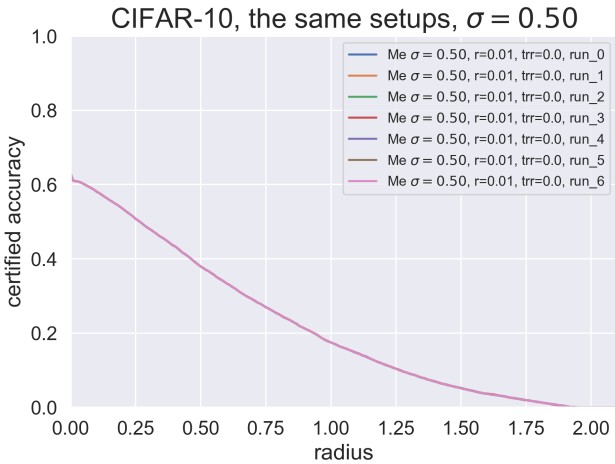

Figure 19: The variance of evaluation. Parameters are $\sigma_b = 0.50, r = 0.01, trr = 0.0$, the evaluated model is the same for all runs. There are 7 runs on CIFAR10.

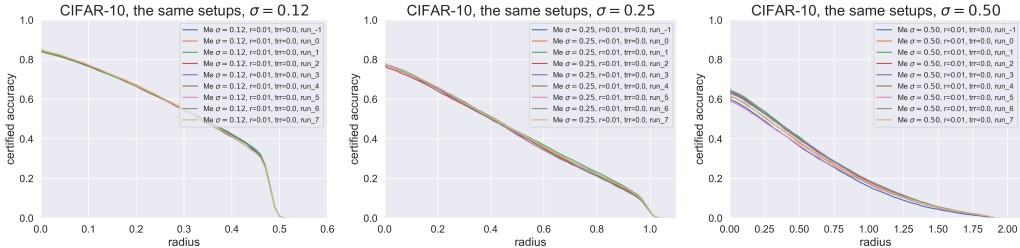

Figure 20: The certified accuracies of our procedure on CIFAR10 for $\sigma_b = 0.12, 0.25, 0.50$, rate $r = 0.01$ and training rate $trr = 0.0$ evaluated on 9 different trained models for each of the setups.

|  | $\sigma = 0.12$ | $\sigma = 0.25$ | $\sigma = 0.50$ |
|---|---|---|---|
| accuracy | 0.61% | 0.40% | 1.86% |
| abstention rate | 0.17% | 0.34% | 0.59% |
| misclassification rate | 0.60% | 0.24% | 1.48% |

Table 11: Standard deviations of clean accuracies, abstention rates and misclassification rates for 9 runs of each parameter configuration on CIFAR10.

### E.6.2    VARIANCE OF THE TRAINING

To estimate the variance of the training, we train several models for one specific training setup and evaluate them with the same evaluation setup (knowing, that there is no variance in the evaluation phase, this is equivalent to measuring directly the training variance). We pick our classical non-constant $\sigma(x)$ for the evaluation, but we train with constant variance data augmentation. The concrete parameters we work with are: $\sigma_b \in \{0.12, 0.25, 0.50\}, r = 0.01, trr = 0.0$ and we run 9 trainings for each of these parameter configurations. Then we run full certification to not only see the variance in clean accuracy, but also the variance in the certified radiuses. The results are depicted in Figure 20.

From the figures we see, that the variance of the training is strongly $\sigma_b$-dependent. Most volatile clean accuracy is present for the case $\sigma_b = 0.50$. However, fortunately, the biggest variability is present for the clean accuracy and the curves seem to be less scattered in the areas of high certified radiuses. The concrete standard deviations of clean accuracies are in Table 11. The standard deviations of clean accuracies for MNIST dataset and the same parameters are in Table 12.

|  | $\sigma = 0.12$ | $\sigma = 0.25$ | $\sigma = 0.50$ |
|---|---|---|---|
| accuracy | 0.036% | 0.042% | 0.044% |
| abstention rate | 0.037% | 0.027% | 0.058% |
| misclassification rate | 0.043% | 0.029% | 0.021% |

Table 12: Standard deviations of clean accuracies, abstention rates and misclassification rates for 8 runs of each parameter configuration on MNIST.

Since the differences in accuracies of different methods are very subtle, it is hard to obtain statistically trustworthy results. For instance, given, that the standard deviation $0.4\%$ is the true standard deviation of the $\sigma_b = 0.25$ runs, we would need 16 runs to decrease it to a standard deviation of $0.1\%$, which might be considered to be precise-enough. To do the same in the case of $\sigma_b = 0.50$ on CIFAR10, we would roughly need 400 runs to decrease the standard deviation below $0.1\%$. Therefore, the results we provide in the subsequent subsections, being the average of "just" 8 runs, have to be taken just modulo variance in the results, which might still be considerable.

### E.6.3 EFFECT OF INPUT-DEPENDENT EVALUATION

In this ablation study, we compare the certification method for particular $\sigma_b, r = 0.01, trr = 0.0$ with the constant-$\sigma$ certification method with $C\sigma_b, r = 0.0, trr = 0.0$, where $C$ is an appropriate constant. The motivation behind such an experiment is, that our $\sigma(x)$ is generally bigger than $\sigma_b$, but originally, we compare this method to constant $\sigma = \sigma_b$ evaluation. Therefore, in average, samples in our method enjoy bigger values of $\sigma(x)$. Natural question is, whether we cannot obtain the same performance boost using just the constant $\sigma$ method with $C\sigma_b > \sigma_b$ set to such value, which roughly corresponds to the average of $\sigma(x_i)$ for $x_i, i \in \{1, \dots, T\}$ being the test set. The problem of using bigger $C\sigma_b$ is, that we encounter performance drop and more severe case of shrinking, but we need to check, to what extent is the performance drop present in the input-dependent $\sigma(x)$ method. Comparing the performance drops of larger constant $C\sigma_b$ and input-dependent $\sigma(x)$, which is in average larger (but in average the same as the $C\sigma_b$), we will be able to answer, to what degree is the usage of input-dependent $\sigma(x)$ really justified. If we remind ourselves, that

$$\sigma(x) = \sigma_b \exp\left(r\left(\frac{1}{k}\left(\sum_{x_i \in \mathcal{N}_k(x)} \|x - x_i\|\right) - m\right)\right),$$

then we see, that the constant $C$ we are searching for is the average (or rather median) value of

$$\exp\left(r\left(\frac{1}{k}\left(\sum_{x_i \in \mathcal{N}_k(x)} \|x - x_i\|\right) - m\right)\right).$$

Fortunately, empirically, the mean and median of the above expression are roughly equal for both CIFAR10 and MNIST, so we are not forced to choose between them. For $r = 0.01, m = 5$, we choose the rounded value of $C = \exp(0.05)$ on CIFAR10. For $r = 0.01, m = 1.5$ as in MNIST, the constant is set to $C = 1.035$. In the end, the values of $C\sigma$ used in this experiment are $C\sigma = 0.126, 0.263, 0.53$ for CIFAR10 and $C\sigma = 0.124, 0.258, 0.517$ for MNIST. To obtain a fair comparison, though, we evaluate the input-dependent $\sigma(x)$ evaluation strategy on models trained with constant $C\sigma_b$ standard deviation of gaussian augmentation. This is because this level of $\sigma$ is equal to the mean value of the $\sigma(x)$ and we believe, that such a training data augmentation standard deviation is more consistent with our $\sigma(x)$ function. We provide the plots of single-run evaluations of certified accuracies for CIFAR10 in Figure 21 and for MNIST in Figure 22. The models on which we evaluate differ because for the increased constant $\sigma$ evaluations we needed to also use an increased level of the training data augmentation variance.

From the figures, it is obvious, that our method is not able to outperform the constant $\sigma$ method using the same mean $\sigma$ in terms of certified accuracy, not even for our strongest $\sigma_b = 0.12$. This fact might not be in general bad news, if we demonstrated, that our method suffers from less pronounced accuracy drop or less pronounced disbalance in class-wise accuracies. To find out, we measure

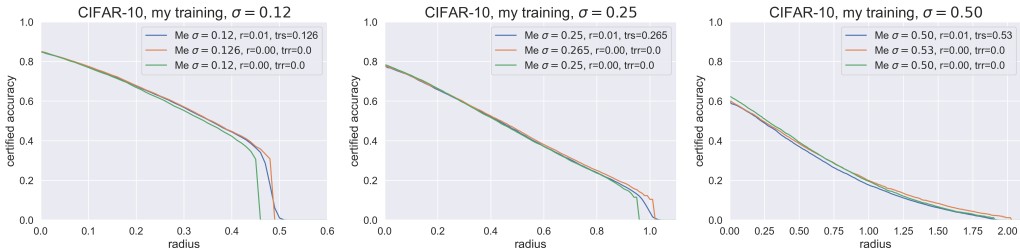

Figure 21: The certified accuracies of our procedure on CIFAR10 for $\sigma_b = 0.12, 0.25, 0.50$, rate $r = 0.01$ and constant, yet increased $C\sigma_b$ training variance, compared to certified accuracies of the constant $\sigma$ method for $\sigma = \sigma_b = 0.12, 0.25, 0.50$ and also $\sigma = C\sigma_b = 0.126, 0.265, 0.53$. Evaluated on a single training.

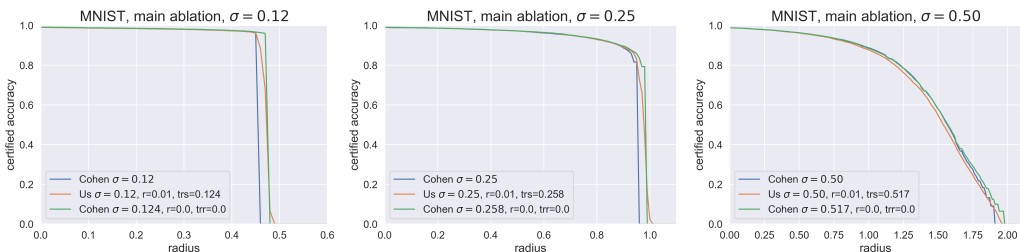

Figure 22: The certified accuracies of our procedure on MNIST for $\sigma_b = 0.12, 0.25, 0.50$, rate $r = 0.01$ and constant, yet increased $C\sigma_b$ training variance, compared to certified accuracies of the constant $\sigma$ method for $\sigma = \sigma_b = 0.12, 0.25, 0.50$ and also $\sigma = C\sigma_b = 0.124, 0.258, 0.517$. Evaluated on a single training.

|  | $\sigma = 0.12$ | $\sigma = 0.25$ | $\sigma = 0.50$ |
|---|---|---|---|
| $r = 0.01, trs$ increased | 0.852 | 0.780 | 0.673 |
| $r = 0.00$ classical | 0.851 | 0.792 | 0.674 |
| $r = 0.00$ increased | 0.853 | 0.780 | 0.673 |

Table 13: Clean accuracies for both input-dependent and constant $\sigma$ evaluation strategies on CIFAR10.

|  | $\sigma = 0.12$ | $\sigma = 0.25$ | $\sigma = 0.50$ |
|---|---|---|---|
| $r = 0.01, trs$ increased | 0.076 | 0.099 | 0.120 |
| $r = 0.00$ classical | 0.076 | 0.097 | 0.122 |
| $r = 0.00$ increased | 0.076 | 0.101 | 0.123 |

Table 14: Class-wise accuracy standard deviations for both input-dependent and constant $\sigma$ evaluation strategies on CIFAR10.

average accuracies of the evaluation strategies from 8 runs for each, as well as average class-wise accuracy standard deviations from 8 runs. The results are provided in Tables 13 and 14 for CIFAR10 and Tables 15 and 16 for MNIST.

As for CIFAR10, except for $\sigma = \sigma_b = 0.25$, the differences in accuracies between different evaluation strategies are so small, that we cannot consider them to be statistically significant. Even though the difference for $\sigma = \sigma_b = 0.25$ is high, it is still not possible to draw some definite conclusions, especially for the difference between the input-dependent $\sigma(x)$ and the increased constant $C\sigma_b$ evaluations. In general, it is not easy to judge, whether our method possesses some advantage (or disadvantage) over the increased $C\sigma_b$ method in terms of clean accuracy. Similar conclusions can be drawn in the context of the shrinking phenomenon. Here, the differences are also very small, but

|  | $\sigma = 0.12$ | $\sigma = 0.25$ | $\sigma = 0.50$ |
|---|---|---|---|
| $r = 0.01, trs$ increased | 0.9913 | 0.9905 | 0.9885 |
| $r = 0.00$ classical | 0.9914 | 0.9907 | 0.9886 |
| $r = 0.00$ increased | 0.9914 | 0.9904 | 0.9885 |

Table 15: Clean accuracies for both input-dependent and constant $\sigma$ evaluation strategies on MNIST.

|  | $\sigma = 0.12$ | $\sigma = 0.25$ | $\sigma = 0.50$ |
|---|---|---|---|
| $r = 0.01, trs$ increased | 0.00757 | 0.00798 | 0.00929 |
| $r = 0.00$ classical | 0.00751 | 0.00778 | 0.00934 |
| $r = 0.00$ increased | 0.00750 | 0.00798 | 0.00925 |

Table 16: Class-wise accuracy standard deviations for both input-dependent and constant $\sigma$ evaluation strategies on MNIST. Printed are multiples of 100 of the real values.

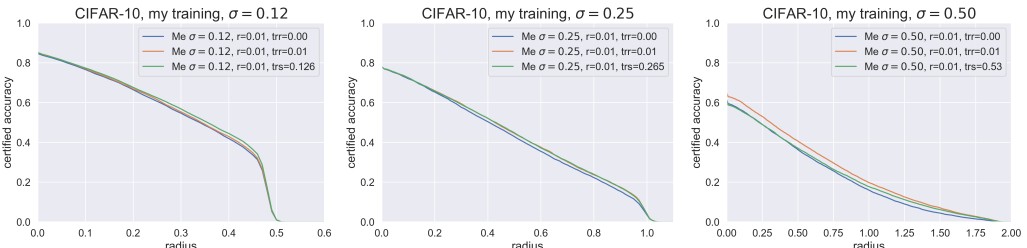

Figure 23: The certified radiuses on CIFAR10 of the non-constant $\sigma(x)$ method with rate $r = 0.01$, but different training strategies. Used training strategies are input-dependent training with the same $\sigma(x)$ function and constant-$\sigma$ training with either $\sigma_b$ or $C\sigma_b$ variance level. Evaluations are being done from single run.

unlike in the comparison with Cohen et al. (2019) models, where we evaluate our input-dependent $\sigma(x)$ method on classifiers trained with inconsistent data-augmentation variance, here we observe the general trend, that our method is able to outperform the increased constant $C\sigma_b$ evaluation. This is good news and it confirms our suspicion, that the bad results from Subsection E.2 could come from the train-test $\sigma$ inconsistency.

The results on MNIST suggest similar conclusions for the accuracy vs. robustness tradeoff. Similarly, the $\sigma = \sigma_b = 0.12, 0.50$ are not telling much, and for $\sigma = \sigma_b = 0.25$, the differences are still rather small (yet the standard deviation of the results should be $\sim 0.0001$, so it is rather on the edge). The conclusions for the shrinking phenomenon are a bit more pesimistic than in the case of CIFAR10. Here we don't see any improvement over the constant $\sigma$, not even the one with increased $\sigma$ level.

### E.6.4 Effect of input-dependent training

In this last ablation study, we compare our input-dependent data augmentation for particular $\sigma_b, r$ and particular training rate $trr$ with the constant $C\sigma_b$ data augmentation, where the training rate $trr$ is set to 0. The strategy for choosing the constant $C$ is exactly the same as in the first experiment. Particularly, we evaluate our method with $r = 0.01$ and $\sigma_b = 0.12, 0.25, 0.50$, trained with the same level of $\sigma_b$ and training rate $trr = 0.01$ with the evaluations using $r = 0.01$ and $\sigma_b = 0.12, 0.25, 0.50$ during test time, while during train time using training rate $trr = 0.0$, but using the constant $\sigma = 0.126, 0.263, 0.53$ for CIFAR10 and $\sigma = 0.124, 0.258, 0.517$ for MNIST. This way, we compensate for the "increased levels of $\sigma(x)$" with respect to $\sigma_b$. We present our comparisons in the Figure 23 for CIFAR10 and 24 for MNIST, providing the evaluations with $r = 0.01, \sigma_b = 0.12, 0.25, 0.50$ and the same $\sigma_b$ and $trr = 0.0$ during train time as a reference.

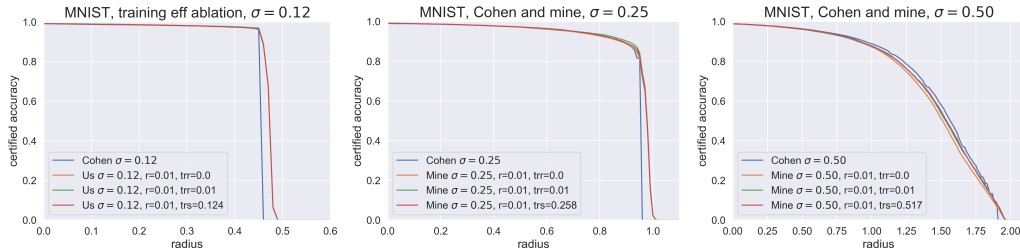

Figure 24: The certified radiuses on MNIST of the non-constant $\sigma(x)$ method with rate $r = 0.01$, but different training strategies. Used training strategies are input-dependent training with the same $\sigma(x)$ function and constant-$\sigma$ training with either $\sigma_b$ or $C\sigma_b$ variance level. Evaluations are being done from single run.

| | $\sigma = 0.12$ | $\sigma = 0.25$ | $\sigma = 0.50$ |
|---|---|---|---|
| $trr = 0.01$ | 0.843 | 0.780 | 0.671 |
| $trr = 0.00$ classical | 0.849 | 0.790 | 0.670 |
| $trr = 0.00$ increased | 0.852 | 0.780 | 0.673 |

Table 17: Clean accuracies for both input-dependent and constant $\sigma$ training strategies on CIFAR10.

| | $\sigma = 0.12$ | $\sigma = 0.25$ | $\sigma = 0.50$ |
|---|---|---|---|
| $trr = 0.01$ | 0.080 | 0.101 | 0.121 |
| $trr = 0.00$ classical | 0.080 | 0.105 | 0.135 |
| $trr = 0.00$ increased | 0.076 | 0.099 | 0.120 |

Table 18: Class-wise accuracy standard deviations for both input-dependent and constant $\sigma$ training strategies on CIFAR10.

| | $\sigma = 0.12$ | $\sigma = 0.25$ | $\sigma = 0.50$ |
|---|---|---|---|
| $trr = 0.01$ | 0.9912 | 0.9910 | 0.9883 |
| $trr = 0.00$ classical | 0.9914 | 0.9906 | 0.9884 |
| $trr = 0.00$ increased | 0.9913 | 0.9905 | 0.9885 |

Table 19: Clean accuracies for both input-dependent and constant $\sigma$ training strategies on MNIST.

| | $\sigma = 0.12$ | $\sigma = 0.25$ | $\sigma = 0.50$ |
|---|---|---|---|
| $trr = 0.01$ | 0.00757 | 0.00800 | 0.00947 |
| $trr = 0.00$ classical | 0.00743 | 0.00789 | 0.00929 |
| $trr = 0.00$ increased | 0.00757 | 0.00798 | 0.00929 |

Table 20: Class-wise accuracy standard deviations for both input-dependent and constant $\sigma$ training strategies on MNIST.

The certified accuracy results for the CIFAR10 and the MNIST differ a bit. For CIFAR10 training with rate $r = 0.01$ is does not overperform the constant $C\sigma_b$ training. For $\sigma_b = 0.12$, the constant $C\sigma_b$ training clearly outperforms the input-dependent training. For $\sigma_b = 0.25$, these two training strategies seem to have almost identical performances. For $\sigma_b = 0.50$, the input-dependent $\sigma(x)$ strategy outperforms the constant $\sigma$ ones, but now we know, that it is purely due to the variance in the training. On the other hand on MNIST, we either have very similar performance or even slightly outperform the constant $\sigma$ training.

Looking at the accuracy and standard deviation Tables 17, 18, 19 and 20, we can deduce the following. In terms of clean accuracy, the input-dependent training strategy performs worst in most of the cases, even though the differences in performance might not be statistically significant. We see, that we would need far more evaluations to see some clear pattern. However, these results are definitely not good news for the use of input-dependent $\sigma(x)$ during training.

In terms of the class-wise accuracy standard deviation, we again see countering results for CIFAR10 and MNIST datasets. For CIFAR10 the input-dependent $\sigma(x)$ clearly outperforms the smaller constant $\sigma_b$ training method, particularly for $\sigma_b = 0.50$. However, the constant $C\sigma_b$ method seem to outperform even the input-dependent $\sigma(x)$. For MNIST, the smaller constant $\sigma_b$ outperforms both other methods, while they are rather similar.

Together with findings from previous sections, these results suggest, that usage of this particular design of input-dependent $\sigma(x)$ might not be worthy until a more precise evaluation is conducted. However, the combination of input-dependent test-time evaluation with constant, yet increased train-time augmentation is possibly the strongest combination that can be achieved using input-dependent sigma at all (especially for CIFAR10).

## F PROOFS

**Lemma 17** (Neyman-Pearson). Let $X, Y$ be random vectors in $\mathbb{R}^N$ with densities $x, y$. Let $h : \mathbb{R}^N \to \{0, 1\}$ be a random or deterministic function. Then, the following two implications hold:

1. If $S = \left\{ z \in \mathbb{R}^N : \frac{y(z)}{x(z)} \leq t \right\}$ for some $t > 0$ and $\mathbb{P}(h(X) = 1) \geq \mathbb{P}(X \in S)$, then $\mathbb{P}(h(Y) = 1) \geq \mathbb{P}(Y \in S)$.

2. If $S = \left\{ z \in \mathbb{R}^N : \frac{y(z)}{x(z)} \geq t \right\}$ for some $t > 0$ and $\mathbb{P}(h(X) = 1) \leq \mathbb{P}(X \in S)$, then $\mathbb{P}(h(Y) = 1) \leq \mathbb{P}(Y \in S)$.

*Proof.* See Cohen et al. (2019). $\square$

**Lemma 1:** Out of all possible classifiers $f$ such that $G_f(x)_B \leq p_B = 1 - p_A$, the one, for which $G_f(x + \delta)_B$ is maximized is the one, which predicts class $B$ in a region determined by the likelihood ratio:

$$B = \left\{ x \in \mathbb{R}^N : \frac{f_1(x)}{f_0(x)} \geq \frac{1}{r} \right\},$$

where $r$ is fixed, such that $\mathbb{P}_0(B) = p_B$. Note, that we use $B$ to denote both the class and the region of that class.

*Proof.* Let $f$ be arbitrary classifier. To invoke the Neyman-Pearson Lemma 17, define $h \equiv f$ (with the only difference, that $h$ goes to $\{0, 1\}$ instead of $\{A, B\}$). Moreover, let $S \equiv B$ and $X \sim \mathcal{N}(x, \sigma_0^2 I), Y \sim \mathcal{N}(x + \delta, \sigma_1^2 I)$. Let also $f^*$ classify $S$ as $B$. Then obviously, $\mathbb{P}(X \in S) = \mathbb{P}_0(B) = p_B$. Since $G_f(x)_B \leq p_B$, we have $\mathbb{P}(h(X) = 1) \leq p_B$. Using directly the second part of Neyman-Pearson Lemma 17, this will yield $\mathbb{P}(Y \in S) \geq \mathbb{P}(h(Y) = 1)$. Rewritten in the words of our setup, $G_{f^*}(x + \delta) \geq G_f(x + \delta)$. $\square$

**Theorem 2:** If $\sigma_0 > \sigma_1$, then $B$ is a $N$-dimensional ball with the center at $S_>$ and radius $R_>$:

$$S_> = x + \frac{\sigma_0^2}{\sigma_0^2 - \sigma_1^2} \delta, \ R_> = \sqrt{\frac{\sigma_0^2 \sigma_1^2}{(\sigma_0^2 - \sigma_1^2)^2} \|\delta\|^2 + 2N \frac{\sigma_0^2 \sigma_1^2}{\sigma_0^2 - \sigma_1^2} \log\left(\frac{\sigma_0}{\sigma_1}\right) + \frac{2\sigma_0^2 \sigma_1^2}{\sigma_0^2 - \sigma_1^2} \log(r)}.$$

If $\sigma_0 < \sigma_1$, then $B$ is the complement of a $N$-dimensional ball with the center at $S_<$ and radius $R_<$:

$$S_< = x - \frac{\sigma_0^2}{\sigma_1^2 - \sigma_0^2} \delta, \ R_< = \sqrt{\frac{2\sigma_0^4 - \sigma_0^2 \sigma_1^2}{(\sigma_1^2 - \sigma_0^2)^2} \|\delta\|^2 + 2N \frac{\sigma_0^2 \sigma_1^2}{\sigma_1^2 - \sigma_0^2} \log\left(\frac{\sigma_1}{\sigma_0}\right) - \frac{2\sigma_0^2 \sigma_1^2}{\sigma_1^2 - \sigma_0^2} \log(r)}.$$

*Proof.* From spherical symmetry of isotropic multivariate normal distribution, it follows, that without loss of generality we can take $\delta \equiv (a, 0, \ldots, 0)$. With little abuse of notation, let $a$ refer to $(a, 0, \ldots, 0)$ as well as $\|(a, 0, \ldots, 0)\|$. With this, the $B$ is a set of all $x$, for which:

$$\frac{f_1(x)}{f_0(x)} \geq \frac{1}{r} \iff$$

$$\frac{1}{(2\pi)^{N/2}\sigma_0^N} \exp\left(-\frac{1}{2\sigma_0^2}\sum_{i=1}^{N} x_i^2\right) \leq \frac{r}{(2\pi)^{N/2}\sigma_1^N} \exp\left(-\frac{1}{2\sigma_1^2}\left[(x_1-a)^2 + \sum_{i=2}^{N} x_i^2\right]\right) \iff$$

$$\frac{1}{2\sigma_1^2}\left((x_1-a)^2 + \sum_{i=2}^{N} x_i^2\right) - \frac{1}{2\sigma_0^2}\sum_{i=1}^{N} x_i^2 \leq N\log\left(\frac{\sigma_0}{\sigma_1}\right) + \log(r) \iff$$

$$(\sigma_0^2 - \sigma_1^2)\sum_{i=2}^{N} x_i^2 + (\sigma_0^2 - \sigma_1^2)x_1^2 - 2\sigma_0^2 x_1 a + \sigma_0^2 a^2 \leq 2N\sigma_0^2\sigma_1^2\log\left(\frac{\sigma_0}{\sigma_1}\right) + 2\sigma_0^2\sigma_1^2\log(r)$$

Now assume $\sigma_0 > \sigma_1$ and continue:

$$(\sigma_0^2 - \sigma_1^2)\sum_{i=2}^{N} x_i^2 + (\sigma_0^2 - \sigma_1^2)x_1^2 - 2\sigma_0^2 x_1 a + \sigma_0^2 a^2 \leq 2N\sigma_0^2\sigma_1^2\log\left(\frac{\sigma_0}{\sigma_1}\right) + 2\sigma_0^2\sigma_1^2\log(r) \iff$$

$$\sum_{i=2}^{N} x_i^2 + x_1^2 - \frac{2\sigma_0^2}{\sigma_0^2 - \sigma_1^2}ax_1 + \frac{a^2\sigma_0^2}{\sigma_0^2 - \sigma_1^2} \leq 2N\frac{\sigma_0^2\sigma_1^2}{\sigma_0^2 - \sigma_1^2}\log\left(\frac{\sigma_0}{\sigma_1}\right) + \frac{2\sigma_0^2\sigma_1^2}{\sigma_0^2 - \sigma_1^2}\log(r) \iff$$

$$\left(x_1 - \frac{\sigma_0^2}{\sigma_0^2 - \sigma_1^2}a\right)^2 + \sum_{i=2}^{N} x_i^2 \leq \frac{\sigma_0^2\sigma_1^2}{(\sigma_0^2 - \sigma_1^2)^2}a^2 + 2N\frac{\sigma_0^2\sigma_1^2}{\sigma_0^2 - \sigma_1^2}\log\left(\frac{\sigma_0}{\sigma_1}\right) + \frac{2\sigma_0^2\sigma_1^2}{\sigma_0^2 - \sigma_1^2}\log(r)$$

Such inequality defines exactly the ball from the statement of the theorem. On the other hand, if $\sigma_0 < \sigma_1$:

$$(\sigma_1^2 - \sigma_0^2)\sum_{i=2}^{N} x_i^2 + (\sigma_1^2 - \sigma_0^2)x_1^2 + 2\sigma_0^2 x_1 a - \sigma_0^2 a^2 \geq 2N\sigma_0^2\sigma_1^2\log\left(\frac{\sigma_1}{\sigma_0}\right) - 2\sigma_0^2\sigma_1^2\log(r) \iff$$

$$\sum_{i=2}^{N} x_i^2 + x_1^2 + \frac{2\sigma_0^2}{\sigma_1^2 - \sigma_0^2}ax_1 - \frac{a^2\sigma_0^2}{\sigma_1^2 - \sigma_0^2} \geq 2N\frac{\sigma_0^2\sigma_1^2}{\sigma_1^2 - \sigma_0^2}\log\left(\frac{\sigma_1}{\sigma_0}\right) - \frac{2\sigma_0^2\sigma_1^2}{\sigma_1^2 - \sigma_0^2}\log(r) \iff$$

$$\left(x_1 + \frac{\sigma_0^2}{\sigma_1^2 - \sigma_0^2}a\right)^2 + \sum_{i=2}^{N} x_i^2 \geq \frac{2\sigma_0^4 - \sigma_0^2\sigma_1^2}{(\sigma_1^2 - \sigma_0^2)^2}a^2 + 2N\frac{\sigma_0^2\sigma_1^2}{\sigma_1^2 - \sigma_0^2}\log\left(\frac{\sigma_1}{\sigma_0}\right) - \frac{2\sigma_0^2\sigma_1^2}{\sigma_1^2 - \sigma_0^2}\log(r)$$

This is exactly the complement of a ball from the second part of the statement of the theorem. $\square$

**Theorem 3:**

$$\mathbb{P}_0(B) = \chi_N^2\left(\frac{\sigma_0^2}{(\sigma_0^2 - \sigma_1^2)^2}\|\delta\|^2, \frac{R_{<,>}^2}{\sigma_0^2}\right), \mathbb{P}_1(B) = \chi_N^2\left(\frac{\sigma_1^2}{(\sigma_0^2 - \sigma_1^2)^2}\|\delta\|^2, \frac{R_{<,>}^2}{\sigma_1^2}\right),$$

where the sign $<$ or $>$ is choosed according to the inequality between $\sigma_0$ and $\sigma_1$.

*Proof.* Assume first $\sigma_0 > \sigma_1$. Let us shift the coordinates, such that $x + \frac{\sigma_0^2}{\sigma_0^2 - \sigma_1^2}\delta \leftarrow 0$. Now, the $x$ will have coordinates $-\frac{\sigma_0^2}{\sigma_0^2 - \sigma_1^2}\delta$. Assume $X \sim \mathcal{N}(-\frac{\sigma_0^2}{\sigma_0^2 - \sigma_1^2}\delta, \sigma_0^2 I)$. To obtain

$$\mathbb{P}_0(B) = \mathbb{P}(X \in B) = \mathbb{P}\left(\|X\|^2 < \frac{\sigma_0^2\sigma_1^2}{(\sigma_0^2 - \sigma_1^2)^2}\|\delta\|^2 + 2N\frac{\sigma_0^2\sigma_1^2}{\sigma_0^2 - \sigma_1^2}\log\left(\frac{\sigma_0}{\sigma_1}\right) + \frac{2\sigma_0^2\sigma_1^2}{\sigma_0^2 - \sigma_1^2}\log(r)\right),$$

we could almost use NCCHSQ, but we don't have correct scaling of variance of $X$. However, for any regular square matrix $Q$, it follows that $\mathbb{P}(X \in B) = \mathbb{P}(QX \in QB)$, where $QB$ is interpreted as set projection. Therefore, if we choose $Q \equiv \frac{1}{\sigma_0} I$, we will get

$$\mathbb{P}_0(B) = \mathbb{P}\left(\|X/\sigma_0\|^2 < \frac{\sigma_1^2}{(\sigma_0^2 - \sigma_1^2)^2}\|\delta\|^2 + 2N\frac{\sigma_1^2}{\sigma_0^2 - \sigma_1^2}\log\left(\frac{\sigma_0}{\sigma_1}\right) + \frac{2\sigma_1^2}{\sigma_0^2 - \sigma_1^2}\log(r)\right).$$

Now, since $X/\sigma_0 \sim \mathcal{N}(-\frac{\sigma_0}{\sigma_0^2 - \sigma_1^2}\delta, I)$, we can use the definition of NCCHSQ to obtain the final:

$$\mathbb{P}_0(B) = \chi_N^2\left(\frac{\sigma_0^2}{(\sigma_0^2 - \sigma_1^2)^2}\|\delta\|^2, \frac{\sigma_1^2}{(\sigma_0^2 - \sigma_1^2)^2}\|\delta\|^2 + 2N\frac{\sigma_1^2}{\sigma_0^2 - \sigma_1^2}\log\left(\frac{\sigma_0}{\sigma_1}\right) + \frac{2\sigma_1^2}{\sigma_0^2 - \sigma_1^2}\log(r)\right).$$

To obtain $\mathbb{P}_1(B)$, we will do similar calculation, yet we need to compute the offset:

$$x + \frac{\sigma_0^2}{\sigma_0^2 - \sigma_1^2}\delta - x - \delta = \frac{\sigma_1^2}{\sigma_0^2 - \sigma_2^2}a.$$

Thus, after shifting coordinates in the same way, our alternative $X$ will be distributed like $X \sim \mathcal{N}(-\frac{\sigma_1^2}{\sigma_0^2 - \sigma_2^2}a, \sigma_1 I)$. Now, the same idea as before will yield the required formula.

In the case of $\sigma_0 < \sigma_1$, we do practically the same thing, yet now, we have to keep in mind, that $B$ will not be a ball, but its complement, therefore we will obtain "$1-$" in the formulas. $\qquad\square$

**Lemma 18.** Functions $\xi_>(a), \xi_<(a)$ are continuous on the whole $\mathbb{R}_+$. Particularly, they are continuous at 0.

*Proof.* Assume for simplicity $\sigma_0 > \sigma_1$ and fix $x_0$, whose position is irrelevant and fix $x_1$ such that $\|x_0 - x_1\| = a_m$, where $a_m$ is the point, where we prove the continuity. Note, that $\chi_N^2(\lambda, x)$ can be interpreted (as we have seen) as a probability of an offset ball with radius $\sqrt{x}$ and offset $\sqrt{\lambda}$. Assume we have a sequence $\{a_i\}_{i=1}^\infty, a_i \to a_m$. Define $x_i$ to be a point lying on the line defined by $x_0, x_1$ s.t. $\|x_0 - x_i\| = a_i$. define $B_{a_i}$ to be the worst-case ball corresponding to $a_i, x_i$. Now, without loss of generality, we can assume, that all $B$'s are open. We have already seen from Lemma 2, that centers of $B_i$ converge to the center of $B_m$. Define $X_i = \mathbf{1}(B_i), X_m = \mathbf{1}(B_m)$.

First we need to prove, that $r_i$, radiuses of the balls converge. Assume for contradiction, that $r_i$ do not converge. Without loss of gererality, let $r_s = \limsup\limits_{i \to \infty} r_i > r_m$. If $r_s = \infty$, it is trivial to see, that $P_0(B_i) \neq p_B$ for some $i$ for which $r_i$ is too big. If $r_s < \infty$, consider $\{a_{i_k}\}_{k=0}^\infty$ to be the subsequence for which $r_s$ is monotonically attained. Define $B_s$ the ball with center $x_1$ and radius $r_s$ and $X_s = \mathbf{1}(B_s)$. Then, $X_{i_k} \overset{a.s.}{\to} X_s$ for $k \to \infty$ and from dominated convergence theorem, $\mathbb{P}_0(B_{i_k}) \to \mathbb{P}_0(B_s)$. However, $\mathbb{P}_0(B_s) > \mathbb{P}_0(B_m) = p_B$, what is contradiction, since obviously $\mathbb{P}_0(B_{i_k}) \neq p_B$ for some $k$.

Since $\sigma_1$ is fixed, the $\mathbb{P}_{1_i}$, probability measures corresponding to $\mathcal{N}(x_i, \sigma_1 I)$ are actually the same probability measure up to a shift. Therefore, $\mathbb{P}_{1_i}(B_i)$ can be treated as $\mathbb{P}_2(\bar{B}_i)$, where $\mathbb{P}_2$ is simply measure corresponding to $\mathcal{N}(0, \sigma_1 I)$ and $\bar{B}_i$ is simply $B_i$ shifted accordingly s.t. $\mathbb{P}_{1_i}(B_i) = \mathbb{P}_2(\bar{B}_i)$ (and assume, without loss of generality, that for each $i$, they are shifted such that their centers lie on a fixed line). Now, since we know, that "position" of both the centers of the balls and the $x_1$ is continuous w.r.t $a$, as can be seen from Lemma 2 (and the radiuses are still $r_i$ and converge), we see, that even $\mathbf{1}(\bar{B}_i) \to \mathbf{1}(\bar{B}_m)$ almost surely. Now, we can simply use dominated convergence theorem using $\mathbb{P}_2$ to obtain $\mathbb{P}_2(\bar{B}_i) \to \mathbb{P}_2(\bar{B}_m)$ and thus $\mathbb{P}_{1_i}(B_i) \to \mathbb{P}_{1_m}(B_m)$, what we wanted to prove.

Note that the proof for the case $\sigma_0 < \sigma_1$ is fully analogous, yet instead of class $B$ probabilities, we work with class $A$ probabilities to still work with balls and not with less convenient complements. $\qquad\square$

**Lemma 19.** If $\lambda_1 > \lambda_2$, then $\chi_N^2(\lambda_1^2, x^2) \leq \chi_N^2(\lambda_2^2, x^2)$.

*Proof.* Let us fix $\mathcal{N}(0, I)$ and respective measure $\mathbb{P}$ and respective density $f$. From symmetry, the NCCHCSQ defined as distribution of $\|X\|^2$ for an offset normal distribution can be as well defined as $\|X - s\|^2$ under centralized normal distribution. Define $B_1$ a ball with center at $(\lambda_1, 0, \dots, 0)$ and radius $x$ and $B_2$ a ball with center at $(\lambda_2, 0, \dots, 0)$ and radius $x$. Denote $C(B)$ as the center

of a ball $B$. From definition of NCCHSQ it now follows, that $\mathbb{P}(B_i) = \chi_N^2(\lambda_i^2, x^2), i \in \{1, 2\}$. Therefore, it suffices to show $\mathbb{P}(B_1) \leq \mathbb{P}(B_2)$.

Define $D_1 = B_1 \backslash B_2$ and $D_2 = B_2 \backslash B_1$. Then we know:

$$\mathbb{P}(B_1) = \int_{B_1} f(z)dz = \int_{B_1 \cap B_2} f(z)dz + \int_{B_1 \backslash B_2} f(z)dz = \int_{B_1 \cap B_2} f(z)dz + \int_{D_1} f(z)dz$$

$$\mathbb{P}(B_2) = \int_{B_2} f(z)dz = \int_{B_2 \cap B_1} f(z)dz + \int_{B_2 \backslash B_1} f(z)dz = \int_{B_2 \cap B_1} f(z)dz + \int_{D_2} f(z)dz.$$

Thus,

$$\mathbb{P}(B_1) \leq \mathbb{P}(B_2) \iff \int_{D_1} f(z)dz \leq \int_{D_2} f(z)dz.$$

Let $S = \frac{C(B_1) + C(B_2)}{2}$. Define a central symmetry $M$ with center $S$. Let $z_1 \in D_1$. Then $z_1$ can be decomposed as $z_1 = C(B_1) + d, \|d\| \leq x$. Then, $z_2 := M(z_1) = C(B_2) - d$ from symmetry. This way, we see, that $D_1 = M(D_2)$ and $D_2 = M(D_1)$ under a bijection $M$ which does not distort the geometry and distances of the euclidean space. Therefore, it suffices to show:

$$\forall z \in D_2 : f(z) \geq f(M(z)), M(z) \in D_1.$$

From the monotonicity of $f(y)$ w.r.t $\|y\|$ it actually suffices to show $\|z\| \leq \|M(z)\| \ \forall z \in D_2$. Fix some $z \in D_2$. By the fact that $M$ is central symmetry and $z \to M(z)$, it is obvious, that $z = S + p$, $M(z) = S - p$, where $p$ is some vector. Now, using law of cosine, we can write:

$$\|z\|^2 = \|S\|^2 + \|p\|^2 - 2\|S\|\|p\|\cos(\alpha),$$

where $\alpha$ is angle between $S$ and $-p$. On the other hand:

$$\|M(z)\|^2 = \|S\|^2 + \|-p\|^2 - 2\|S\|\|-p\|\cos(\pi - \alpha).$$

It is obvious from these equations, that

$$\|z\| \leq \|M(z)\| \iff \alpha \leq \pi/2 \iff p^T S \leq 0.$$

Here, the crucial observation is, that $D_1$ and $D_2$ are separated by a hyperplane perpendicular to $S$ (vector), such that $S$ (point) is in this hyperplane. From this it follows:

$$y \in D_2 \implies y^T S \leq \|S\|^2, \ \ y \in D_1 \implies y^T S \geq \|S\|^2.$$

Now, since $z = S + p$ and $z \in D_2$, this implies $\|S\|^2 \geq z^T S = \|S\|^2 + p^T S$ and thus $p^T S \leq 0$. $\square$

**Lemma 20.** Functions $\xi_>(a), \xi_<(a)$ are non-decreasing in $a$.

*Proof.* First assume $\sigma_0 > \sigma_1$ and analyse $\xi_>(a)$. From Lemma 18, we can without loss of generality assume, that $a > 0$, since $\xi_>(0)$ is simply the limit for $a \to 0$ and cannot change the monotonicity status.

Now, fix $a > 0$ and define $x_0, x_1$ s.t. $\|x_0 - x_1\| = a$. Denote $B_a$ to be the worst-case ball corresponding to $x_0, x_1$ and $\mathbb{P}_1$ as usual. Choose $\epsilon < \frac{\sigma_1^2}{\sigma_0^2 - \sigma_1^2} a$, which is the distance between $x_1$ and $C(B_a)$, as can be seen from Lemma 2.

Now, assume $x_2$ lies on line defined by $x_0, x_1$ with $\|x_0 - x_2\| = a + \epsilon$. Let $B_{a+\epsilon}$ be the corresponding worst-case ball and $\mathbb{P}_2$ as usual. First observe, that $\mathbb{P}_2(B_a) \geq \mathbb{P}_1(B_a)$, since $\|x_1 - C(B_a)\| > \|x_2 - C(B_a)\|$. Here we use $\chi_N^2(\lambda_1, x) \leq \chi_N^2(\lambda_2, x)$ if $\lambda_1 > \lambda_2$, what is proved in Lemma 19. Second, note, that since $B_{a+\epsilon}$ is the worst-case ball for $x_2$, it follows $\mathbb{P}_2(B_{a+\epsilon}) \geq \mathbb{P}_2(B_a)$. Thus, $\mathbb{P}_1(B_a) \leq \mathbb{P}_2(B_{a+\epsilon})$, but that is exactly $\xi_>(a) \leq \xi_>(a + \epsilon)$.

To prove $\xi_>(a) \leq \xi_>(a + \epsilon)$ also for $\epsilon > \frac{\sigma_1^2}{\sigma_0^2 - \sigma_1^2} a$, it suffices to consider finite sequence of points $a_i$ starting at $a$ and ending at $a + \epsilon$ that are "close enough to each other" such that the respective $\epsilon_i$ that codes the shift $a_i \to a_{i+1}$ satisfy $\epsilon_i < \frac{\sigma_1^2}{\sigma_0^2 - \sigma_1^2} a_i$.

Now, assume $\sigma_0 < \sigma_1$ and analysie $\xi_<(a)$. The proof is similar, but we have to be a bit careful about some details. Again, fix $a > 0$, define $x_0, x_1$ accordingly and all other objects as before, except now, let us denote $A_a = B_a^C$ and $A_{a+\epsilon} = B_a^C$ to be the class $A$ balls which are complements to the anti-balls $B$. Again, $\mathbb{P}_2(A_a) \leq \mathbb{P}_1(A_a)$, since $\|x_1 - C(A_a)\| < \|x_2 - C(A_a)\|$. We again used the monotonicity from Lemma 19. Moreover, $\mathbb{P}_2(A_{a+\epsilon}) \leq \mathbb{P}_2(A_a)$, since $B_{a+\epsilon}$ is the worst-case set for $x_2$. Therefore, $\mathbb{P}_1(A_a) \geq \mathbb{P}_2(A_{a+\epsilon})$, but after reverting to $B$'s, it follows $\xi_>(a) \leq \xi_>(a + \epsilon)$.

Here, we don't even need to care about $\|\epsilon\|$, since the centers of $A$'s are on the opposite half-lines from $x_0$ than $x_1$ and $x_2$. $\qquad \square$

To prove the main theorem, we need a simple bound on a median of central chi-squared distribution, shown in Robert (1990) in a more general way.

**Lemma 21.** For all $c \geq 0$,

$$N - 1 + c \leq \chi^2_{N,qf}(c, 0.5) \leq \chi^2_{N,qf}(0.5) + c.$$

*Proof.* See Robert (1990). $\qquad \square$

**Theorem 4 (the curse of dimensionality):** Let $x_0, x_1, p_A, \sigma_0, \sigma_1, N$ be as usual. Then, the following two implications hold:

1. If $\sigma_0 > \sigma_1$ and

$$\log\left(\frac{\sigma_1^2}{\sigma_0^2}\right) + 1 - \frac{\sigma_1^2}{\sigma_0^2} < \frac{2\log(1 - p_A)}{N},$$

then $x_1$ is not certified w.r.t. $x_0$.

2. If $\sigma_0 < \sigma_1$ and

$$\log\left(\frac{\sigma_1^2}{\sigma_0^2}\frac{N-1}{N}\right) + 1 - \frac{\sigma_1^2}{\sigma_0^2}\frac{N-1}{N} < \frac{2\log(1 - p_A)}{N},$$

then $x_1$ is not certified w.r.t. $x_0$.

*Proof.* We will first prove first statement, thus let us assume $\sigma_0 > \sigma_1$. Then $\mathbb{P}_1(B) = \xi_>(\|x_0 - x_1\|)$. From monotonicity of $\xi$ showed in Lemma 20, we know $\xi_>(\|x_0 - x_1\|) \geq \xi_>(0)$. We will show $\xi_>(0) > 0.5$. We have, using definition of $\xi_>$ plugging in $a = 0$:

$$\xi_>(0) = \chi^2_N\left(\frac{\sigma_0^2}{\sigma_1^2}\chi^2_{N,qf}(1 - p_A)\right).$$

Note, that here, we work with central chi-square cdf and quantile function. In order to show $\xi_>(0) > 0.5$, it suffices to show

$$\frac{\sigma_0^2}{\sigma_1^2}\chi^2_{N,qf}(1 - p_A) \geq N,$$

because it is well-known, that median of central chi-square distribution is smaller than mean, which is $N$, i.e. from strict monotonicity of cdf, we will get $\chi^2_N(N) > 0.5$. To show the above inequality, we will use Chernoff bound on chi-squared, which states the following: If $0 < z < 1$, then $\chi^2_N(zN) \leq (z\exp(1-z))^{N/2}$. Putting $z \equiv \frac{\sigma_1^2}{\sigma_0^2}$, using chernoff bound we get:

$$\chi^2_N\left(\frac{\sigma_1^2}{\sigma_0^2}N\right) \leq \left[\frac{\sigma_1^2}{\sigma_0^2}\exp\left(1 - \frac{\sigma_1^2}{\sigma_0^2}\right)\right]^{\frac{N}{2}} \overset{!}{<} 1 - p_A.$$

The last inequality is required to hold. If it holds, then necessarily $\chi^2_{N,qf}(1 - p_A) > \frac{\sigma_1^2}{\sigma_0^2}N$ and thus $\frac{\sigma_0^2}{\sigma_1^2}\chi^2_{N,qf}(1 - p_A) > N$. Manipulating the required inequality, we will get exactly

$$\log\left(\frac{\sigma_1^2}{\sigma_0^2}\right) + 1 - \frac{\sigma_1^2}{\sigma_0^2} < \frac{2\log(1 - p_A)}{N},$$

what is the assumption of 1.

Similarly we will also prove the statement 2. Assume $\sigma_0 > \sigma_1$. Like in part 1, we will just prove $\xi_<(0) > 0.5$ by using chernoff bound. This time, however, we have:

$$\xi_<(0) = 1 - \chi_N^2 \left( \frac{\sigma_0^2}{\sigma_1^2} \chi_{N,qf}^2(p_A) \right),$$

i.e. we need to prove

$$\chi_N^2 \left( \frac{\sigma_0^2}{\sigma_1^2} \chi_{N,qf}^2(p_A) \right) < \frac{1}{2}.$$

The second part of Chernoff bound states: If $1 < z$, then $\chi_N^2(zN) \geq 1 - (z \exp(1-z))^{N/2}$. Let us choose $z \equiv \frac{\sigma_1^2}{\sigma_0^2} \frac{N-1}{N}$. Then Chernoff bound yields:

$$\chi_N^2 \left( \frac{\sigma_1^2}{\sigma_0^2} \frac{N-1}{N} N \right) \geq 1 - \left[ \frac{\sigma_1^2}{\sigma_0^2} \frac{N-1}{N} \exp \left( 1 - \frac{\sigma_1^2}{\sigma_0^2} \frac{N-1}{N} \right) \right] \overset{!}{>} p_A.$$

If this holds, then

$$\frac{\sigma_1^2}{\sigma_0^2} \frac{N-1}{N} N > \chi_{N,qf}^2(p_A) \iff \frac{\sigma_0^2}{\sigma_1^2} \chi_{N,qf}^2(p_A) < N - 1.$$

Now, using Lemma 21 for the easy case of central chi-squared, we see: $\chi_N^2(N-1) < 0.5$ and thus

$$\chi_N^2 \left( \frac{\sigma_0^2}{\sigma_1^2} \chi_{N,qf}^2(p_A) \right) < \frac{1}{2},$$

what we wanted to prove. $\qquad \square$

**Corollary 5 (one-sided simpler bound):** Let $x_0, x_1, p_A, \sigma_0, \sigma_1, N$ be as usual and assume now $\sigma_0 > \sigma_1$. Then, if

$$\frac{\sigma_1}{\sigma_0} < \sqrt{1 - 2\sqrt{\frac{-\log(1-p_A)}{N}}},$$

then $x_1$ is not certified w.r.t $x_0$.

*Proof.* We will simply prove

$$\frac{\sigma_1}{\sigma_0} < \sqrt{1 - 2\sqrt{\frac{-\log(1-p_A)}{N}}} \implies \log \left( \frac{\sigma_1^2}{\sigma_0^2} \right) + 1 - \frac{\sigma_1^2}{\sigma_0^2} < \frac{2\log(1-p_A)}{N}.$$

Assume expression $\log(1-y)+y; 1 > y > 0$. From Taylor series, it is apparent, that $\log(1-y)+y < -\frac{y^2}{2}$. Therefore, if $-\frac{y^2}{2} < \frac{2\log(1-p_A)}{N}$, then also $\log(1-y)+y < \frac{2\log(1-p_A)}{N}$. Solving for $y$ in the first inequality, we get sufficient condition $y > 2\sqrt{\frac{-\log(1-p_A)}{N}}$. Plugging $1 - \frac{\sigma_1^2}{\sigma_0^2}$ into $y$ we get:

$$1 - \frac{\sigma_1^2}{\sigma_0^2} > 2\sqrt{\frac{-\log(1-p_A)}{N}},$$

which is very easily manipulated to the inequality from theorem statement. $\qquad \square$

**Theorem 6:** Let $x_0, x_1, p_A, \sigma_0$ be as usual and let $\|x_0 - x_1\| = R$. Then, the following two statements hold:

1. Let $\sigma_1 \leq \sigma_0$. Then, for all $\sigma_2 : \sigma_1 \leq \sigma_2 \leq \sigma_0$, if $\xi_>(R, \sigma_2) > 0.5$, then $\xi_>(R, \sigma_1) > 0.5$.

2. Let $\sigma_1 \geq \sigma_0$. Then, for all $\sigma_2 : \sigma_1 \geq \sigma_2 \geq \sigma_0$, if $\xi_<(R, \sigma_2) > 0.5$, then $\xi_>(R, \sigma_1) > 0.5$.

*Proof.* We will first prove the first statement. Denote, as usual in the proofs $B_i$ the worst-case ball for $\sigma_i$, $\mathbb{P}_i$ the probability associated to $\mathcal{N}(x_1, \sigma_i^2 I)$. Since $\xi_>(R, \sigma_2) > 0.5$ and since it is essentially $\mathbb{P}_2(B_2)$, we see, that the probability of a ball under normal distribution is bigger than half. This is obviously possible just if $x_1 \in B_2$. From the fact, that $B_1$ is the worst-case ball for $\sigma_1$ we see $\xi_>(R, \sigma_1) = \mathbb{P}_1(B_1) \geq \mathbb{P}_1(B_2)$. It suffices to show $\mathbb{P}_1(B_2) \geq \mathbb{P}_2(B_2)$.

This follows, since $\sigma_1 \leq \sigma_2$ and $x_1 \in B_2$. We know, that we can rescale the space such that

$$\mathbb{P}_1(B_2) = \mathbb{P}_2\left(\frac{\sigma_2}{\sigma_1}(B_2 - x_1) + x_1\right),$$

using the fact that $\sigma$ just scales the normal distribution. The set $\frac{\sigma_2}{\sigma_1}(B_2 - x_1) + x_1$ is just an image of $B_2$ via homothety with center $x_1$ and rate $\frac{\sigma_2}{\sigma_1}$. So it suffices to prove

$$\mathbb{P}_2\left(\frac{\sigma_2}{\sigma_1}(B_2 - x_1) + x_1\right) \geq \mathbb{P}_2(B_2).$$

However, obviously $\frac{\sigma_2}{\sigma_1}(B_2 - x_1) + x_1 \supset B_2$ from convexity of a ball. If, namely, $x_1 + z \in B_2$, then from convexity also $x_1 + \frac{\sigma_1}{\sigma_2}z \in B_2$ and this maps back to $x_1 + z$, thus $x_1 + z$ is in an image. Applying monotonicity of $\mathbb{P}$, we obtain the result.

Now we will prove the second statement and as usual, let $A_1, A_2, \mathbb{P}_1, \mathbb{P}_2$ be as usual ($A$ is now the ball connected to class $A$). As always, $\mathbb{P}_1(A_1) \leq \mathbb{P}_1(A_2)$, so it suffices to show $\mathbb{P}_2(A_2) < 0.5 \implies \mathbb{P}_1(A_2) < 0.5$. Now, we need to distinguish two cases. If $x_1 \in A_2$, the proof is completely analogical to the first part, but now reasoning on $A$'s rather than $B$'s. In this case, we will even get stronger $\mathbb{P}_1(A_2) \leq \mathbb{P}_2(A_2)$ just like in the first part. If $x_1 \notin A_2$, then it is easy to see, that indeed $\mathbb{P}_2(A_2) < 0.5$, yet it is also obvious to see that $\mathbb{P}_1(A_2) < 0.5$. This finishes the proof of the lemma. $\square$

**Theorem 7:** Let $\sigma(x)$ be $r$-semi-elastic function and $x_0, p_A, N, \sigma_0$ as usual. Then, the certified radius at $x_0$ guaranteed by our method is

$$CR(x_0) = \max\left\{0, \sup\left\{R \geq 0; \xi_>(R, \sigma_0 \exp(-rR)) < 0.5 \text{ and } \xi_<(R, \sigma_0 \exp(rR)) < 0.5\right\}\right\}.$$

*Proof.* This follows easily from Theorem 6 $\square$

**Lemma 22.** Let us have $f(x) = \sum_{i=1}^{M} f_i(x)\mathbf{1}(x \in R_i)$, where $\{R_i\}_{i=1}^{M}$ is finite set of regions that divide the $\mathbb{R}^N$ and $\{f_i\}_{i=1}^{M}, f_i : R_i \to \mathbb{R}$ is finite set of 1-Lipschitz continuous (1-LC) functions. Moreover assume, that $f(x)$ is continuous. Then, $f(x)$ is 1-LC.

*Proof.* We do not assume any nice behaviour from our decision regions, what can make the situation quite ugly. For instance, regions might not be measurable. However, it will not be a problem for us.

Fix $x_1, x_2$. Consider line segment $S = x_1 + \alpha(x_2 - x_1), \alpha \in [0, 1]$. Let us instead of points in $S$ work with numbers in $[0, 1]$ via the $\alpha$ encoding. Consider the following coloring $C$ of $[0, 1]$: Each point $a \in [0, 1]$ will be assigned one of $M$ colors according to which region the $x_1 + a(x_2 - x_1)$ belongs. Define $d_1 \equiv 0$ and $d_2 = \sup\{z \in [0, 1], C(z) = C(d_1)\}$. Thus, $d_2$ is the supremum of all numbers colored the same color as 0. Then,

$$|f(x_1 + d_2(x_2 - x_1)) - f(x_1 + d_1(x_2 - x_1))| \leq (d_2 - d_1)\|x_2 - x_1\|.$$

Why? Let $\{z_j\}_{j=1}^{\infty}$ be a non-decreasing sequence s.t. $C(z_j) = C(0)$ and $z_j \to d_2$. Since $f$ is continuous, obviously $f(x_1 + d_2(x_2 - x_1)) = \lim_{j \to \infty} f(x_1 + z_j(x_2 - x_1))$. Now, since norm and absolute value are both continuous functions and since from 1-LC of $f_{C(0)}$ on $R_{C(0)}$ we have

$$\forall j \in \mathbb{N} : \frac{|f(x_1 + z_j(x_2 - x_1)) - f(x_1 + d_1(x_2 - x_1))|}{(z_j - d_1)\|x_2 - x_1\|} \leq 1,$$

we also necessarily have

$$\frac{|f(x_1 + d_2(x_2 - x_1)) - f(x_1 + d_1(x_2 - x_1))|}{(d_2 - d_1)\|x_2 - x_1\|} \leq 1.$$

If $d_2 = 1$, we finish the construction. If not, distinguish two cases. First assume $C(d_2) = C(d_1)$. In this case, take some color $C$ s.t.

$$\exists \{z_j\}_{j=1}^{\infty} : z_{j+1} \leq z_j \ \forall j \in \mathbb{N} \ \text{ and } \ z_j \to d_2 \ \text{ and } \ z_j > d_2 \ \forall j \in \mathbb{N},$$

and fix one such $\{z_j\}_{j=1}^{\infty}$. Obviously, $C \neq C(0)$, since $d_2$ is upper-bound on points of color $C(0)$. Then, define $d_3 = \sup\{z \in [0,1], C(z) = C\}$ and also define $\{\overline{z_j}\}_{j=1}^{\infty} : \overline{z_{j+1}} \geq \overline{z_j} \ \forall j \in \mathbb{N}$ and $\overline{z_j} \to d_3$. From continuity of $f$, we again have $f(x_1 + d_2(x_2 - x_1)) = \lim_{j \to \infty} f(x_1 + z_j(x_2 - x_1))$ and similarly $f(x_1 + d_3(x_2 - x_1)) = \lim_{j \to \infty} f(x_1 + \overline{z_j}(x_2 - x_1))$. Again from continuity of absolute value and norm and 1-LC of all the partial functions we have:

$$\forall j \in \mathbb{N} : \frac{|f(x_1 + \overline{z_j}(x_2 - x_1)) - f(x_1 + z_j(x_2 - x_1))|}{(\overline{z_j} - z_j)\|x_2 - x_1\|} \leq 1$$

and

$$\frac{|f(x_1 + d_3(x_2 - x_1)) - f(x_1 + d_2(x_2 - x_1))|}{(d_3 - d_2)\|x_2 - x_1\|} \leq 1.$$

Now assume $C(d_2) \neq C(d_1)$. Then, we can take as $C$ directly $C(d_2)$ and do the same as in the last paragraph (note, that this case could have implicitly come up in the previous construction too, but we would need to not take $C = C(0)$ and we find this case distinction to be more elegant).

If $d_3 = 1$, we finish the construction. If not, we continue in exactly the same manner as before. Since the number of colors $M$ is finite, we will run out of colors in finite number of steps and thus, eventually there will be $l \leq M$ s.t. $d_l = 1$. The final 1-LC is now trivially obtained as follows:

$$|f(x_1 + 1(x_2 - x_1)) - f(x_1 + 0(x_2 - x_1))| = \left| \sum_{i=1}^{l-1} f(x_1 + d_{i+1}(x_2 - x_1)) - f(x_1 + d_i(x_2 - x_1)) \right|$$

$$\leq \sum_{i=1}^{l-1} \left| f(x_1 + d_{i+1}(x_2 - x_1)) - f(x_1 + d_i(x_2 - x_1)) \right| \leq \sum_{i=1}^{l-1} \left| (d_{i+1} - d_i)\|x_2 - x_1\| \right|$$

$$= \|x_2 - x_1\| \sum_{i=1}^{l-1} (d_{i+1} - d_i) = \|x_2 - x_1\|$$

$\square$

**Theorem 8:** The $\sigma(x)$ defined in Equation 1 is $r$-semi-elastic.

*Proof.* Our aim is to prove, that

$$\log(\sigma(x)) = \log(\sigma_b) + r \left( \frac{1}{k} \left( \sum_{x_i \in \mathcal{N}_k(x)} \|x - x_i\| \right) - m \right)$$

is $r$ lipschitz continuous. Obviously, this does not depend neither on $\log(\sigma_b)$, nor on $-rm$, so we will focus just on $\frac{r}{k} \sum_{x_i \in \mathcal{N}_k(x)} \|x - x_i\|$. Obviously, this function is $r$ lipschitz continuous if and only if $\frac{1}{k} \sum_{x_i \in \mathcal{N}_k(x)} \|x - x_i\|$ is 1-LC.

Let us fix $y \in \mathbb{R}^N$. We will first prove $\|x - y\|$ is 1-LC. Let us fix $x_1, x_2$. From triangle inequality we have

$$\big| \|x_1 - y\| - \|x_2 - y\| \big| \leq \|x_1 - x_2\|,$$

what is exactly what we wanted to prove.

Now fix $y_1, y_2, \ldots, y_k$ and $x_1, x_2$. Then

$$\left| \frac{1}{k} \sum_{i=1}^{k} \|x_1 - y_i\| - \frac{1}{k} \sum_{i=1}^{k} \|x_2 - y_i\| \right| = \frac{1}{k} \left| \sum_{i=1}^{k} \|x_1 - y_i\| - \|x_2 - y_i\| \right|$$

$$\leq \frac{1}{k} \sum_{i=1}^{k} \left| \|x_1 - y_i\| - \|x_2 - y_i\| \right| \leq \frac{1}{k} \sum_{i=1}^{k} \|x_1 - x_2\| = 1$$

Finally note, that using the $k$ nearest neighbors out of finite training dataset will divide $\mathbb{R}^N$ in a finite number of regions, where each region is defined by the set of $k$ nearest neighbors for $x$ in that region. Note, that the average distance from $k$ nearest neighbors is obviously continuous. Then, using Lemma 22, the claim follows. $\square$

