# OpenReview forum: "Intriguing Properties of Input-dependent Randomized Smoothing"
_ICLR.cc/2022/Conference — ICLR 2022 Submitted_

### Official Review · Reviewer_UXiJ · 2021-10-28

**Correctness:** 4
**Technical Novelty And Significance:** 3
**Empirical Novelty And Significance:** 3
**Recommendation:** 8
**Confidence:** 4

**Main Review:**

### strengths

- It is important to build the input-dependent RS methods.
- While there have been some previous works, most of them are mathematically invalid (as the authors argued).
- It is challenging to formulate such input-dependent RS with a valid certification, and the authors successfully address the problems.
- The paper is well written in a logical order.
    - Lemma 1 and Theorem 2 describe the worst-case decision region for $\sigma_0$ and $\sigma_1$. Given $\sigma_0$, to use the input-dependent $\sigma(x)$, it is required to certify for all valid $\sigma_1$ as stated in Theorem 6. If we know upper/lower bounds on the $\sigma_1$, it is easy to obtain a certified radius. Therefore, Theorem 8 justifies the design of $\sigma(x)$ in eqn (1) as we can easily compute the bounds of $\sigma_1$ as in Theorem 7 because it has globally bounded variations ($r$-semi-elastic).
- It is a general approach that can be extended and applied to other RS scenarios.

### (minor) weaknesses

- In Abstract, the authors (over)state that "... to overcome these flaws (certification vs acc trade-off or fairness issues)" which may mislead the readers to thinking that the paper solve these problems. However it is not quite true as stated in Section 4 regarding the results in Table 2 and 3 (the last paragraph in page 7).
    - However, I still believe that the paper has enough contributions.
- It would be nice to compare the results with ($r=0.00$ and $\sigma_{tr}$ increased) too.
    - concrete, actionable feedback: please show the corresponding additional results in Table 2 and 3.
- Except for the above two comments, the paper has no significant weakness in its content, but there are some major/minor presentation issues that must be addressed before published.
- "Using non-constant \sigma(x) defined in Section 2, ..." (page 2)
    - It is defined as eq 1. in Section 4.
- It would be better to use the same color for each method to avoid confusion in Figure 4.
- $\sigma_{tr}$ and trs (and $r$ and trr) seem to indicate the same thing, but the terms trs and trr have never been defined (in Figure 4).
- not useful [README.md](http://readme.md/) copied from the original RS repository.
    - for example, see [https://github.com/paperswithcode/releasing-research-code](https://github.com/paperswithcode/releasing-research-code)
- Please kindly correct me if I misunderstood anything.

### Others

- Compared to other parts, the curse of dimensionality part is hard to understand. Could you elaborate it more in detail (in easy words)?
- How long does it take to compute the $\sigma(x)$ in eqn (1), compared to the Monte-Carlo process? Is it negligible? It would be nice to provide the (additional) computation cost.
- The authors mention that "Here, we compare Cohen et al. (2019)’s evaluations for ... with our evaluations, setting ... (for CIFAR10 and MNIST, respectively), applied on models trained with Gaussian data augmentation, **using constant standard deviation roughly equal to the average $\sigma(x)$ or $\sigma$ used during evaluation**." How do you choose the training $\sigma_{tr}$ **before** evaluation phase?
- How do you choose $r, k$ and $m$?

**Summary Of The Paper:**

The authors propose an input-dependent randomized smoothing method with non-constant input-dependent $\sigma(x)$.

### contributions

- The authors generalize randomized smoothing with non-constant input-dependent $\sigma(x)$.
- The proposed method successfully addresses some intrinsic limitations/flaws in the previous (input-independent) RS.

**Summary Of The Review:**

The authors tackle important problems of RS.
They use their original method to solve the problem in a rigorous way and the method seems to be effective.
I thereby recommend the paper.

---

> ### Author Response · Authors · 2021-11-16
> **Thank you for your review!**
>
> $\textbf{PART 1/2:}$
>
> Dear reviewer UXiJ,
>
> Thank you for your time and your insightful comments, as well as the fact that you appreciate our work. Let us address your concerns in the following points:
>
> $\textbf{We overstate about input-dependent randomized smoothing (IDRS) in the abstract.}$
> - Thank you for this insight. We tried to change the formulation slightly to avoid such a strong indication that the mentioned problems are actually solved by IDRS.
>
> $\textbf{It would be nice to compare the results with } r=0.00 \textbf{ and } σ_{tr} \textbf{ increased too.}$
> - We do not find it necessary to compare with the setup $r=0.00$ and $\sigma_{tr}$ increased, because this setup is sharply worse than the setup of $r=0.00$ and $\sigma_{tr}=\sigma$. This was already discussed and empirically confirmed in [1], where authors demonstrate in Appendix H and F.3 that for constant $\sigma$ smoothing, the optimal training variance is on the level of evaluation variance. The clean accuracy of a smoothed classifier would quickly degrade if it was trained both with smaller or with a bigger level of training $\sigma$.
>
> $\textbf{The readme file is not useful.}$
> - We are working on the readme file and we will publish the changes by the rebuttal period deadline.
>
> $\textbf{Compared to other parts, the curse of dimensionality part is hard to understand. Could you elaborate it more in detail (in easy words)?}$
> - Thanks again for pointing this out. We try to elaborate on the curse of dimensionality extensively with explanations in Appendix B.2, B.3, B.4 and also partially in the main text. If the $\sigma(x)$ function varies too strongly the method is not able to certify any robust region by theory and, most importantly, the higher the dimension of the input the less variation is allowed.
> - Let us explain it here in more detail as well. In high dimensions, the probability of the worst-case ball (given fixed $p_B, \sigma_0, \sigma_1$) for $x_0=x_1$ (i.e. the case where we are trying to certify at least the very same point we are in, but with different smoothing variance) increases rapidly. For $x_0 \neq x_1$ and the same parameters, the probability of the worst-case ball is even bigger (as shown in Lemma 20). This means that having a fixed ratio $\sigma_0/\sigma_1$, our probability soon (with respect to increasing $N$) reaches the certification threshold 0.5, what implies that $x_1$ can no longer be certified with respect to $x_0$. The more dissimilar the $\sigma_0$ and $\sigma_1$ are, the smaller the dimension $N$ needs to be in order to make the $P_1(B)>0.5.$
>
> $\textbf{How long does it take to compute the } σ(x) \textbf{ in eqn (1), compared to the Monte-Carlo process? Is it negligible?}$
> - The computation time per sample ranges from practically instant to at most 4-5 sec per sample (compared to 15 sec per MC sampling on CIFAR). The average evaluation time depends on the $\sigma_b$ and $r$. For the case $r=0.01$ and bigger, the evaluation runs practically instantly. The 4-5 sec. evaluation time might occur for $r=0.005$ and smaller and small levels of $\sigma$. In general, the bottleneck is the computation of the cdf and quantile function of NCCHSQ if $\sigma_0$ and $\sigma_1$ are too close to each other. We discuss the computational complexity in Appendix E (the introduction to this section).
> - To sum it up, our computation is negligible for moderately high dimensions. For smaller dimensions and smaller networks, such as MNIST, the computation time could be comparable to MC sampling, but it is important to note that for smaller dimensions, we can also afford bigger rate r and in that case the evaluation is almost instant.
>
> $\textbf{The authors mention that "Here, we compare Cohen et al. (2019)’s evaluations for ... with our evaluations, setting ...}$
> $\textbf{for CIFAR10 and MNIST, respectively), applied on models trained with Gaussian data augmentation, using constant}$
> $\textbf{standard deviation roughly equal to the average } \sigma(x) \textbf{ or } \sigma \textbf{ used during evaluation."}$
> $\textbf{How do you choose the training } \sigma_{tr} \textbf{ before the evaluation phase?}$
> - We are sorry for the confusion. This description is actually a description of the $\sigma_{tr}$. In other words, the $\sigma_{tr}$ is set to be roughly equal to average evaluation-time $\sigma(x)$ or evaluation-time $\sigma$. We changed our phrasing in the main text too.

---

> > ### Comment · Reviewer_UXiJ · 2021-11-19
> > **Thank you for the detailed answer.**
> >
> > Dear authors,
> >
> > Thank you for the detailed answers. It's a nice paper. I'll keep my score.

---

> ### Author Response · Authors · 2021-11-16
> **Thank you for the review!**
>
> $\textbf{PART 2/2:}$
>
> $\textbf{How do you choose } r,k \textbf{ and } m \textbf{ ?}$
> - Each of the hyperparameters has its own purpose and it should be adjusted according to our needs as well as the situation. The $r$ needs to be small-enough so that we don’t lose too much due to the curse of dimensionality, but large-enough to provide significant improvement over constant $\sigma$. It can be designed either by trial and error or using plots for linear classifier or from Theorem 4 adjusting it so that the $\sigma_1/\sigma_0$ can’t go anywhere near enough this threshold within the certified radius. The $k$ should be large enough so that $k$-NN is robust, but on the other hand not too big so that the distance changes significantly with $x$. In our case it was chosen by observation of histograms of the resulting $\sigma(x)$ on the train dataset. The $m$ needs to be chosen so that the smallest $\sigma$ values roughly correspond to the base $\sigma$. Or, alternatively, so that average $\sigma(x)$ is roughly equal to the base $\sigma.$
> - We also added Appendix E.1 section to discuss the hyperparameter selection.
>
> $\textbf{Other:}$
> - Please note that we also fixed all your minor comments pointing to writing or presentation imperfections.
>
> In case you would have any further questions / concerns or you would like to react to our answers to your comments, please do so. We are happy to discuss anything unclear or controversial.
>
> REFERENCES:
>
> [1] Cohen, Jeremy, Elan Rosenfeld, and Zico Kolter. "Certified adversarial robustness via randomized smoothing." International Conference on Machine Learning. PMLR, 2019.

---

### Official Review · Reviewer_jyXD · 2021-11-01

**Correctness:** 4
**Technical Novelty And Significance:** 3
**Empirical Novelty And Significance:** 1
**Recommendation:** 3
**Confidence:** 4

**Main Review:**

Strengths:
- The main merit of this work is the theoretical contribution in section 2. It is quite useful deriving the conditions under which the input independent smoothing is theoretically valid.
- The summary for the steps in deriving the radius in page 3 is neat and useful.

Weaknesses: I will summarize the main points below and detail the reasons behind each of them afterwards.
1.  The experimental evaluation presented in this work is insufficient. Moreover, the proposed methodology has no notable improvement over the considered baseline.
2. The writing of this work along with the presented figures can be significantly improved.
3. There are several parts that need more elaborative discussion.

Regarding the first point:
- While the motivation of this work is the "certified accuracy waterfalls" the proposed objective in Equation (1) does not solve the problem. The results presented in Figure 4 show that the proposed method suffer from a similar but slightly delayed issue.
- The presented experiments considered the weakest baseline (Cohen et.al) were the effectiveness of the method should be tested with stronger baselines including MACER and SmoothADV and preferably on larger datasets such as ImageNet.
- While the objective in (1) could arguably increase the value of $\sigma$ for points "far from decision boundary", it is not clear how would it decrease it for points close to the boundary. This point is related to the results on CIFAR10: why does the proposed method perform slightly worse on small radii? Shouldn't the values of $\sigma$ get adjusted for points nearby decision boundaries such that some improvement is got in that regime?

Regarding the second point:
- It is not clear why did you need to assume the semi-elasticity bound.
- There are several notations introduced within text such as $a := \|\delta\|$ and used later without referring to the same variable.
- The plot in Figure 4 should have consistent colors. An envelop curve over the different values of $\sigma$ could benefit the comparison.

Regarding the third point:
- It is not clear how in the special case for when $\sigma_0 = \sigma_1$ will the results be reduced to the one in Cohen et. al.
- It is clear how the objective in (1) could be used during training, but it is not clear how is it used at test time. What is $\mathcal N_k(x)$ in that case?
- The discussion in the paragraph below (1) is a little vague. Instances belonging to the same class could significantly differ in the input space (e.g. instance of white dog and instance of black dog). How could these distances relate to the position to decision boundaries?

Other questions:
- Regarding the curse of dimensionality: While the variation between $\sigma_0$ and $\sigma_1$ is restricted by Theorem 4, is it possible to break the input space into small regions where the variation is small within the region but large enough between instances far from each other?
- Since the proposed method requires knowing $\mathcal N_k(x)$, How will this neighborhood be obtained in test time?





**Summary Of The Paper:**

This work extends the theoretical results from Cohen et.al to the input dependent setup. In particular, the derive the theoretical conditions for the validity of data-dependent randomized smoothing through a similar procedure to the one in Cohen et.al. At last, a function that optimizes the smoothing parameters per input was proposed and validated experimentally on MNIST and CIFAR10.

**Summary Of The Review:**

The theoretical analysis presented in this paper is both novel and of interest. However, there are several weaknesses of this paper that need to be addressed including the experimental setup, the effectiveness of the proposed method, and the presentation.

---

> ### Author Response · Authors · 2021-11-16
> **Thank you for your review!**
>
> $\textbf{PART 1/2:}$
>
> Dear reviewer jyXD,
>
> Thank you for your time and your insightful comments. Let us address your concerns in the following points:
>
> $\textbf{Our } \sigma(x) \textbf{ design does not solve our motivation, which is “certified accuracy waterfalls”}$
> - It is true that our design also suffers from certified accuracy waterfalls. Important thing to notice here is, however, that our waterfalls are no longer that “cliffy” and they are in general less steep. Please have a look at our new evaluations for MNIST, which are depicted on new Figure 4, where we use stronger rate $r=0.02$ (this is justified, because MNIST is a smaller-dimensional dataset than CIFAR). You can see that the drops are considerably less steep, which results in outperformance of constant sigma for these big certified radiuses.
> - Our motivation, though, is not just to treat certified accuracy waterfalls problem. Our main motivation and contribution is to show that input-dependent RS (IDRS) has a limited power to mitigate this issue. Even if we used a better $\sigma(x)$ design and would be able to achieve even nicer certified accuracy curves, our Theorem 4 says that this gain is limited, since the maximal possible ratio between $\sigma$’s used for different samples is limited.
> - Please also have a look at Appendix E.2 and E.3, where we plot certified accuracy curves using different rates to show that the waterfalls could be mitigated even further (with a possible price of losing a certain amount of clean accuracy).
>
> $\textbf{We do not compare to the state-of-the-art models like SmoothAdv or MACER and preferably on datasets like ImageNet.}$
> - We explain why we do not compare with SmoothAdv and MACER in Appendix E.4. The only problem we claim to treat empirically with our particular design of $\sigma(x)$ is the certified accuracy waterfalls, which is a phenomenon completely independent of the training method (what is the power of SmoothAdv and MACER). Since the waterfalls are independent of the training method, it is justified to compare to simpler models, such as Cohen’s. It would be necessary to compare to SmoothAdv and MACER, if we would propose a new robust training method.
> - We are not able to compare on datasets of ImageNet size, because, as we discuss in the last paragraph of Section 3, this method cannot be used in such high dimensions (yet), until certain numerical issues are solved (see Section 3).
>
> $\textbf{Do we use smaller } \sigma(x) \textbf{ for points close to the decision boundary than the constant } \sigma \textbf{ for Cohen’s evaluation?}$
> $\textbf{If yes, how come that our clean accuracy is slightly worse on CIFAR10?}$
> - This is the right intuition. However, we designed the sigma function in (1) such that it “starts” at values equal to the constant $\sigma$ and can only get larger. Therefore, $\sigma(x)$ values are actually always at least as large as those used in the constant $\sigma$ evaluation. This means, we can’t outperform Cohen in terms of clean accuracy, by design. Our motivation is, however, that if the $\sigma(x)$ function is chosen well, the points near to the decision boundary should enjoy values very close to the base $\sigma$, thus we should not lose much clean accuracy.  To see a comparison where the average level of $\sigma(x)$ is roughly the same as the constant $\sigma$, see Appendix E.6.3. There we show that our $\sigma(x)$ function is not the best possible design, because we can’t achieve much better certified accuracy without losing at least some amount of clean accuracy.
>
> $\textbf{It is not clear we need to assume the semi-elasticity bound.}$
> - The result in Theorem 4 bounds the ratio between $\sigma_0$ and $\sigma_1$ for which we can certify $x_1$ w.r.t. $x_0$. In order to obtain a valid certified radius, we need to check for all $x_1$ in the neighborhood of $x_0$. Therefore, nowhere near the $x_0$, the $\sigma_1$ can be out of these bounds. This creates an implicit restriction on the ratio between $\sigma_0$ and $\sigma_1$ near x_0. That is essentially semi-elasticity. In summary, the semi-elasticity was derived as a necessary condition for the verification procedure to work. We provide a thorough explanation behind the necessity of semi-elasticity in the second-to-last paragraph of Section 2 and in between Theorems 6 and 7.
>
> $\textbf{There are several notations introduced within text such as } a:=|δ| \textbf{ and used later without referring to the same variable.}$
> - The variable $a$ is always used as the distance between $x_0$ and $x_1$. We will be thankful if you could point out the inconsistencies of the notation and the place where these occur.

---

> > ### Comment · Reviewer_jyXD · 2021-11-20
> > **Thank you for the response.**
> >
> > I would like to thank the authors for the provided response. While I appreciate the theoretical results presented in this work, my earlier concerns about the proposed algorithm along with the experimental evaluations are not resolved. Moreover, while the theoretical results show that indeed input independent RS suffer from the curse of dimensionality, the limited power of the proposed IDRS can be mainly attributed, in this work, to the proposed method along with $\mathcal N_k$. That is, better designs of the IDRS with leveraging better initial points for the optimization could mitigate this issue as dictated in the main review. Next, I am going to detail my response for *some* of the answers provided by the authors.
> >
> > - Regarding the "certified accuracy waterfalls": The effect of the proposed method, if $\sigma$ is only increasing from its initialized value (as dictated by the authors in the response), will be very similar to increasing a global $\sigma$ for all points: degrading clean accuracy while delaying the "certified accuracy waterfalls". Note that this confirms with the reported results at high levels of $\sigma$ such as 0.5 where the proposed method underperforms the baseline. One potential comparison is to compare envelop curves of the proposed method along with the baseline cross validated at several values of $\sigma$.
> >
> > - Regarding the comparison to state-of-the-art: While indeed both MACER and SmoothAdv suffer from the "certified accuracy waterfalls" similar to Cohen et.al, methods that provide enhancements to weaker baselines are not necessary improving the stronger ones. Thus, it is still important to show that the proposed method is able to provide *notable* improvements on the strong baselines as well.
> >
> > - Another *very important* missing comparison is the certification running time between the proposed method and the fixed $\sigma$ RS. The later outputs  a certified radius for a given model and input while the earlier verifies if an input is certified with a given radius. Please correct me if I am missing anything.
> >
> > - It is still not clear to me why the need to the semi-elasticity assumption. Corollary 5 and Theorem 6 do not need such an assumption. For example, why not bounding the variation of  $\sigma_1$ from $\sigma_0$ linearly?

---

> > > ### Author Response · Authors · 2021-11-21
> > > **Thank you for your further response**
> > >
> > > Dear reviewer jyXD,
> > >
> > > thank you again for your response. We would like to kindly ask you to elaborate on what you meant by "leveraging better initial points for the optimization"? We do not do any optimization in our method.
> > >
> > > You are correct about the fact that our method has limited power. This is clearly stated in our work and also follows from our experiments and ablation studies. We also explain, why it is so. However, note, that the main point and the main contribution of our work is the theoretical one - the curse of dimensionality. By presenting results of our proposed $\sigma(x)$ design, we mainly wanted to demonstrate that the IDRS is finally ready to be used rigorously, under the same testing conditions as the constant RS and it is of great importance for further research to design better $\sigma(x)$ function, which would, for instance, be able to outperform constant smoothing in terms of certified accuracy even without losing the clean accuracy. In our opinion (which might be considered biased, yet we try to be objective) a paper that shows such a significant phenomenon in IDRS and prepares fully the rigorous practical usage of $\textit{any}$ $r$-semi-elastic $\sigma(x)$ function (so the only one that is enabled by the curse of dimensionality) is a very important piece of work for both the researchers and wider public. The regime presented in our paper is provably (and we prove it) the only input-dependent regime except for locally constant regions (which have many different issues) that has a chance to work. This is immensely important information for everyone who wants to either research IDRS or use it.
> > >
> > > Put differently, we present three contributions (the curse of dimensionality, the practical framework for usage of any $\sigma(x)$ design, and the concrete $\sigma(x)$ design) from which the one we focused least on in our paper is the only one that possesses issues. We are aware that this particular contribution could be improved, but on the other hand, are quite convinced that the previous two contributions are sufficiently important for the community. As it is with all papers, there is always some part, which is left for further research. Wouldn't it work that way, research would sooner or later recede because single papers would do everything that is possible to be done in the field. In our case the part left for further research it is the improvement of the $\sigma(x)$ design.
> > >
> > >  Next, let us address your points:
> > >
> > > - It is correct that using $\sigma(x)$ values starting from the $\sigma_b$ which is equal to the constant $\sigma$ used for comparison is similar to increasing the global $\sigma$. This, however, is the case just in the case when the design of $\sigma(x)$ is not optimal. If the design of $\sigma(x)$ is optimal for the task, such as in the toy example in new Appendix A., the difference between our method and just increasing the global level of $\sigma$ is significant - IDRS is capable of mitigating the waterfalls significantly while not losing the accuracy. The reason why we lose a certain amount of accuracy is, that we occasionally use bigger $\sigma(x)$ values for samples close to the decision boundary - an effect caused by suboptimal design since the $\sigma(x)$ function does not perfectly correspond to the distance from the decision boundary. Now, the problem with envelope curves is that they mask the most interesting regions - the waterfalls. Therefore the comparison using envelope curves is not suitable for our contribution.
> > >
> > > - We agree that in general methods that provide enhancements to weaker baselines are not necessarily improving the stronger ones. In this case, we are sure they do. The reason is simple - consider the samples for which the maximal certified radius using constant $\sigma$ is reported (i.e. those that are in the waterfall region). For these samples our $\sigma(x)$ uses different, but in average bigger values than $\sigma$. For this reason, it is obvious that the certified accuracy waterfalls are mitigated also in this case. In general, they will be mitigated in any case where the waterfalls are present. Though we are sorry that we don't provide the comparison for SmoothAdv and MACER, we can't make these comparisons anymore.
> > >
> > > - We must disagree on this point. First, the discussion on this is not missing, it is just present in Appendix E, first paragraph. We also elaborate on this in our answer to reviewer UXiJ. In all cases, the computation time of the certified radius is much smaller than the computation time of the MC sampling and forward passes. In most cases, the computation is almost instant (compared to seconds or tens of seconds of the MC sampling). Second, we disagree that our method does something qualitatively different from constant RS. We also take a model and a sample and provide the certified radius.

---

> > > ### Author Response · Authors · 2021-11-21
> > > **Thank you for your further response**
> > >
> > > As for the last point:
> > >
> > > - We tried to explain it as well as we could, but maybe it will help if we put it in contrast with Lipschitz continuity (what would be equivalent to bounding the variation linearly). The reason is that our Theorem 4 gives constraints on the $\textit{ratio}$ $\sigma_1 / \sigma_0$ instead of the difference $\sigma_1 - \sigma_0$. This is the reason, why we need to bound the variability of ratios between used $\sigma(x)$ values instead of differences. The variability of ratios is controlled by semi-elasticity.

---

> ### Author Response · Authors · 2021-11-16
> **Thank you for your review!**
>
> $\textbf{PART 2/2:}$
>
> $\textbf{It is not clear how in the special case of } \sigma_0=\sigma_1 \textbf{ will the original Cohen’s results be obtained.}$
> - Thank you for pointing this out. We make a small note on this in the paragraph after Theorem 2. The case $\sigma_0=\sigma_1$ is relevant just for Theorem 2 and Theorem 3, so we find this note sufficient for the text.
> - Note that the computation of the balls in Theorem 2 will change, so that all quadratic terms vanish if $\sigma_1=\sigma_0$. This will result in linear half-space. Also, the balls in Theorem 2 “converge” to this linear half-space as $\sigma_1$ goes to $\sigma_0$. Similarly, it would be possible to prove that $\xi$ functions have a limit for $\sigma_1$ going to $\sigma_0$ and this limit would be the probability of the worst case classifier in the constant $\sigma$ regime.
> - From Theorem 6 it follows that in practical evaluation, the case $\sigma_0=\sigma_1$ is not important for us, because it is always better than if $\sigma_0 \neq \sigma_1.$
>
> $\textbf{What is } N_k(x) \textbf{ during test time?}$
> - During test time, we use the train set again to compute the distances. So $N_k(x)$ is the set of $k$ nearest train neighbors. This also makes sense, since the network’s decision boundary is determined by train samples, not test samples and thus the distance from the decision boundary should also depend on the train set.
> - We make this clear in the text right after Equation 1, where we denote $\{x_i\}_{i=1}^d$ to be the training set.
>
> $\textbf{Instances belonging to the same class could significantly differ in the input space (e.g. instance of white dog and instance of black dog).}$
> $\textbf{How could these distances relate to the position to decision boundaries?}$
> - This is an important question. First, let us explain that we do not require the instances close to each other in euclidean norm to come from the same class. What is important is the intuitive assumption that if there is a small cluster of instances that are close in euclidean sense (not necessarily from the same class, but imagine similarly positioned white dog and white cat), the network will be forced to adapt its decision boundary to these samples precisely, since these samples have a high leverage in terms of loss.
> - We agree, however, that this intuitive view is rather vague and we find it important to find a design of $\sigma(x)$ function that would be motivated less vaguely as a future work.
>
> $\textbf{While the variation between } \sigma_0 \textbf{ and } \sigma_1 \textbf{ is restricted by Theorem 4,}$
> $\textbf{is it possible to break the input space into small regions where the variation is small within the region}$
> $\textbf{but large enough between instances far from each other?}$
> - It would be possible, but we would lose certification on the boundaries of the regions, it is highly non-trivial to design reasonable division and a similar setup was tried for instance by [1], but their methodology had issues mentioned both in the reviews of the work and our work, Appendix C.1. We don’t see an easy way to employ this technique efficiently.
>
> $\textbf{Other:}$
> - Please note that we also fixed the issues with Figure 4.
>
> In case you would have any further questions / concerns or you would like to react to our answers to your comments, please do so. We are happy to discuss anything unclear or controversial.
>
> REFERENCES:
>
> [1] Wang, Lei, et al. "Pretrain-to-Finetune Adversarial Training via Sample-wise Randomized Smoothing." (2020).

---

### Official Review · Reviewer_Q6Nf · 2021-11-03

**Correctness:** 3
**Technical Novelty And Significance:** 3
**Empirical Novelty And Significance:** 1
**Recommendation:** 5
**Confidence:** 2

**Main Review:**

Strengths:

The "takeaway" from this paper is a negative result (both theoretical and experimental), but I think it is of relevance to the community. The idea of input-varying noise $\sigma(x)$ is attractive (as the authors note several other papers have recently touched upon this idea), and this paper makes a strong theoretical case for why it's unlikely to outperform Cohen et al's baseline.

Weaknesses:

The main weakness lies in the fact that this is a negative result -- it doesn't provide any real improvement over Cohen et al. 2019, and the authors acknowledge that this method doesn't scale to modern ImageNet-scale datasets.

Experimentally, the results could use some work. Writing-wise, it would be helpful to more clearly differentiate CIFAR-10 and MNIST in Tables 2 and 3. In Figure 4, the colors orange and blue are swapped between the baseline and the proposed methods when the setting changes between CIFAR-10 and MNIST (this is extremely confusing)! It would additionally be helpful to make clear that the $r=0$ case in Table 2 corresponds to clean accuracy.

It's unclear how the authors derived Equation (1) -- is it possible that there exists another choice of $\sigma(x)$ that could yield superior empirical results?

It seems unlikely that training with a fixed noise level would be the optimal way to train input-dependent smoothing models -- have the authors tried, for example, changing the noise level of the Gaussian at training time to be input-dependent as well?

It would be helpful to evaluate the proposed Equation (1) on toy simulated datasets in low dimensions, simply to verify that we can indeed see an improvement from input-dependent smoothing. Figure 1 is helpful for building intuition, but I'd be curious in seeing how that result scales with dimensionality $N$.

Miscellaneous:

Typo in Figure 4, "coparison".

I'd be willing to revisit my rating if the experiments and writing could be cleaned up substantially.

**Summary Of The Paper:**

The authors study input-dependent adversarial smoothing i.e. the setting where the smoothing distribution has noise level $\sigma$ dependent on $x$. They begin by deriving the robustness certificate in this setting, and show that it suffers from the curse of dimensionality -- in order to maintain non-vacuous certificates in high dimensions, $\sigma(x)$ must be prohibitively smooth. They then provide empirical experiments using an example function $\sigma(x)$ on CIFAR-10 and MNIST and show that performance is more or less the same compared to the fixed noise level baseline (Cohen et al. 2019).

**Summary Of The Review:**

This paper presents theoretical and experimental negative results on input-dependent randomized smoothing.

I think it's likely to be of *some* interest to those working on randomized smoothing.

But the wider community will likely be uninterested by this negative result.

(Additionally the paper could use more work on the experiments and writing in particular).

---

> ### Author Response · Authors · 2021-11-16
> **Thank you for your review!**
>
> Dear reviewer Q6Nf,
>
> Thank you for your time and your insightful comments. Let us address your concerns in the following points:
>
> $\textbf{Our main message is a negative result, we do not outperform classical smoothing much and can’t scale to high-dimensional problems.}$
> - Negative results are as important and valuable as the positive ones, because they force the community to critically and carefully evaluate and validate popular or currently trending approaches and move us towards science of a higher quality. Essentially, negative results (including the machine learning field) are necessary for the future research to fit the high scientific standards. In particular, with our work, we notify both a wide community and RS researchers, that the intuitive, well-motivated and popular idea of input-dependent RS (IDRS) needs to be approached in a systematic manner with particular caution. The existence of works that already tried and failed to implement IDRS just amplifies the need for this notification. The curse of dimensionality is an important phenomenon and knowledge about it is inevitable to better understand IDRS and it might prevent lost research time or even misuse of the technique in practice.
> - On the other hand, we do not claim that IDRS is now useless. As we clearly state in our paper, the $\sigma(x)$ function we use is not the optimal design and improvement of this design might lead to considerable (though limited, due to the curse of dimensionality) improvements. Our framework enables researchers to implement the new design of $\sigma(x)$ very comfortably.
> - The fact that the method can’t scale to high-dimensional problems such as ImageNet classification is true just because of numerical reasons (see last paragraph of Section 3). These issues might be possibly resolved in future by introducing more precise implementations of the cdf and quantile function of NCCHSQ.
> - Also, please have a look at the new Figure 4 and Tables 2, 3, where we decided to change evaluations on MNIST, now using rate $r=0.02$ instead of $r=0.01$. This change is justified by the fact that MNIST has a smaller dimension and we can afford a bigger rate. The results are now even stronger.
>
> $\textbf{How the Equation (1) is derived and is this design of } \sigma(x) \textbf{ function optimal?}$
> - The $\sigma$ function was derived first adhering to the restrictions we have: $r$-semi-elasticity and at least rough correspondence to the distance from the decision boundary. As discussed in the main text (the text between Equation (1) and Theorem 8), the naive intuition behind this function is as follows:
> - “Intuitively, if a sample is far from all other samples, it will be far from the decision boundary, unless the network overfits to this sample. On the other hand, the dense clusters of samples are more likely to be positioned near the decision boundary, since such clusters have a high leverage on the network’s weights, forcing the decision boundary to adapt well to the geometry of the cluster.”
> - Yes, it is possible that there exists a better choice of $\sigma(x)$ function and we believe it is the most promising direction for the future work as we indicate at the end of Section 6.
>
> $\textbf{What about input-dependent train-time Gaussian data-augmentation?}$
> - We also believed the input-dependent train-time Gaussian data-augmentation should be helpful. We investigate training with the non-constant sigma as well in Appendix E.4. We use the same input-dependent $\sigma(x)$ function as in the evaluation time, possibly with different scaling. From the results we see, however, that the usage of input-dependent train-time augmentation performs very similarly (possibly even touch worse) as the setup where we augment with a constant level of $\sigma$ put so that in average it corresponds to the test-time average $\sigma(x).$
>
> $\textbf{Provide the evaluations on toy dataset scaling with dimension $N$.}$
> - We are working on this comment and we will add the results of these experiments by the rebuttal deadline.
>
> $\textbf{The wider community will likely be uninterested by this negative result.}$
> - We are of the belief that this result is of great relevance for the wider community too. Please see our answer to your first comment.
>
> $\textbf{The paper could use more work on the experiments.}$
> - Please note that just the most relevant experiments and results could be included in the main text. We provide almost 10 pages of additional experiments in Appendix E, investigating many different scenarios and performing massive ablation studies. We also provide insights of our two-dimensional toy experiments in Appendix A.
>
> $\textbf{Others:}$
> - We also addressed your minor comments regarding the writing and figures (for instance fixing Figure 4). We hope the updated version is easier to read. We will be glad to address your further comments if you have any.
>
> In case of any further questions, please don't hesitate to ask. We are happy to answer.

---

> > ### Author Response · Authors · 2021-11-19
> > **The experiment is finished**
> >
> > Dear reviewer Q6Nf,
> >
> > we would like to inform you that we finished our multidimensional toy experiment, which you requested. We added a subsection Appendix A.5, where we present the results of the experiment. As it can be clearly seen, IDRS can also outperform RS considerably, if the $\sigma(x)$ corresponds very well to the geometry of the data and the network's decidison boundary.

---

### Official Review · Reviewer_j688 · 2021-11-04

**Correctness:** 4
**Technical Novelty And Significance:** 2
**Empirical Novelty And Significance:** 1
**Recommendation:** 5
**Confidence:** 4

**Main Review:**

Strengths:

1. The main theorem looks reasonable to me. Despite all the complex equations, I think the intuition is that the worst-case classifier f* when using input dependent randomized smoothing has a spherical structure, and its volume shrinks in high dimensional space.

2. The proposed input-dependent randomized smoothing standard deviation function $\sigma(x)$ is novel and different from existing works (but unfortunately the improvements are quite small).

Weaknesses:

1. The theoretical study is limited to the setting where isotropic Gaussian noise is applied. However, when considering input dependent smoothing, I feel a non-isotropic Gaussian noise would be more useful - for example, on MNIST, it can be quite reasonable to apply a larger noise on the pixels that are black (background) and apply smaller noise on the white pixels. By applying this kind of input dependent smoothing, the limitation of input dependent smoothing found in this paper does not necessarily hold. I hope the authors can discuss more on this more useful setting.

2. One missing factor in the analysis, is that in high dimensional datasets, the distances between the images are actually pretty large: for example, the L2 distance between two ImageNet images from different classes are actually large, grown by the factor of $\sqrt{N}$ ($N$ is the data dimension).  In terms of smoothing, I think it totally makes sense that when we are certifying a single image, the $\sigma(x)$ function should not change too much within a small radius $\| \delta \|$. However, since different images are already very far away, we can certainly apply different smoothing $\sigma(x)$ on different images.

3. The design of $\sigma(x)$ is pretty naive and the empirical results are very weak - compared to smoothing with uniform noise, the gain is too little.  I feel this is due to the restriction of using isotropic Gaussian noise.  Since the variance of the Gaussian noise is the only thing we can change and they cannot be changed too much, the proposed input dependent smoothing scheme in this paper is not very useful.

4. No comparisons to existing baselines (such as Wang et al. 2021) using input dependent smoothing were given.

**Summary Of The Paper:**

This paper studies the input-dependent randomized smoothing technique - it adds isotropic Gaussian noise with different variance on different input x to give robustness certificates.  The hope is that this can perform better than applying a uniform noise.  The main theoretical claim in this paper is that when adding input dependent isotropic Gaussian noise, the variance of the noise should not change too much among different inputs. Based on this, the authors propose an input dependent smoothing mechanism based on k-nearest neighbor and evaluate its performance on MNIST and CIFAR datasets.

**Summary Of The Review:**

Overall, I do like the analysis for input dependent randomized smoothing, but I think the overall impact of this work is rather limited. I rate the paper at borderline because I feel the theoretical part of this paper has certain limitations and the proposed procedure is also not very effective in practice and has quite small improvements. To further improve the contribution of this paper, as I mentioned in the "weaknesses" above, the authors can study the more useful setting of applying non-isotropic input dependent noise, and develop better $\sigma(x)$ function to improve smoothing performance.

---

> ### Author Response · Authors · 2021-11-16
> **Thank you for your review!**
>
> $\textbf{PART 1/2:}$
>
> Dear reviewer j688,
> Thank you for your time and your insightful comments. Let us address your concerns in the following points:
>
> $\textbf{Why don’t we study the input-dependent anisotropic randomized smoothing?}$
> - The focus of our paper is to study the possible gain of using the input-dependent RS (IDRS). Since the main focus of researchers in RS is now the isotropic case and the IDRS has not yet been successfully integrated / studied even for this case, we find it natural to first answer the question of IDRS for this case. As you can see in this paper, IDRS is a self-standing and challenging research topic already in the isotropic regime. Therefore, it is only natural to leave the anisotropic case as a direction for future work.
> - To elaborate on anisotropic smoothing: Assume the covariance matrices at $x_0$ and $x_1$ differ only by a scalar factor ($\sigma_0^2 \Sigma$ and $\sigma_1^2 \Sigma$). To compute the worst case classifier and probability of class $B$, we just need to project the data by matrix $\Sigma^{-1/2}$ to obtain isotropic smoothing. In this space, we can use our theory to prove the curse of dimensionality. Then, we can just backproject the input-space back to the original one and prove the curse of dimensionality there. For the more general case, where we allow for arbitrary covariance matrices, the worst-case classifier will become a level set of a general quadratic form and the analysis becomes tedious. We are not able to prove the curse of dimensionality in this case (and it is questionable, whether it is indeed present in all scenarios), but, we believe that neither the rigorous certification can be established in this case and if yes, it would be worthy of a separate publication.
> - We will be glad if you could provide references to the work using this type of randomized smoothing. The only work we are aware of is [1].
>
> $\textbf{With increasing dimension, the distances grow as $\sqrt{N}$, what is a countereffect to the curse of dimensionality.}$
> - This is a very insightful comment and we agree that it is important to mention this effect in the main text. We added a paragraph to our theory section (the last paragraph of Section 2) elaborating on this. Moreover, we double-checked the statements we make in our text to make sure they are not misleading or in conflict with this (the only change we made is in the first paragraph of the conclusion, where we change one sentence not to sound so pessimistic).
> - However, please note that this effect doesn’t affect neither our claims, nor our message, nor the statements we make in our text. The curse of dimensionality is still present and it is important to know about it in order to avoid false certifications or fruitless attempts to introduce IDRS schemes that don't comply with these restrictions. Moreover, even though the interval of feasible ratios between $\sigma_1$ and $\sigma_0$ does not shrink for increasing dimension, but rather stays roughly constant, this serves as a proof that regardless of the dimension, the usage of IDRS is limited (though not infeasible).
> - It is also important to say, that even though theoretically the IDRS is feasible regardless of the dimension, practically the curse of dimensionality really makes a big difference. The problem is, that for very small rates $r$, the computation of $\xi$ functions is numerically impossible (because we do not have precise-enough methods to compute NCCHSQ cdf and quantile function for extremely big inputs. See more at the end of Section 3).
>
> $\textbf{Our proposed $\sigma(x)$ is naive and the empirical results are weak.}$
> - Yes, the proposed $\sigma(x)$ is rather naive (yet the design of the most sophisticated $\sigma(x)$ is not the main focus of this work). This is because it is challenging to design a simple semi-elastic function with known constant which would at least roughly correspond to the decision boundary. The design of a better $\sigma(x)$ function is, therefore, the most important practical direction of the further development of IDRS, as we claim in Section 6. The empirical results do not improve upon the constant RS much, however that only demonstrates that it is indeed not easy to do IDRS in a high-dimensional regime, which is the main message of this paper.
> Regarding the usage of non-isotropic noise, we believe it is of immense difficulty to implement input-dependent anisotropic RS with rigorous certificates. More is explained in the answer to your first comment.
> - By the “scheme” you mention in your comment, do you mean the general framework we introduce in Section 3 or our concrete design of $\sigma(x)$ introduced in Equation 1? We believe the scheme (by which we mean the general framework we introduce in Section 3) is useful, because it is rather general and enables us to use any $\sigma(x)$ function, possibly one that is better than ours.

---

> ### Author Response · Authors · 2021-11-16
> **Thank you for your review!**
>
> $\textbf{PART 2/2:}$
>
> - Additionally, we have decided to change our visualizations in the main text for MNIST, using instead of rate $r=0.01$ a rate $r=0.02$, which is still useful for MNIST, because it has a smaller dimension. This rate yields empirically stronger results. Please, find these experiments in new Figure 4 and Tables 2, 3.
>
> $\textbf{Missing comparisons to existing baselines (such as Wang et al. 2021 [2]) using input dependent smoothing.}$
> - Unfortunately, it is not possible to compare to existing IDRS baselines, because we are aware of none, which would provide results suitable for a fair comparison. We discuss this extensively in Appendix C. In the case of Wang et. al. [2], the main reasons are as follows:
> - Their method is memory-based, meaning it requires storing the test samples, which is a much different setup from ours and provides a lot of beneficial information.
> - Their performance is order-dependent, meaning that the order of samples in the test set might change the resulting smoothed classifier and the results. Since authors don’t provide results for the worst-case order (which would account for a possible adversarial attack on the order), their results are not justified.
> - Their performance depends on the number of test samples. If this method was to be evaluated on more test samples, conflicts would begin to occur frequently (because the distances between test samples would become on average smaller than the certified radiuses), resulting in severe performance degradation.
>
> In case you would have any further questions / concerns or you would like to react to our answers to your comments, please do so. We are happy to discuss anything unclear or controversial.
>
> REFERENCES:
>
> [1] Eiras, Francisco, et al. "ANCER: Anisotropic Certification via Sample-wise Volume Maximization." arXiv preprint arXiv:2107.04570 (2021).
>
> [2] Wang, Lei, et al. "Pretrain-to-Finetune Adversarial Training via Sample-wise Randomized Smoothing." (2020).

---

> > ### Comment · Reviewer_j688 · 2021-11-29
> > **Thank you for your response**
> >
> > Thank you for your answer and apologize for my late reply. I like the discussions in the reply especially on the part for increasing dimensions N, and I am happy that the authors added these discussions in the revision.
> >
> > My most important concern on this paper is still on the limited applicability of the theoretical results. Reviewer jyXD also shares the same concern. I feel it is not a very attractive story to first propose a relatively naive algorithm, and then show this algorithm has theoretical limitations and does not work. In fact, with a better design of the algorithm (as also mentioned by Reviewer jyXD) these limitations might be mitigated. The paper does contain a lot of technical efforts, but I am not sure how much impact this paper can have and how useful the results can be in its current version.
> >
> > Regarding the comparison to Wang et al., 2021 [1], it is a minor concern, but I think it still makes sense to make a best-effort comparison, e.g., using a few different random orderings and a fixed number of test samples (e.g., 500).
> >
> > And I also checked out the new figures but still feel the improvements are minimal (and for MNIST $\sigma=0.5$ it is worse than Cohen's).
> >
> > Thus, I still rate the paper at borderline, and hope the authors can work on extending their results and ideally propose an improved input-dependent smoothing scheme that performs better based on these theoretical insights.

---

> > > ### Author Response · Authors · 2021-11-29
> > > **Thank you for your answer**
> > >
> > > Dear reviewer j688,
> > >
> > > thank you for your final remarks.
> > >
> > > Even though we don't think it could change neither your opinion nor your decision, let us make one small answer anyway.
> > >
> > > "I feel it is not a very attractive story to first propose a relatively naive algorithm, and then show this algorithm has theoretical limitations and does not work. In fact, with a better design of the algorithm (as also mentioned by Reviewer jyXD) these limitations might be mitigated."
> > >
> > > We are of opinion that this view misunderstands our message and our work in general. It is not true that we show that our naive algorithm has theoretical limitations. That would, indeed, make no sense and have no value to the community. On the contrary, we show that $\textbf{any}$ input-dependent algorithm has these limitations. The fact that our design has them too is just a mere demonstration of our theoretical results. Indeed, the limitations of our algorithm could have been mitigated by a better design, but the point is that we show that any improvement would be $\textit{limited}$ by the curse of dimensionality.
> > >
> > > We hope you based your decision on this true interpretation because we feel there is a huge difference between your and our phrasing which makes a huge difference in the contribution of this paper.
> > >
> > > (*) In case you meant these sentences such as: "You worked just with isotropic, instead of anisotropic randomized smoothing", then we think we have already reasoned this decision. In short recap, the isotropic smoothing is acknowledged and most researched direction in the community and the step from constant isotropic RS to IDRS is not naive and it is rather general. But we think you meant it the first way we described anyway, so consider this to be a side note.

---

> > > > ### Comment · Reviewer_j688 · 2021-11-29
> > > > **Thanks for the quick clarification, but there is no misunderstanding here**
> > > >
> > > > Thanks for the clarification, but to be clear, there is no misunderstanding here. I believe one major limitation in this work is to only consider isotropic Gaussian noise in the entire analysis. I feel this is not quite useful for input dependent randomized smoothing, because there is only one free parameter (variance) to adjust and it leaves too few freedom, thus it does encounter the theoretical limitations and has very minor empirical improvements. I feel input-dependent smoothing can be more useful if a more flexible smoothing scheme can be considered (which I referred to "better algorithm design"), especially considering the high-level features of input images (e.g., main object vs. background).

---

> > > > > ### Author Response · Authors · 2021-11-29
> > > > > **Thank you for clarification**
> > > > >
> > > > > Thank you for quick clarification of your sentences.
> > > > >
> > > > > We just realised you might have meant it in this way at the very same moment you have written it.
> > > > >
> > > > > In this case there is probably nothing more to be said here, because we already explained extensively why we didn't consider the anisotropic smoothing and that we think the anisotropic smoothing and input-dependent smoothing are somewhat orthogonal ideas which can be combined only after both ideas are properly investigated independently.
> > > > >
> > > > > Thank you, again, for your time and review.

---

> ### Author Response · Authors · 2021-11-17
> **One more insight**
>
> Dear reviewer j688,
>
> we realized one more important argument to the issue with increasing distances.
>
> $\textbf{With increasing dimension, the distances grow as } \sqrt{N} \textbf{, what is countereffect to the curse of dimensionality.}$
> - Apart from everything, what we have written in our original answer, let us realize also the following fact. As distances between samples grow, grow also the distances of samples from decision boundary (*). This means that in order to obtain reasonable certified radiuses in comparison to the possible (real) robust radiuses, we need to increase the smoothing variance $\sigma$ (or $\sigma_b$) also roughly as $c\sqrt{N}$, where $c$ is scaling constant. However, in the input-dependent regime, if we want to do smoothing on bigger $\sigma_b$, we need to adjust the rate $r$ even more. This is because our aim is that in the distance of maximal possible certified radius ($R_M$) using constant $\sigma_b$ (given number of Monte-Carlo samplings), we want the expression $\sigma_0 \exp(\pm rR_M)$ from Theorem 7 to be close-enough to $\sigma_0$ so that we can still certify such a distance (otherwise from Theorem 4 we would fail to certify such a distance). Since the $R_M$ is linearly dependent on $\sigma_b$, it will be bigger for bigger $\sigma_b$. For this reason, we can't decrease the rate $r$ just as $1/\sqrt{N}$, but as $1/N$ (taking into account, that we need the $\sigma_0 \exp(\pm rR_M)$ close-enough to $\sigma_0$ for increasing distance $R_M$). So in the end, the possible ratio between $\sigma(x)$ for different samples would decrease as $1/\sqrt{N}$, given that we want to increase $\sigma_b$ with dimension, what we should, as explained above and below. We also already explain this phenomenon indirectly in Appendix B.3 with visual demonstrations. $\textbf{We also added Appendix B.4 with very detailed elaboration on this phenomenon}$
> $\textbf{and adjusted the last paragraph of Section 2 once again.}$
>
>
> (*) At least in practice. Assume we increase dimension 100 times. Since samples are now 10x more far away from each other, if the distances of samples from the decision boundaries of networks should remain constant, it would mean that on a line connecting two very close samples, network should change the prediction around 10 times. From all the evidence we have, this networks don't usually do, at least in practice. In other words, if distances between samples scale, also the areas of decision boundaries of the network usually scale.

---

### Author Response · Authors · 2021-11-19
**Summary of changes to the paper**

Dear reviewers, area chairs and program chairs,

we would like to summarize the changes we made to the paper during the rebuttal period. All the changes followed directly or indirectly from wishes / remarks of reviewers.

$\textbf{MAJOR:}$

$\textbf{Discussion on the total effect of the curse of dimensionality on the IDRS.}$
- As reviewer j688 pointed out, we missed a discussion about how the effect of distances increasing with dimension as $\sqrt{N}$ affect the curse of dimensionality. Now, we added last paragraph of Section 2 and Appendix B.4 to discuss closely this effect. We also realized another effect during our discussion, which is also included in the analysis. As a result, we arrive at (approximate) asymptotic behavior of the total variability of $\sigma(x)$. Please, read the two texts if you are interested in this discussion.

$\textbf{Experiments in the main text for MNIST changed from $r=0.01$ to $r=0.02$}$
- We made this change to demonstrate that we can afford stronger results (and stronger outperformance over constant smoothing) for MNIST, since the dimension enables us to use $r=0.02$ too.

$\textbf{Experiments on multi-dimensional toy dataset as a proof of concept}$
- Reviewer Q6Nf kindly asked us to provide visualizations on toy datasets for increasing dimension $N$. We add Appendix A.5, where we provide such experiments. Please have look at the results of these experiments if you want to see, how can IDRS outperform constant smoothing in simpler datasets with suitable design, even in the presence of the curse of dimensionality.

$\textbf{Discussion on hyperparameter selection}$
- Upon question of reviewer UXiJ, we add Appendix E.1 to discuss the hyperparameter selection.

$\textbf{MINOR:}$
- We change the formulation in abstract so that it doesn't sound, that IDRS is able to solve all the problems of RS.
- We fix the colors in Figure 4 and adjust Tables 2 and 3 to include information about dataset.
- We change the formulation of experiment setup so that it is more clear how the level of $\sigma_{tr}$ is being chosen.

Apart from that, we also made several very small changes of formulations to make some sentences more compact.

---

### Decision · Program_Chairs · 2022-01-20

**Decision:**

Reject

**Comment:**

The paper considers input-dependent randomized smoothing to obtain certified robust classification. The main contribution is the derivation of necessary conditions on how the variance of the smoothing distributions (assumed to be spherically symmetric Gaussian distributions) has to change to achieve certified robustness. All reviewers like this result, as it provides guidance on designing input-dependent smoothing, which is an interesting result for the community and certainly helps future research.

On the negative side, the smoothing method derived based on the theory provides little (if at all) improvement in practice, it cannot be scaled to higher dimensions, it does not address the problems it claims to address (the "waterfall" effect, as also admitted by the authors in the discussion), and the presentation should be significantly improved.

The paper received mixed reviews. While I think that the presented theoretical results are useful and interesting, the problems mentioned above make me to side with the negative reviewers and suggest rejection of the paper at this point (although this was not an easy decision).

While this is only lightly touched in the reviews, I strongly recommend the authors to make the presentation of the theoretical results more comprehensible. It is quite hard to follow the paper as notation is introduced continuously in an ad-hoc and confusing way (e.g., in the proof of Theorem 2, $a$ denotes $\delta$ and $\|\delta\|$), and things are often not adequately defined (e.g., the certified robust radius is not defined formally; in Lemma 1, $x$ is undefined and used for $x_0$ as well as a free parameter, $\chi_N^2$ is only implicitly defined, etc.)